# Lattice T-duality from non-invertible symmetries in quantum spin chains

**Salvatore D. Pace[1], Arkya Chatterjee[1] and Shu-Heng Shao[1,2]**

**1** Department of Physics, Massachusetts Institute of Technology, Cambridge, MA 02139, USA
**2** Yang Institute for Theoretical Physics, Stony Brook University, Stony Brook, NY 11794, USA

## Abstract

Dualities of quantum field theories are challenging to realize in lattice models of qubits. In this work, we explore one of the simplest dualities, T-duality of the compact boson CFT, and its realization in quantum spin chains. In the special case of the XX model, we uncover an exact lattice T-duality, which is associated with a non-invertible symmetry that exchanges two lattice U(1) symmetries. The latter symmetries flow to the momentum and winding U(1) symmetries with a mixed anomaly in the CFT. However, the charge operators of the two U(1) symmetries do not commute on the lattice and instead generate the Onsager algebra. We discuss how some of the anomalies in the CFT are nonetheless still exactly realized on the lattice and how the lattice U(1) symmetries enforce gaplessness. We further explore lattice deformations preserving both U(1) symmetries and find a rich gapless phase diagram with special $\mathrm{Spin}(2k)_1$ WZW model points and whose phase transitions all have dynamical exponent $z > 1$.

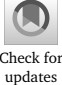

# 1  Introduction

Dualities in quantum field theories and lattice models are a cornerstone of contemporary theoretical physics, signaling that two seemingly distinct theories are secretly the same. They emphasize how it is insufficient to organize the space of quantum field theories/lattice models by Lagrangians/Hamiltonians and related degrees of freedom. They further act as a crucial aid in calculations since certain physical observables become manifest only after utilizing a duality transformation.

When discussing duality, however, one must specify what is meant by two theories being "the same." There are three primary ways in which the word duality is used throughout the literature:

1. The strongest form of duality is exact duality, which states that two theories are dual if they are isomorphic. Namely, for a given boundary condition, there is a one-to-one mapping between *every* state in the Hilbert space, *every* operator acting on *all* of these states, the models' *entire* spectra, and the correlation functions of *all* observables. This is the notion of duality that we adopt in this paper, and we will refer to exact duality just as duality from here on. Therefore, two dual theories are simply different presentations of the same physical model. Standard examples of such dualities include T-dualities [1], S-dualities (*e.g.*, electromagnetic duality) [2], dualities of free theories [3], and level-rank dualities [4–6].

2. A weaker version of duality is that of IR duality. Two theories that are IR dual are distinct in the UV but flow to the same theory in the IR. The theories they flow to in the IR could be the same or distinct (dual) presentations of the IR theory. Some prototypical examples are particle-vortex duality [7,8], its generalizations [9–11], and dualities for $3+1$ dimensional $\mathcal{N}=1$ supersymmetric gauge theories (*e.g.*, Seiberg duality) [12].

3. An unrelated notion of duality is discrete gauging (*e.g.*, orbifolding).[1] Gauging a discrete symmetry generically relates two theories that are globally distinct, both in the UV *and* IR. However, while distinct, the correlation functions and spectra of one model can be deduced from that of the other by inserting various symmetry defects (*i.e.*, changing boundary conditions). Discrete gauging, therefore, implements an isomorphism between two families of models and their combined states and spectra subject to different boundary conditions, but not between one theory and another. Alternatively, by restricting to fixed symmetry charge sectors (e.g., their symmetric subspaces) of two models related by discrete gauging with appropriate symmetry defect insertions, gauging then acts bijectively on states and operators for these fixed sectors.[2] Well-known examples of discrete gauging maps include the Kramers-Wannier transformation[3] [16] and the Kennedy-Tasaki transformation [17].

These three notations are generically unrelated. However, there can be instances where they coincide. In particular, two theories related by discrete gauging *may* also be dual or IR dual. Importantly, however, in such cases the gauging map and duality transformations between states and operators are different. For instance, the latter map is always bijective while the former is not.

## 1.1 T-duality

One of the simplest dualities in quantum field theory is T-duality. In its most general form, it is a duality between particular nonlinear sigma models with different target spaces and

---

[1]Discrete gauging is sometimes referred to as just duality, especially in the condensed matter and quantum information literature (*e.g.*, see Ref. 13). A reader familiar with category theory may be inclined to call it *Morita* duality to distinguish it from the meaning of the word duality we use here. The adjective Morita emphasizes that the (higher-)category describing the symmetry of the theory obtained after discrete gauging is Morita equivalent to the original theory's symmetry [14].

[2]For gauge-related models with appropriate boundary conditions that are projected into corresponding fixed charge sectors, the gauging maps become exact duality transformations. However, these projected models are non-local due to the fixed charge sector constraint involving a non-local operator—the symmetry operator. From a continuum field theory point of view, these models violate modular invariance and Haag duality and/or additivity (see, e.g., Ref. 15).

[3]In this context, the Kramers-Wannier duality refers to a transformation between the ordered and disordered phases of the $1+1$D Ising model. It is implemented by gauging the model's $\mathbb{Z}_2$ symmetry. On the other hand, there is also an exact duality that maps the ordered/disordered phase of the Ising model to the disordered/ordered phase of the Ising model *coupled to a $\mathbb{Z}_2$ gauge field*. This exact duality is also sometimes called Kramers-Wannier duality. In the Hamiltonian formalism, it is implemented by adding ancillas and then performing a unitary transformation.

backgrounds. The simplest example of this is the T-duality of a compact boson, which relates a $1+1$D sigma model whose target space is $S^1$ with radius $R$ to another sigma model whose target space is $S^1$ with radius $1/R$ [18,19]. In terms of their Euclidean Lagrangians, this means that

$$\mathcal{L}_R = \frac{R^2}{4\pi}\partial_\mu\Phi\partial^\mu\Phi \xleftrightarrow{\text{T-duality}} \mathcal{L}_{1/R} = \frac{1}{4\pi R^2}\partial_\mu\Theta\partial^\mu\Theta\,, \tag{1}$$

where $\Phi \sim \Phi + 2\pi$ and $\Theta \sim \Theta + 2\pi$ are compact boson fields. A further defining feature of this T-duality is that it exchanges the U(1) momentum and U(1) winding symmetries[4] of $\mathcal{L}_R$ and $\mathcal{L}_{1/R}$. In terms of their conserved currents, this means that

$$
\begin{array}{ll}
\dfrac{iR^2}{2\pi}\partial_\mu\Phi & \text{(momentum)} \\[2mm]
\dfrac{\epsilon_{\mu\nu}}{2\pi}\partial^\nu\Phi & \text{(winding)}
\end{array}
\quad \xleftrightarrow{\text{T-duality}} \quad
\begin{array}{ll}
\dfrac{\epsilon_{\mu\nu}}{2\pi}\partial^\nu\Theta & \text{(winding)}, \\[2mm]
\dfrac{i}{2\pi R^2}\partial_\mu\Theta & \text{(momentum)}.
\end{array}
\tag{2}
$$

T-duality of a compact boson has played a significant role in understanding string theory on a circle and has found diverse applications in condensed matter theory. As it will be the focus of this paper, from here on we will implicitly refer to the T-duality of a compact boson as just T-duality. We refer the reader to Section 2 for a review of the compact free boson, its T-duality, and its symmetries.

T-duality is not unique to the continuum and can also arise in lattice regularizations of the compact free boson [20–23] (see Refs. 24, 25 for similar dualities in related lattice models). These models with lattice T-duality flow to the compact boson in the IR and have exact U(1) momentum and winding symmetries on the lattice. The constructions of these theories are often directly inspired by the continuum field theory. For instance, they can be obtained by putting the compact free boson on a Euclidean spacetime lattice or are modified Villain lattice models that have infinite-dimensional local Hilbert spaces when quantized.

The traditional perspective of T-duality is based on the $\mathcal{L}_R \to \mathcal{L}_{1/R}$ transformation and exchange of momentum and winding symmetries. However, a modern and more invariant perspective of T-duality uses non-invertible symmetries of the compact free boson (see Refs. [26–29] for reviews on non-invertible symmetry). For example, when $R^2 = N \in \mathbb{Z}_{>0}$, the compact free boson has a non-invertible symmetry that implements the $\mathbb{Z}_N$ momentum symmetry gauging map followed by the T-duality transformation [30–33]. The non-invertible symmetry operator $\mathcal{D}$ acts on the momentum and winding symmetry charges, $\mathcal{Q}^\mathcal{M}$ and $\mathcal{Q}^\mathcal{W}$ respectively, by

$$\mathcal{D}\,\mathcal{Q}^\mathcal{M} = N\,\mathcal{Q}^\mathcal{W}\mathcal{D}\,, \qquad \mathcal{D}\,\mathcal{Q}^\mathcal{W} = \frac{1}{N}\,\mathcal{Q}^\mathcal{M}\mathcal{D}\,. \tag{3}$$

The existence of the two U(1) symmetries generated by $\mathcal{Q}^\mathcal{M}$ and $\mathcal{Q}^\mathcal{W}$ alongside the non-invertible symmetry $\mathcal{D}$ satisfying (3) implies T-duality, and can further serve as its definition. Generalizations of the non-invertible symmetry $\mathcal{D}$ arising from T-duality have been proposed for arbitrary radii $R$ [34]. However, for simplicity, we will specialize to T-duality associated with $R^2 = N \in \mathbb{Z}_{>0}$ (in particular, $R^2 = 2$).

Adopting this symmetry-based definition of T-duality is especially useful in the context of lattice models. Indeed, consider a $1+1$D lattice model that bears no resemblance to the compact free boson yet flows to it in the IR. It is entirely possible that despite its differing appearance to the continuum, the lattice model can have two U(1) symmetries that become the momentum and winding symmetries in the continuum and have a non-invertible symmetry implementing a lattice T-duality as in (3). This then begs the question: there are many known

---

[4]For the compact free boson at radius $R$, the U(1) momentum and U(1) winding symmetries are shift symmetries of $\Phi$ and $\Theta$, respectively. The terminology "momentum" and "winding" symmetries are used in the string theory literature, and they are sometimes respectively referred to as "charge" and "vortex" symmetries.

lattice models that flow to the compact free boson, but do any have lattice T-duality? In particular, what about the simplest quantum lattice model of qubits on a chain?

## 1.2 Summary

In this paper, we explore lattice T-duality in $1+1$D lattice models of qubits. We start in Section 3 by considering the XX model, a quantum spin chain model whose Hamiltonian is

$$H_{\text{XX}} = \sum_{j=1}^{L} \left( X_j X_{j+1} + Y_j Y_{j+1} \right). \tag{4}$$

Unless otherwise specified, throughout this paper, we assume that $L$ is even for all quantum spin chain models considered. It is well known that the IR of $H_{\text{XX}}$ is described by the compact free boson at radius $R = \sqrt{2}$.[5] The integer-quantized operator $Q^{\text{M}} = \frac{1}{2} \sum_{j=1}^{L} Z_j$ commutes with $H_{\text{XX}}$ and is the generator of a $U(1)^{\text{M}}$ symmetry that becomes the momentum symmetry of the compact free boson in the IR.[6] The XX model Hamiltonian also commutes with the integer-quantized charge [36–39]

$$Q^{\text{W}} = \frac{1}{4} \sum_{n=1}^{L/2} (X_{2n-1} Y_{2n} - Y_{2n} X_{2n+1}). \tag{5}$$

This operator is the generator of another $U(1)$ symmetry of the XX model. In fact, we show that it becomes the $U(1)^{\text{W}}$ winding symmetry of the $R = \sqrt{2}$ compact free boson in the IR. However, unlike in the continuum, the lattice charges $Q^{\text{M}}$ and $Q^{\text{W}}$ do not commute: $[Q^{\text{M}}, Q^{\text{W}}] \neq 0$.

Given the existence of two $U(1)$ symmetries of the XX model that respectively flow to the momentum and winding symmetries of the compact free boson, it is then natural to ask whether the XX model has a lattice T-duality. From the symmetry perspective advocated above, this would mean that the XX model has a non-invertible symmetry that exchanges $Q^{\text{M}}$ and $Q^{\text{W}}$ (with appropriate factors of $R^2 = 2$). This non-invertible symmetry would implement the gauging map of the lattice $\mathbb{Z}_2$ momentum symmetry $\mathrm{e}^{\mathrm{i}\pi Q^{\text{M}}}$ and then the lattice T-duality transformation.

We show that the XX model does, in fact, have lattice T-duality. To do so, we first gauge the $\mathbb{Z}_2$ symmetry generated by $\mathrm{e}^{\mathrm{i}\pi Q^{\text{M}}}$ and find a basis in which the gauged XX model Hamiltonian becomes

$$H_{\text{XX}/\mathbb{Z}_2^{\text{M}}} = \sum_{j=1}^{L} (X_j + Z_{j-1} X_j Z_{j+1}). \tag{6}$$

The key observation is that there exists a unitary operator $U_{\text{T}}$ for which $H_{\text{XX}} = U_{\text{T}} H_{\text{XX}/\mathbb{Z}_2^{\text{M}}} U_{\text{T}}^{-1}$. Therefore, there exists a non-invertible operator $\mathsf{D}$ that commutes with the Hamiltonian,

$$\mathsf{D} H_{\text{XX}} = \left( U_{\text{T}} H_{\text{XX}/\mathbb{Z}_2^{\text{M}}} U_{\text{T}}^{-1} \right) \mathsf{D} = H_{\text{XX}} \mathsf{D}, \tag{7}$$

such that $U_{\text{T}}^{-1} \mathsf{D}$ implements the $\mathrm{e}^{\mathrm{i}\pi Q^{\text{M}}}$ gauging map. Crucially, we find that this non-invertible symmetry operator satisfies

$$\mathsf{D} Q^{\text{M}} = 2 Q^{\text{W}} \mathsf{D}, \qquad \mathsf{D} Q^{\text{W}} = \frac{1}{2} Q^{\text{M}} \mathsf{D}. \tag{8}$$

---

[5] Our convention for $R$ is chosen such that $R = 1$ is the self-dual, $SU(2)_1$ WZW model point of the compact free boson. This convention differs from Ginsparg's [35] whose radius $R_{\text{Ginsparg}}$ is related to ours by $R = \sqrt{2} R_{\text{Ginsparg}}$.

[6] We generally use calligraphic and ordinary fonts for operators in the continuum and on the lattice, respectively.

Therefore, just as in the continuum theory (*i.e.*, Eq. (3) with $R^2 = 2$), the XX model has T-duality. In particular, the lattice T-duality transformation is implemented by the unitary operator $U_T$. Since unitary transformations preserve the spectrum of the Hamiltonian, act bijectively on states in the Hilbert space, and preserve the operator algebra, this is an exact duality.

Having found new symmetries in the XX model that exactly match those in the continuum compact boson—the symmetries formed by $e^{i\theta Q^W}$ and D—we then investigate their anomalies in Section 4. Anomalies in the XX model are well known, in particular the Lieb-Schultz-Mattis anomalies involving translations and the $U(1)^M \rtimes \mathbb{Z}_2^C$ symmetry formed by $e^{i\phi Q^M}$ and $\prod_{j=1}^L X_j$ [22, 40–44]. With the identification of $Q^W$ and D, we find new 't Hooft anomalies. In particular, while $Q^M$ and $Q^W$ do not commute, the symmetry operators $e^{i\pi Q^M}$ and $e^{i\theta Q^W}$ do. Therefore, there is a $\mathbb{Z}_2^W \times \mathbb{Z}_2^C \ltimes U(1)^M$ sub-symmetry of the XX model, and we show that its anomalies manifest on the lattice the same way as they do in the continuum. Firstly, the sub-symmetry $\mathbb{Z}_2^W \times U(1)^M$ exhibits a spectral flow where inserting a $2\pi \, U(1)^M$ symmetry defect changes the $\mathbb{Z}_2^W$ charge of states (*i.e.*, $\mathbb{Z}_2^W$ charge pumping upon adiabatically inserting a $2\pi$ flux of $U(1)^M$ in the SPT in one-higher dimension). Secondly, the $\mathbb{Z}_2^W \times \mathbb{Z}_2^C \times \mathbb{Z}_2^M$ sub-symmetry exhibits a type III anomaly where inserting a symmetry defect of one $\mathbb{Z}_2$ causes the other two $\mathbb{Z}_2$ sub-symmetries to be projectively represented. We further prove that any deformation of the XX model that preserves the $U(1)$ symmetries generated by $Q^M$ and $Q^W$ does not gap out the Hamiltonian. In other words, just like the perturbative anomaly in the compact free boson, the lattice $U(1)^M$ and $U(1)^W$ symmetries enforce gaplessness.

The unitary symmetries and their interplay with the non-invertible symmetry arising from T-duality are much richer on the lattice than in the continuum. This stems from the fact that while the momentum and winding symmetries commute in the continuum, $Q^M$ and $Q^W$ do not commute on the lattice. Instead, they generate a large Lie algebra known as the Onsager algebra [36, 37, 39]. The Onsager charges $Q_n$ and $G_n$, with $n$ an integer, are quantized and satisfy

$$[Q_n, Q_m] = i G_{m-n}, \qquad [G_n, G_m] = 0, \qquad [Q_n, G_m] = 2i(Q_{n-m} - Q_{n+m}), \qquad (9)$$

where $Q_0 = Q^M$ while $Q_1 = 2Q^W$. In Section 5, we find closed expressions for $Q_n$ and $G_n$ in terms of $Q^M$ and $Q^W$ and explore their interplay with the non-invertible symmetry operator D. For instance, we find that

$$D Q_n = Q_{1-n} D, \qquad D G_n = -G_n D. \qquad (10)$$

The transformation (8) signaling T-duality is then the special case of (10) for $n = 0, 1$. We further show that for finite $n$ in the IR limit, once the momentum and winding charges commute,

$$Q_n \xrightarrow{\text{IR limit}} \begin{cases} 2\mathcal{Q}^W, & n \text{ odd}, \\ \mathcal{Q}^M, & n \text{ even}, \end{cases} \qquad G_n \xrightarrow{\text{IR limit}} 0. \qquad (11)$$

In particular, while the lattice charges $Q_n$ don't commute with each other, their IR limits do: $[\mathcal{Q}^M, \mathcal{Q}^W] = 0$.

In Section 6, we discuss the fate of lattice T-duality upon deforming the XX model. Generic deformations will break the lattice momentum and/or winding symmetries and destroy the corresponding T-duality. For example, in the XYZ chain, the only points in the phase diagram with a lattice T-duality are those for which the XYZ chain is unitarily related to the XX model. However, in Section 6.2, we find the most general class of qubit models with lattice momentum and winding symmetries and corresponding T-duality. Due to the gapless constraint we prove, these Hamiltonians are all gapless. For example, the simplest one-parameter family of Hamiltonians is

$$H(g_2) = \sum_{j=1}^L \left( X_j X_{j+1} + Y_j Y_{j+1} + g_2 \left( Y_{j-1} Z_j X_{j+1} - X_{j-1} Z_j Y_{j+1} \right) \right). \qquad (12)$$

We find the phase diagram of this Hamiltonian plus the next simplest deformation exactly by fermionizing. As shown in Fig. 5, all phases are gapless and there are various phase transitions all with dynamical exponent $z > 1$. There are also special points described by $\mathrm{Spin}(2k)_1$ WZW models.

## 2 Review of the compact free boson

In this section, we review T-duality in the $c = 1$ compact scalar boson conformal field theory (CFT)—the compact free boson. We work in Euclidean signature and denote by $M_2$ two-dimensional Euclidean spacetime. We refer the reader to Ref. 31 for a contemporary, detailed introduction to the compact free boson.

The partition function of the compact free boson at radius $R$ is

$$\mathcal{Z}_R[M_2] = \int \mathcal{D}\Phi \, e^{-S_R[M_2, \Phi]} \,, \qquad S_R[M_2, \Phi] = \frac{R^2}{4\pi} \int_{M_2} \partial_\mu \Phi \, \partial^\mu \Phi \, \mathrm{d}^2 x \,, \tag{13}$$

where $\Phi \sim \Phi + 2\pi$ is a compact scalar boson. Since $\Phi$ is multi-valued, it satisfies the quantization condition $\int_C \partial_\mu \Phi \, \mathrm{d}x^\mu \in 2\pi\mathbb{Z}$ for all cycles $C$ of $M_2$. Furthermore, it is differentiable only locally. In particular, while $\mathrm{d}\Phi \equiv \partial_\mu \Phi \, \mathrm{d}x^\mu$ is globally a closed 1-form (*i.e.*, $\epsilon^{\mu\nu}\partial_\mu \partial_\nu \Phi = 0$), it is not globally an exact 1-form. For our purposes, it is sufficient to handle this by viewing $\partial_\mu \Phi$ as a shorthand for $\partial_\mu \widetilde{\Phi} + 2\pi\omega_\mu$, where $\widetilde{\Phi}$ is a real scalar field and $\omega_\mu$ is a representative of the de Rham cohomology class $[\omega] \in H^1_{\mathrm{dR}}(M_2)$ satisfying $\epsilon^{\mu\nu}\partial_\nu \omega = 0$ and $\int_C \omega_\mu \mathrm{d}x^\mu \in \mathbb{Z}$.[7]

### 2.1 T-duality

The compact free boson at radii $R$ and $1/R$ are equivalent, and the duality between these different presentations of the CFT is T-duality. To see this equivalence, let us consider the partition function

$$\int \mathcal{D}\Theta \, \mathcal{D}W_\mu \, \exp\left[ -\int_{M_2} \left( \frac{R^2}{4\pi} W_\mu W^\mu - \frac{\mathrm{i}}{2\pi} \epsilon^{\mu\nu} \partial_\mu \Theta \, W_\nu \right) \mathrm{d}^2 x \right] \,, \tag{14}$$

where $\Theta \sim \Theta + 2\pi$ is a compact scalar boson. This compact boson is a Lagrange multiplier field. In particular, treating $\partial_\mu \Theta$ as shorthand for $\partial_\mu \widetilde{\Theta} + 2\pi\eta_\mu$, integrating out $\widetilde{\Theta}$ enforces $\epsilon^{\mu\nu}\partial_\mu W_\nu = 0$ and summing over $[\eta]$ enforces $\int_C W_\mu \mathrm{d}x^\mu \in 2\pi\mathbb{Z}$. These constraints are solved by $W_\mu = \partial_\mu \Phi$, with $\Phi \sim \Phi + 2\pi$, from which it is clear that the partition functions (13) and (14) both describe the compact free boson. Integrating out $W_\mu$ in (14), on the other hand, yields

$$\mathcal{Z}_{1/R}[M_2] = \int \mathcal{D}\Theta \, e^{-S_{1/R}[M_2, \Theta]} \,, \qquad S_{1/R}[M_2, \Theta] = \frac{1}{4\pi R^2} \int_{M_2} \partial_\mu \Theta \, \partial^\mu \Theta \, \mathrm{d}^2 x \,, \tag{15}$$

which is the compact free boson at radius $1/R$. Therefore, while the classical actions $S_R$ and $S_{1/R}$ differ, the partition functions

$$\mathcal{Z}_R[M_2] = \mathcal{Z}_{1/R}[M_2] \,. \tag{16}$$

---

[7]The quantity $(\partial_\mu \widetilde{\Phi} + 2\pi\omega_\mu)\mathrm{d}x^\mu$ is the Hodge decomposition of the closed 1-form $\mathrm{d}\Phi$. The notion of $\Phi$ being differentiable only locally is made precise by treating $\Phi$ as the map $\Phi: M_2 \to \mathbb{R}/2\pi\mathbb{Z}$ and using Čech cohomology.

More precisely, T-duality is a bijective map from the states and operators of the compact free boson in the $\Phi$ presentation (13) to those in the $\Theta$ presentation (15). By inserting operators in the $\Phi$ presentation partition function (13) and repeating the above manipulations, we can derive how local operators of $\Phi$ are related to those of $\Theta$. For example, inserting $\partial_\mu \Phi$ in (13) is equivalent to inserting $W_\mu$ into (14), and by then integrating out $W_\mu$ we arrive at the partition function (15) with $-\frac{i}{R^2}\epsilon_{\mu\nu}\partial^\nu\Theta$ inserted. Therefore,

$$\partial_\mu \Phi \xrightarrow{\text{T-duality}} -\frac{i}{R^2}\epsilon_{\mu\nu}\partial^\nu\Theta\,. \tag{17}$$

Using the T-duality map, we can also relate the symmetries of the compact free boson in the $\Phi$ presentation to those in the $\Theta$ presentation. For example, the CFT has a $U(1)^{\mathcal{M}} \times U(1)^{\mathcal{W}}$ symmetry whose conserved currents in the $\Phi$ presentation are

$$J_\mu^{(\mathcal{M})} = i\frac{R^2}{2\pi}\partial_\mu\Phi\,, \qquad J_\mu^{(\mathcal{W})} = \frac{1}{2\pi}\epsilon_{\mu\nu}\partial^\nu\Phi\,. \tag{18}$$

On the other hand, these currents in the $\Theta$ presentation are

$$\widehat{J}_\mu^{(\mathcal{M})} = \frac{i}{2\pi R^2}\partial_\mu\Theta\,, \qquad \widehat{J}_\mu^{(\mathcal{W})} = \frac{1}{2\pi}\epsilon_{\mu\nu}\partial^\nu\Theta\,. \tag{19}$$

Therefore, using (17), we see that T-duality maps the momentum and winding $U(1)$ symmetries in the $\Phi$ presentation to the winding and momentum $U(1)$ symmetries, respectively, in the $\Theta$ presentation.

The local primary operators of the compact free boson are $V_{n,w} \equiv e^{in\Phi}e^{iw\Theta}$,[8] and their conformal weights are

$$h = \frac{1}{4}\left(\frac{n}{R} + wR\right)^2\,, \qquad \bar{h} = \frac{1}{4}\left(\frac{n}{R} - wR\right)^2\,. \tag{20}$$

Therefore, their scaling dimension and spin are $\Delta = h + \bar{h} \equiv \frac{1}{2}\left(\frac{n^2}{R^2} + w^2 R^2\right)$ and $s \equiv h - \bar{h} = nw$. The $U(1)^{\mathcal{M}}$ and $U(1)^{\mathcal{W}}$ symmetries are shift symmetries of $\Phi$ and $\Theta$, respectively and act on the local primary operator as

$$U(1)^{\mathcal{M}}: V_{n,w} \to e^{in\phi}\,V_{n,w}\,, \tag{21}$$

$$U(1)^{\mathcal{W}}: V_{n,w} \to e^{iw\theta}\,V_{n,w}\,. \tag{22}$$

These transformations commute and are described by the product group $U(1)^{\mathcal{M}} \times U(1)^{\mathcal{W}}$. Furthermore, the compact free boson has a $\mathbb{Z}_2^{\mathcal{C}}$ charge conjugation symmetry that transforms the momentum and winding currents by

$$\mathbb{Z}_2^{\mathcal{C}}: (J^{(\mathcal{M})}, J^{(\mathcal{W})}) \to (-J^{(\mathcal{M})}, -J^{(\mathcal{W})})\,. \tag{23}$$

This transformation, of course, also applies to the currents $\widehat{J}$ in the $\Theta$ presentation. Therefore, it transforms that local primary operators as $V_{n,w} \to V_{-n,-w}$, which in the $\Phi$ and $\Theta$ presentations implies $\Phi \to -\Phi$ and $\Theta \to -\Theta$. Since $\mathbb{Z}_2^{\mathcal{C}}$ transforms the $U(1)^{\mathcal{M}}$ and $U(1)^{\mathcal{W}}$ currents, the total group describing these symmetries is

$$(U(1)^{\mathcal{M}} \times U(1)^{\mathcal{W}}) \rtimes \mathbb{Z}_2^{\mathcal{C}}\,. \tag{24}$$

---

[8]Writing the local primary operator $V_{n,w}$ as $V_{n,w} \equiv e^{in\Phi}e^{iw\Theta}$ is a bit of an abuse of notation. Denoting the T-duality map by T, it is more precise to write $V_{n,w}$ in the $\Phi$ and $\Theta$ presentations of the CFT as $e^{in\Phi}\,\mathrm{T}^{-1}(e^{iw\Theta})$ and $\mathrm{T}(e^{in\Phi})\,e^{iw\Theta}$, respectively.

## 2.2 't Hooft anomalies

The symmetry (24) has 't Hooft anomalies involving its sub-symmetries. These 't Hooft anomalies obstruct the symmetry (24) from being gauged and preclude a unique gapped symmetric ground state. They are most conveniently labeled by SPT theories in $2+1$D Euclidean spacetime $M_3$ that serve as their anomaly inflow theories. Here, we characterize these SPTs by their response theory, the invertible topological field theory that describes the universal, long-wavelength properties of the corresponding SPT phase.

There is a mixed 't Hooft anomaly between the momentum and winding symmetries. It corresponds to the SPT

$$\mathcal{Z}[A^{\mathcal{M}}, A^{\mathcal{W}}, M_3] = \exp\left[\frac{i}{2\pi} \int_{M_3} \epsilon^{\mu\nu\rho} A_\mu^{\mathcal{M}} \partial_\nu A_\rho^{\mathcal{W}} \, d^3x\right], \tag{25}$$

where $A^{\mathcal{M}}$ and $A^{\mathcal{W}}$ are $U(1)^{\mathcal{M}}$ and $U(1)^{\mathcal{W}}$ background gauge fields, respectively. This anomaly manifests in $1+1$D through spectral flow [22, 45–47]. In particular, inserting a momentum symmetry defect labeled by an angle $\phi \in U(1)^{\mathcal{M}}$ modifies the winding symmetry charge $\mathcal{Q}^{\mathcal{W}} \equiv \int J_0^{\mathcal{W}} dx$ to [22][9]

$$\mathcal{Q}_\phi^{\mathcal{W}} = \mathcal{Q}^{\mathcal{W}} + \frac{\phi}{2\pi}. \tag{26}$$

In particular, the winding charge is no longer an integer in the presence of a nontrivial momentum symmetry defect. Dually, inserting a winding symmetry defect labeled by an angle $\theta \in U(1)^{\mathcal{W}}$ modifies the momentum symmetry charge $\mathcal{Q}^{\mathcal{M}} \equiv \int J_0^{\mathcal{M}} dx$ to

$$\mathcal{Q}_\theta^{\mathcal{M}} = \mathcal{Q}^{\mathcal{M}} + \frac{\theta}{2\pi}. \tag{27}$$

It follows that inserting a $2\pi$ winding (momentum) symmetry defect increases energy eigenstates' momentum (winding) charge by one. The $2+1$D SPT is a bosonic integer quantum Hall state [48], and the spectral flow arises from a $2\pi$ winding (momentum) symmetry flux pumping momentum (winding) charge. However, we emphasize that the $2+1$D bulk is not required for $2\pi$ momentum (winding) symmetry defects to increase winding (momentum) charge in $1+1$D (See Ref. 22 and Section 4.1).

There is also an 't Hooft anomaly involving $\mathbb{Z}_2^{\mathcal{C}}$ and the $\mathbb{Z}_2^{\mathcal{M}}$ and $\mathbb{Z}_2^{\mathcal{W}}$ sub-symmetries of $U(1)^{\mathcal{M}}$ and $U(1)^{\mathcal{W}}$. The corresponding SPT is

$$\mathcal{Z}[a^{\mathcal{M}}, a^{\mathcal{W}}, a^{\mathcal{C}}, M_3] = \exp\left[i\pi \int_{M_3} a^{\mathcal{C}} \cup a^{\mathcal{M}} \cup a^{\mathcal{W}}\right], \tag{28}$$

where each $a^{\bullet} \in Z^1(M_3, \mathbb{Z}_2)$ is a $\mathbb{Z}_2^{\bullet}$ gauge field and $\cup$ denotes the cup product. This 't Hooft anomaly, known as the type III anomaly [49, 50], manifests in $1+1$D by inserting a $\mathbb{Z}_2^{\bullet}$ symmetry defect (i.e., choosing $\mathbb{Z}_2^{\bullet}$ twisted boundary conditions) causing the other two $\mathbb{Z}_2$ symmetries to be represented projectively. For example, the $\mathbb{Z}_2^{\mathcal{M}} \times \mathbb{Z}_2^{\mathcal{W}}$ symmetry is projectively represented in the presence of a $\mathbb{Z}_2^{\mathcal{C}}$ symmetry defect. Relatedly, the $2+1$D SPT state described by (28) is characterized by $\mathbb{Z}_2^{\bullet}$ symmetry charges dressing trivalent junctions formed by the other two $\mathbb{Z}_2$ symmetry defects fusing (e.g., $\mathbb{Z}_2^{\mathcal{C}}$ charges dressing junctions formed by $\mathbb{Z}_2^{\mathcal{M}} \times \mathbb{Z}_2^{\mathcal{W}}$ symmetry defects fusing). Such a decoration pattern corresponds to, for instance, the nontrivial element of the group cohomology $H^2(\mathbb{Z}_2^{\mathcal{M}} \times \mathbb{Z}_2^{\mathcal{W}}, H^1(\mathbb{Z}_2^{\mathcal{C}}, U(1))) \simeq \mathbb{Z}_2$ in the Künneth decomposition of $H^3(\mathbb{Z}_2^{\mathcal{C}} \times \mathbb{Z}_2^{\mathcal{M}} \times \mathbb{Z}_2^{\mathcal{W}}, U(1))$ [51].

---

[9]The spectral flow equation $\mathcal{Q}_\phi^{\mathcal{W}} = \mathcal{Q}^{\mathcal{W}} + \frac{\phi}{2\pi}$ is derived by minimally coupling a background $U(1)^{\mathcal{M}}$ gauge field $\mathcal{A}$ to the winding current $J_\mu^{(\mathcal{W})}$. Doing so yields $J_\mu^{(\mathcal{W})}[\mathcal{A}] = \frac{1}{2\pi}\epsilon_{\mu\nu}(\partial^\nu\Phi - \mathcal{A}^\nu)$. The winding symmetry charge $\mathcal{Q}^{\mathcal{W}}$ then becomes $\mathcal{Q}_\phi^{\mathcal{W}} = \mathcal{Q}^{\mathcal{W}} + \frac{\phi}{2\pi}$, where the holonomy $\phi = -\int \mathcal{A}^x \, dx$.

## 2.3 Non-invertible symmetry

The compact free boson at $R^2 = N \in \mathbb{Z}_{>0}$ has a non-invertible symmetry arising from T-duality [30–33]. In particular, there is a Kramers-Wannier symmetry implementing a gauging map of a finite sub-symmetry of $U(1)^{\mathcal{M}} \times U(1)^{\mathcal{W}}$ followed by the T-duality map. From T-duality, however, this implies that the $R^2 = 1/N$ compact free boson has a non-invertible symmetry, too. In what follows, we will review this symmetry for $R^2 = N \in \mathbb{Z}_{>0}$, in which case the Kramers-Wannier transformation implements a $\mathbb{Z}_N^{\mathcal{M}}$ gauging map. We also refer the reader to Ref. 52 for discussion on non-invertible symmetries for multiple copies of compact bosons and Ref. 34 for a proposal of generalizations to arbitrary radius.

We now gauge the $\mathbb{Z}_N^{\mathcal{M}}$ sub-symmetry of $U(1)^{\mathcal{M}}$ in the $\Phi$ presentation of the compact free boson at radius $R$. We couple the compact boson to the $1 + 1D$ $\mathbb{Z}_N$ gauge theory

$$S_R[M_2, \Phi]_{/\mathbb{Z}_N^{\mathcal{M}}} = \int_{M_2} \left[ -\frac{R^2}{4\pi} \left( \partial_\mu \Phi + a_\mu \right)^2 + \frac{iN}{2\pi} \epsilon^{\mu\nu} a_\mu \partial_\nu b \right] d^2 x \,. \tag{29}$$

Here, $a_\mu$ is a $U(1)$ gauge field and $b$ is another compact scalar field (i.e., $b \sim b + 2\pi$). There is now a $U(1)$ gauge redundancy

$$\Phi \sim \Phi + \alpha \,, \qquad a_\mu \sim a_\mu - \partial_\mu \alpha \,. \tag{30}$$

Locally, integrating out $b$ enforces $a_\mu$ to be a $\mathbb{Z}_N$ gauge field, so that its holonomy is $\mathbb{Z}_N$-valued, $\int_C a_\mu \, dx^\mu \in 2\pi \mathbb{Z}/N$. Next, we introduce a new scalar field $\Phi^\vee$ that is invariant under the above gauge transformation. It is defined as

$$\partial_\mu \Phi^\vee = N(\partial_\mu \Phi + a_\mu) \,. \tag{31}$$

The normalization factor of $N$ is introduced so that $\int_C \partial_\mu \Phi^\vee \, dx^\mu \in 2\pi \mathbb{Z}$. In other words, $\Phi^\vee$ is a compact boson with period $2\pi$ (i.e., $\Phi^\vee \sim \Phi^\vee + 2\pi$). Writing the action in terms of the new compact boson field $\Phi^\vee$, it then becomes clear that gauging $\mathbb{Z}_N^{\mathcal{M}}$ reduces the radius from $R$ to $R/N$:

$$\mathcal{Z}_R[M_2] \xrightarrow{\text{Gauge } \mathbb{Z}_N^{\mathcal{M}}} \mathcal{Z}_{R/N}[M_2] \,. \tag{32}$$

After coupling to the gauge field $a_\mu$, the momentum and winding currents in the $\Phi$ and $\Phi^\vee$ presentations are

$$\begin{aligned}
J_\mu^{(\mathcal{M})} &= i \frac{R^2}{2\pi} \left( \partial_\mu \Phi + a_\mu \right) = iN \frac{(R/N)^2}{2\pi} \partial_\mu \Phi^\vee \,, \\
J_\mu^{(\mathcal{W})} &= \frac{1}{2\pi} \epsilon_{\mu\nu} (\partial^\nu \Phi + a^\nu) = \frac{1}{N} \frac{1}{2\pi} \epsilon_{\mu\nu} \partial^\nu \Phi^\vee \,.
\end{aligned} \tag{33}$$

Therefore, the momentum and winding currents at radius $R/N$ (in terms of $\Phi^\vee$) are the same as their $R$ counterparts (in terms of $\Phi$) but rescaled by $N$ and $1/N$, respectively.

Since the partition function after gauging is the compact free boson at a new radius, it is not invariant under this gauging of $\mathbb{Z}_N^{\mathcal{M}}$. However, when followed by the T-duality map, it yields the transformations

$$\begin{aligned}
\mathcal{Z}_R[M_2] &\xrightarrow{\text{Gauge } \mathbb{Z}_N^{\mathcal{M}}} \mathcal{Z}_{R/N}[M_2] \xrightarrow{\text{T-duality}} \mathcal{Z}_{N/R}[M_2] \,, \\
J^{(\mathcal{M})} &\xrightarrow{\text{Gauge } \mathbb{Z}_N^{\mathcal{M}}} N J^{(\mathcal{M})} \xrightarrow{\text{T-duality}} N J^{(\mathcal{W})} \,, \\
J^{(\mathcal{W})} &\xrightarrow{\text{Gauge } \mathbb{Z}_N^{\mathcal{M}}} \frac{1}{N} J^{(\mathcal{W})} \xrightarrow{\text{T-duality}} \frac{1}{N} J^{(\mathcal{M})} \,.
\end{aligned} \tag{34}$$

Therefore, the compact free boson is invariant under gauging $\mathbb{Z}_N^{\mathcal{M}}$ when $R = \sqrt{N}$. Consequently, it has a non-invertible symmetry. This Kramers-Wannier symmetry transforms the momentum and winding currents $J^{(\mathcal{M})}$ and $J^{(\mathcal{W})}$ to $N J^{(\mathcal{W})}$ and $\frac{1}{N} J^{(\mathcal{M})}$, respectively.

## 3 T-duality in the XX model

We now turn our attention to quantum lattice models. In particular, we consider quantum spin chains where a single qubit resides on each site $j$ of a one-dimensional periodic lattice. We always take the number of lattice sites $L$ to be an even integer. The Pauli operators $X_j$, $Z_j$, and $Y_j = iX_jZ_j$ act on the qubit at site $j$ and obey the periodic boundary conditions $X_{j+L} = X_j$, $Z_{j+L} = Z_j$. Furthermore, the total Hilbert space $\mathcal{H}$ admits the tensor product factorization

$$\mathcal{H} = \bigotimes_{j=1}^{L} \mathcal{H}_j \,, \qquad \mathcal{H}_j = \mathbb{C}^2 \,, \tag{35}$$

with each $\mathcal{H}_j$ describing the qubit at site $j$.

The particular quantum spin chain we study is the XX model,[10] whose Hamiltonian is

$$H_{\text{XX}} = \sum_{j=1}^{L} \left( X_j X_{j+1} + Y_j Y_{j+1} \right) . \tag{36}$$

This celebrated Hamiltonian has appeared throughout the literature as various limits of toy models used for studying quantum phases and their transitions [53] and is a prototypical model studied in quantum integrability [54].

Among its conserved quantities, the XX model has a well-known $U(1)^{\text{M}}$ global symmetry generated by

$$Q^{\text{M}} = \frac{1}{2} \sum_{j=1}^{L} Z_j \,. \tag{37}$$

Because we assume $L$ is an even integer, the eigenvalues of $Q^{\text{M}}$ are integers. The U(1) symmetry operator $e^{i\phi Q^{\text{M}}}$ acts on the Pauli operators by the spin rotation

$$e^{i\phi Q^{\text{M}}} \begin{pmatrix} X_j \\ Y_j \end{pmatrix} e^{-i\phi Q^{\text{M}}} = \begin{pmatrix} \cos(\phi) & -\sin(\phi) \\ \sin(\phi) & \cos(\phi) \end{pmatrix} \begin{pmatrix} X_j \\ Y_j \end{pmatrix} . \tag{38}$$

Therefore, the operator $X_j + iY_j$ carries $Q^{\text{M}} = +1$ charge. Another well-known symmetry of the XX model is the $\mathbb{Z}_2^{\text{C}}$ symmetry generated by

$$C = \prod_{j=1}^{L} X_j \,. \tag{39}$$

This symmetry operator acts on $X_j + iY_j$ by conjugation. Furthermore, the $Q^{\text{M}}$ charge transforms under it by $Q^{\text{M}} \to -Q^{\text{M}}$. Therefore, $e^{i\phi Q^{\text{M}}}$ and $C$ furnish a faithful representation of the group $U(1)^{\text{M}} \rtimes \mathbb{Z}_2^{\text{C}} \cong O(2)$.

The IR limit[11] of the XX model is described by the compact free boson (13) at radius $R = \sqrt{2}$ [55, 56]. This is equivalent to the $U(1)_4$ WZW CFT, which is the WZW model whose associated $2 + 1$D Chern-Simons theory is $U(1)_4$. In the IR, the lattice operator $X_j + iY_j$ flows to the primary $e^{i\Phi}$ (up to a multiplicative constant). Therefore, in the IR limit, the $U(1)^{\text{M}}$ and

---

[10]The XX model goes by different names within the literature, sometimes being called the isotopic XY model or just the XY model.

[11]By the IR limit, we mean we focus on the low-energy states—the energy eigenstates that lie within an $\mathcal{O}(L^0)$ energy window above the ground state—of the lattice model and then take the thermodynamic limit $L \to \infty$. For a gapped Hamiltonian, these low-energy states form the ground-state subspace. For a gapless Hamiltonian, they would be a collection of low-lying energy states.

$$( H_{\mathrm{XX}}, \; Q^{\mathrm{M}}, \; Q^{\mathrm{W}} ) \quad \xleftarrow{\; U_{\mathrm{T}} \;} \quad (H_{\mathrm{XX}/\mathbb{Z}_2^{\mathrm{M}}}, \; \widehat{Q}^{\mathrm{W}} = 2 Q^{\mathrm{W}}_{/\mathbb{Z}_2^{\mathrm{M}}}, \; \widehat{Q}^{\mathrm{M}} = \tfrac{1}{2} Q^{\mathrm{M}}_{/\mathbb{Z}_2^{\mathrm{M}}})$$

$$\Big\updownarrow \text{RG} \qquad\qquad\qquad\qquad\qquad\qquad\qquad \Big\updownarrow \text{RG}$$

$$(\mathcal{L}_{\sqrt{2}}, \; \mathcal{Q}^{\mathcal{M}}, \; \mathcal{Q}^{\mathcal{W}} ) \quad \xleftarrow{\; \text{T-duality} \;} \quad (\mathcal{L}_{1/\sqrt{2}}, \; \widehat{\mathcal{Q}}^{\mathcal{W}} = 2 \mathcal{Q}^{\mathcal{W}}_{/\mathbb{Z}_2^{\mathcal{M}}}, \; \widehat{\mathcal{Q}}^{\mathcal{M}} = \tfrac{1}{2} \mathcal{Q}^{\mathcal{M}}_{/\mathbb{Z}_2^{\mathcal{M}}})$$

Figure 1: T-duality in the XX model $H_{\mathrm{XX}}$ is implemented by the unitary operator $U_{\mathrm{T}}$ (55). It relates $H_{\mathrm{XX}}$ to its $\mathbb{Z}_2^{\mathrm{M}}$-gauged version $H_{\mathrm{XX}/\mathbb{Z}_2^{\mathrm{M}}}$. It further relates $\mathbb{Z}_2^{\mathrm{M}}$-gauged momentum and winding symmetries of the XX model to the winding and momentum symmetries, respectively. The lattice T-duality implemented by $U_{\mathrm{T}}$ precisely matches the T-duality of the compact free boson in the IR limit of the XX model.

$\mathbb{Z}_2^{\mathcal{C}}$ symmetries of the XX model become the $U(1)^{\mathcal{M}}$ and $\mathbb{Z}_2^{\mathcal{C}}$ symmetries of the compact free boson at $R = \sqrt{2}$ (this explains the superscripts M and C).

While the XX model realizes the momentum and charge conjugation symmetries of the CFT exactly on the lattice, it is common lore that it does not realize the winding symmetry. Accordingly, without a lattice winding symmetry, the XX model also fails to enjoy a lattice T-duality and any related non-invertible symmetries. However, this piece of lore is a bit surprising when viewing the XX model from an integrability point of view. With an extensive number of conserved charges, perhaps one of them generates a U(1) symmetry on the lattice that flows to the winding symmetry in the CFT. In what follows, we show that T-duality and a related U(1) winding symmetry, in fact, exist on the lattice. Fig 1 summarizes this lattice T-duality and its resemblance to the T-duality in the compact free boson.

### 3.1 Gauging $\mathbb{Z}_2$ momentum symmetry

To motivate T-duality in the XX model, recall that gauging the $\mathbb{Z}_2$ sub-symmetry of the U(1) momentum symmetry in the compact free boson at $R = \sqrt{2}$ leaves the CFT invariant after implementing T-duality (*i.e.*, gauging replaces $R = \sqrt{2}$ with $R/2 = 1/\sqrt{2} \equiv 1/R$, see Section 2.3). In light of this occurring in the IR, we now gauge the $\mathbb{Z}_2^{\mathrm{M}}$ symmetry of the XX model, which is generated by

$$\eta \equiv e^{i \pi Q^{\mathrm{M}}} = \prod_{j=1}^{L} i \, Z_j = \begin{cases} \prod_{j=1}^{L} Z_j, & L = 0 \bmod 4, \\ -\prod_{j=1}^{L} Z_j, & L = 2 \bmod 4. \end{cases} \tag{40}$$

In its current form, $\eta$ is not written in a manifestly onsite[12] manner since the local $i Z_j$ is of order 4 rather than 2. However, since we assume $L$ is even, it can be written as

$$\eta = \prod_{j=1}^{L} (-1)^j \, Z_j. \tag{41}$$

In this form, the local operators $\eta_j \equiv (-1)^j Z_j$ correctly square to one.

To gauge the $\mathbb{Z}_2^{\mathrm{M}}$ symmetry, we first introduce a qubit onto each link $\langle j, j+1 \rangle$ of the lattice, the collection of which plays the role of a $\mathbb{Z}_2$ gauge field. They are acted on by the Pauli operators $\widetilde{X}_{j,j+1}$ and $\widetilde{Z}_{j,j+1}$. The gauging procedure is implemented by enforcing the Gauss law

$$G_j = \widetilde{Z}_{j-1,j} \, \eta_j \, \widetilde{Z}_{j,j+1} = 1, \qquad \eta_j = (-1)^j Z_j. \tag{42}$$

---

[12] Here, we call a unitary symmetry $G$ onsite if its representation $U_g$ has the decomposition $U_g = \prod_{j=1}^{L} U_g^{(j)}$ with $U_g^{(j)}$ linear representations of $G$.

The Gauss operators $G_j$ are mutually commuting for all $j$ and are $\mathbb{Z}_2$ operators. The $\mathbb{Z}_2^{\mathrm{M}}$ symmetry operator (40) can be written as

$$\eta = \prod_{j=1}^{L} G_j \,. \tag{43}$$

Therefore, enforcing the Gauss law $G_j = 1$ projects states into the $\eta = 1$ subspace. Minimally coupling the XX Hamiltonian such that each $G_j$ commutes with it yields the new Hamiltonian

$$\sum_{j=1}^{L} (X_j \widetilde{X}_{j,j+1} X_{j+1} + Y_j \widetilde{X}_{j,j+1} Y_{j+1}) \,. \tag{44}$$

Notice that the lattice translation generated by $T\widetilde{T}$ (where $T X_j T^{-1} = X_{j+1}$, $\widetilde{T}\widetilde{X}_{j-1,j}\widetilde{T}^{-1} = \widetilde{X}_{j,j+1}$, etc) is not gauge-invariant since $T\widetilde{T} G_j (T\widetilde{T})^{-1} = -G_{j+1} \neq G_{j+1}$. However, the gauged model is still translation-invariant because $T\widetilde{T} \prod_j X_j$ is gauge-invariant and commutes with (44).

It is convenient to rotate the enlarged Hilbert space to a basis where the physical subspace has a tensor product factorization. To do so, we use the unitary operator

$$\prod_{j=1}^{L} \frac{1}{4} X_j^j (1 + X_j + \widetilde{Z}_{j,j+1} - X_j \widetilde{Z}_{j,j+1})(1 + X_j + \widetilde{Z}_{j-1,j} - X_j \widetilde{Z}_{j-1,j}) \,, \tag{45}$$

to implement the basis transformation

$$\begin{aligned}
Z_j &\to (-1)^j \, \widetilde{Z}_{j-1,j} Z_j \widetilde{Z}_{j,j+1} \,, & \widetilde{Z}_{j,j+1} &\to \widetilde{Z}_{j,j+1} \,, \\
X_j &\to X_j \,, & \widetilde{X}_{j,j+1} &\to X_j \widetilde{X}_{j,j+1} X_{j+1} \,.
\end{aligned} \tag{46}$$

In this new basis, the Gauss law (42) becomes $Z_j = 1$, which polarizes the site qubits to all spin up, decoupling them from the system. Furthermore, the gauge-invariant translation operator $T\widetilde{T} \prod_j X_j$ becomes $T\widetilde{T}$, which acts as just $\widetilde{T}$ in the physical, $Z_j = 1$ subspace. In the physical subspace of this new basis, the gauged XX model (44) becomes

$$\sum_{j=1}^{L} (\widetilde{X}_{j,j+1} + \widetilde{Z}_{j-1,j} \widetilde{X}_{j,j+1} \widetilde{Z}_{j+1,j+2}) \,. \tag{47}$$

We relabel these Pauli operators by dropping the tildes and performing a half lattice translation $\langle j, j+1 \rangle \to j+1$ to find the final form of the gauged Hamiltonian:[13]

$$H_{\mathrm{XX}/\mathbb{Z}_2^{\mathrm{M}}} = \sum_{j=1}^{L} (X_j + Z_{j-1} X_j Z_{j+1}) \,. \tag{48}$$

Up to an overall minus sign, which can be changed using $\prod_{j=1}^{L} Z_j$, this Hamiltonian is the transition point between the two $\mathbb{Z}_2 \times \mathbb{Z}_2$ SPT phases.[14]

---

[13]The $\mathbb{Z}_2^{\mathrm{M}}$ gauged Hamiltonian $H_{\mathrm{XX}/\mathbb{Z}_2^{\mathrm{M}}}$ is quite similar to the PXP model Hamiltonian $H_{\mathrm{PXP}} = \sum_{j=1}^{L} P_{j-1} X_j P_{j+1}$ where $P_j = (1 - Z_j)/2$ [57]. In fact, it can be written as $H_{\mathrm{XX}/\mathbb{Z}_2^{\mathrm{M}}} = 2(H_{\mathrm{PXP}} + \widetilde{H}_{\mathrm{PXP}})$ where $\widetilde{H}_{\mathrm{PXP}} = \sum_{j=1}^{L} \widetilde{P}_{j-1} X_j \widetilde{P}_{j+1}$ with $\widetilde{P}_j = (1 + Z_j)/2$. We thank Igor Klebanov for pointing this out.

[14]More correctly, it is the transition between the two $\mathbb{Z}_2$ dipole SPT states [58, 59].

Since the IR limit of the XX model is the compact free boson at $R = \sqrt{2}$, the IR limit of (48) is the $\mathbb{Z}_2^{\mathcal{M}}$-gauged compact free boson at $R = \sqrt{2}$, which is the same as the compact free boson at $R = 1/\sqrt{2}$ (see Section 2.3).[15]

The above gauging procedure implements the gauging map on $\eta = +1$ operators described by

$$\begin{pmatrix} Z_j \\ X_j X_{j+1} \end{pmatrix} \xrightarrow{\text{Gauge } \mathbb{Z}_2^{\text{M}}} \begin{pmatrix} (-1)^j Z_j Z_{j+1} \\ X_{j+1} \end{pmatrix}. \tag{50}$$

We emphasize that the gauging map is not implemented by a unitary operator.[16] Indeed, a gauged Hamiltonian is generally not related to the original Hamiltonian by a unitary transformation. However, as we will see in the next subsection, the XX Hamiltonian $H_{\text{XX}}$ is special in that it is unitarily equivalent to its gauged counterpart $H_{\text{XX}/\mathbb{Z}_2^{\text{M}}}$.

The image of the momentum charge (37) under it is

$$Q_{/\mathbb{Z}_2^{\text{M}}}^{\text{M}} = \frac{1}{2} \sum_{j=1}^{L} (-1)^j Z_j Z_{j+1} \,. \tag{51}$$

Because $Q^{\text{M}}$ commutes with the Hamiltonian before gauging, $Q_{/\mathbb{Z}_2^{\text{M}}}^{\text{M}}$ will commute with the gauged Hamiltonian and generate a U(1) symmetry. The conserved charge $Q_{/\mathbb{Z}_2^{\text{M}}}^{\text{M}}$, however, has quantized $2\mathbb{Z}$ eigenvalues for all even $L$.[17] The factor of 2 can be understood from the fact that gauging the $\mathbb{Z}_2^{\text{M}}$ symmetry compactifies the U(1)$^{\text{M}}$ angle from $[0, 2\pi)$ to $[0, \pi)$. Therefore, the gauged XX model (48) has a U(1) symmetry with $2\pi$ periodic angle generated by

$$\widehat{Q}^{\text{M}} = \frac{1}{2} Q_{/\mathbb{Z}_2^{\text{M}}}^{\text{M}} \equiv \frac{1}{4} \sum_{j=1}^{L} (-1)^j Z_j Z_{j+1} \,. \tag{52}$$

Its symmetry operator $\mathrm{e}^{\mathrm{i}\,\theta\,\widehat{Q}^{\text{M}}}$ transforms the Pauli operators as

$$\begin{aligned} \mathrm{e}^{\mathrm{i}\,\theta\,\widehat{Q}^{\text{M}}} X_j \mathrm{e}^{-\mathrm{i}\,\theta\,\widehat{Q}^{\text{M}}} &= X_j \,\mathrm{e}^{\frac{\mathrm{i}\theta}{2}(-1)^j(Z_{j-1}Z_j - Z_j Z_{j+1})} \,, \\ \mathrm{e}^{\mathrm{i}\,\theta\,\widehat{Q}^{\text{M}}} Z_j \mathrm{e}^{-\mathrm{i}\,\theta\,\widehat{Q}^{\text{M}}} &= Z_j \,. \end{aligned} \tag{53}$$

---

[15]When $L = 0 \bmod 4$, the $\mathbb{Z}_2^{\text{M}}$-gauged XX model is unitarily equivalent to the the $1+1$D Levin-Gu edge Hamiltonian

$$H_{\text{LG}} = \sum_{j=1}^{L} (X_j - Z_{j-1} X_j Z_{j+1}) \,. \tag{49}$$

A unitary transformation relating the two Hamiltonians is $(X_j, Z_j) \to (X_j, -Z_j)$ if $j = 2, 3 \bmod 4$, and under this transformation, $\widehat{Q}^{\text{M}}$ is mapped to the conserved charge $\frac{1}{4} \sum_{j=1}^{L} Z_j Z_{j+1}$ of [60]. When $L = 2 \bmod 4$, $H_{\text{XX}/\mathbb{Z}_2^{\text{M}}}$ and $H_{\text{LG}}$ are not unitarily equivalent. In fact, while the former flows to the compact free boson at $R = 1/\sqrt{2}$ with no defects present, the latter flows to the same CFT but with a defect inserted. Indeed, because a lattice translation defect of $H_{\text{LG}}$ flows to a $\mathbb{Z}_4^{\mathcal{W}}$ symmetry defect of the $R = 1/\sqrt{2}$ compact free boson [22], $H_{\text{LG}}$ at $L = 2 \bmod 4$ flows to the $R = 1/\sqrt{2}$ compact free boson with a $\mathbb{Z}_2^{\mathcal{W}}$ symmetry defect inserted. This is consistent with $\frac{1}{4} \sum_{j=1}^{L} Z_j Z_{j+1}$ being integer-quantized for $L = 0 \bmod 4$, but quantized to an integer plus half for $L = 2 \bmod 4$. This Lieb-Schultz-Mattis anomaly between translations and U(1) in (49) matches the 't Hooft anomaly between winding and momentum symmetry in the IR.

[16]Suppose it were, then there exists a unitary operator $V$ such that $V^{-1} X_{j+1} V = X_j X_{j+1}$. However, then $V^{-1} \prod_{j=1}^{L} X_{j+1} V = \prod_{j=1}^{L}(X_j X_{j+1}) = 1$, which is a contradiction.

[17]The quantization condition on $Q_{/\mathbb{Z}_2^{\text{M}}}^{\text{M}}$ can be derived using that the product of $(-1)^j Z_j Z_{j+1}$ in (51) equals $(-1)^{L(L+1)/2} = (-1)^{L/2}$. Therefore, for any $Q_{/\mathbb{Z}_2^{\text{M}}}^{\text{M}}$ eigenstate, there must always be an even (odd) number of the operators $(-1)^j Z_j Z_{j+1}$ whose eigenvalues are $-1$ when $L = 0 \bmod 4$ ($L = 2 \bmod 4$), and $Q_{/\mathbb{Z}_2^{\text{M}}}^{\text{M}}$ has eigenvalues $\frac{1}{2}\{-L, -L+4, \ldots, L-4, L\} \subset 2\mathbb{Z}$ ($\frac{1}{2}\{-L+2, -L+6, \ldots, L-6, L-2\} \subset 2\mathbb{Z}$).

This leaves the Hamiltonian (48), invariant since

$$\mathrm{e}^{\mathrm{i}\,\theta\,\widehat{Q}^{\mathrm{M}}} H_{\mathrm{XX}/\mathbb{Z}_2^{\mathrm{M}}}\,\mathrm{e}^{-\mathrm{i}\,\theta\,\widehat{Q}^{\mathrm{M}}} = \sum_{j=1}^{L} X_j\,\mathrm{e}^{-\frac{\mathrm{i}\theta}{2}(-1)^j Z_j(Z_{j+1}-Z_{j-1})}(1+Z_{j-1}Z_{j+1}), \qquad (54)$$

and $(1+Z_{j-1}Z_{j+1})$ in the local Hamiltonian projects $\mathrm{e}^{-\frac{\mathrm{i}\theta}{2}(-1)^j Z_j(Z_{j+1}-Z_{j-1})}$ to 1 (*i.e.*, each term in the sum vanishes whenever $Z_{j-1} \neq Z_{j+1}$). Furthermore, since $\mathcal{Q}^{\mathcal{M}}$ in the compact free boson at $R=\sqrt{2}$ becomes $2\mathcal{Q}^{\mathcal{M}}$ in the $R=1/\sqrt{2}$ compact free boson after gauging $\mathbb{Z}_2^{\mathcal{M}}$, the U(1) symmetry generated by $\widehat{Q}^{\mathrm{M}}$ flows to the momentum U(1) symmetry of the $R=1/\sqrt{2}$ compact free boson.

## 3.2 Lattice U(1) winding symmetry from T-duality

### 3.2.1 Lattice T-duality

The compact free boson at $R=\sqrt{2}$ and $R=1/\sqrt{2}$ are equivalent by T-duality. It is therefore natural to wonder if the XX model (36) and its $\mathbb{Z}_2^{\mathrm{M}}$-gauged version (48) are equivalent — if there is a lattice T-duality. It turns out that they are *unitarily equivalent*. The equivalence between the XX model and its $\mathbb{Z}_2^{\mathrm{M}}$-gauged version was also noted in Ref. 61.

Consider the unitary operator

$$U_{\mathrm{T}} = \prod_{n=1}^{L/2}\left(\mathrm{e}^{\mathrm{i}\frac{\pi}{4}Z_{2n+1}}\,\mathrm{e}^{\mathrm{i}\frac{\pi}{4}X_{2n+1}}\,\mathrm{e}^{-\mathrm{i}\frac{\pi}{4}X_{2n}}\,\mathsf{CZ}_{2n,2n+1}\right), \qquad (55)$$

where $\mathsf{CZ}_{j,k} = \frac{1}{2}(1+Z_j)+\frac{1}{2}(1-Z_j)Z_k$ is the controlled-Z gate. The unitary $U_{\mathrm{T}}$ acts on local operators as[18]

$$U_{\mathrm{T}}X_j U_{\mathrm{T}}^{-1} = \begin{cases} Y_{j-1}Y_j, & j \text{ odd}, \\ X_j X_{j+1}, & j \text{ even}, \end{cases} \qquad U_{\mathrm{T}}Y_j U_{\mathrm{T}}^{-1} = \begin{cases} Y_{j-1}Z_j, & j \text{ odd}, \\ Z_j X_{j+1}, & j \text{ even}. \end{cases} \qquad (57)$$

Using these transformations, it is straightforward to confirm that[19]

$$H_{\mathrm{XX}} = U_{\mathrm{T}} H_{\mathrm{XX}/\mathbb{Z}_2^{\mathrm{M}}} U_{\mathrm{T}}^{-1}\,. \qquad (58)$$

The unitary operator $U_{\mathrm{T}}$ relates the XX model Hamiltonian to its $\mathbb{Z}_2^{\mathrm{M}}$-gauged version, just as T-duality does in the compact free boson at radius $R=\sqrt{2}$. However, a defining feature of T-duality in the continuum is that it also relates the $\mathbb{Z}_2^{\mathcal{M}}$-gauged momentum (winding) charge at $R=\sqrt{2}$ to the winding (momentum) charge at $R=\sqrt{2}$. Without replicating this feature, we cannot say that $U_{\mathrm{T}}$ implements T-duality. In what follows, however, we will show that the XX model has a U(1) winding symmetry and $U_{\mathrm{T}}$ satisfies this requirement of a lattice T-duality.

---

[18]The action of $U_{\mathrm{T}}^{-1}$ on the Pauli operators is straightforward to derive using (57). Doing so, we find that

$$U_{\mathrm{T}}^{-1}X_j U_{\mathrm{T}} = \begin{cases} Z_j, & j \text{ odd}, \\ X_j Z_{j+1}, & j \text{ even}, \end{cases} \qquad U_{\mathrm{T}}^{-1}Y_j U_{\mathrm{T}} = \begin{cases} -Z_{j-1}X_j, & j \text{ odd}, \\ -Z_j, & j \text{ even}. \end{cases} \qquad (56)$$

[19]We note that Ref. 62 found a unitary transformation that maps $H_{\mathrm{XX}/\mathbb{Z}_2^{\mathrm{M}}}$ to $\sum_{j=1}^{L}(X_j X_{j+1}+Z_j Z_{j+1})$. In terms of $U_{\mathrm{T}}$, their unitary transformation is implemented by $\left(\prod_{j=1}^{L}\mathrm{e}^{-\mathrm{i}\frac{\pi}{4}Z_j}\,\mathrm{e}^{\mathrm{i}\frac{\pi}{4}Y_j}\right)U_{\mathrm{T}}\left(\prod_{j=1}^{L}\mathsf{CZ}_{j,j+1}\right)$. Closely related unitaries were also constructed in Refs. 51 and 63.

### 3.2.2 A quantized winding charge

T-duality exchanges the momentum and the winding symmetry in the compact free boson. Therefore, since $\widehat{Q}^M$ generates a U(1) lattice momentum symmetry of $H_{XX/\mathbb{Z}_2^M}$, we can use $U_T$ to identify a U(1)$^W$ symmetry of the XX model that becomes the winding symmetry in the IR. In doing so, we find the conserved charge

$$Q^W = U_T \widehat{Q}^M U_T^{-1} = \frac{1}{4} \sum_{n=1}^{L/2} (X_{2n-1} Y_{2n} - Y_{2n} X_{2n+1}), \tag{59}$$

where the superscript W emphasizes that it generates a U(1)$^W$ lattice winding symmetry of the XX model (*i.e.*, it flows to the U(1) winding symmetry of the compact free boson at $R = \sqrt{2}$). The lattice charge conjugation symmetry (39) acts on $Q^W$ as $C Q^W C^{-1} = -Q^W$, matching its corresponding action in the IR.[20] Furthermore, because of (52), this implies that

$$Q^W = U_T \left( \frac{1}{2} Q^M_{/\mathbb{Z}_2^M} \right) U_T^{-1}. \tag{60}$$

The U(1)$^W$ symmetry transforms the Pauli operators as

$$e^{i\theta Q^W} X_j e^{-i\theta Q^W} = \begin{cases} X_j, & j \text{ odd}, \\ X_j e^{-\frac{i\theta}{2} Y_j (X_{j-1} - X_{j+1})}, & j \text{ even}, \end{cases} \tag{61}$$

$$e^{i\theta Q^W} Y_j e^{-i\theta Q^W} = \begin{cases} Y_j e^{\frac{i\theta}{2} X_j (Y_{j-1} - Y_{j+1})}, & j \text{ odd}, \\ Y_j, & j \text{ even}. \end{cases} \tag{62}$$

It commutes with $H_{XX}$ since it leaves $X_{2n-1} X_{2n} + X_{2n} X_{2n+1}$ and $Y_{2n} Y_{2n+1} + Y_{2n+1} Y_{2n+2}$ invariant. A local operator carrying charge $Q^W = +1$ charge is

$$\begin{cases} X_j \left( \frac{1+X_{j-1}}{2} \frac{1-Y_j}{2} \frac{1-X_{j+1}}{2} \right), & j \text{ even}, \\ Y_j \left( \frac{1+Y_{j-1}}{2} \frac{1+X_j}{2} \frac{1-Y_{j+1}}{2} \right), & j \text{ odd}. \end{cases} \tag{63}$$

Since the lattice winding symmetry acts differently on qubits residing on even and odds sites, it is a modulated U(1) symmetry [64–68]. Therefore, the translation operator $T$ does not commute with the charge $Q^W$, instead satisfying

$$T Q^W T^{-1} = -\frac{1}{4} \sum_{n=1}^{L/2} (Y_{2n-1} X_{2n} - X_{2n} Y_{2n+1}) \equiv e^{i\frac{\pi}{2} Q^M} Q^W e^{-i\frac{\pi}{2} Q^M}. \tag{64}$$

A unitarily related version of the quantized lattice winding charge $Q^W$ first appeared in [69] and $Q^W$ was later discussed in [36–38] as the generator of a U(1) symmetry of the XX model. Here, we identify the U(1)$^W$ symmetry it generates with the winding symmetry in the IR compact free boson theory. Interestingly, while the momentum and winding symmetries commute in the IR, the lattice charges $Q^W$ and $Q^M$ do not commute, *i.e.*, $[Q^M, Q^W] \neq 0$. Instead, they generate an extensively large Lie algebra known as the Onsager algebra [36,37,39]. However, in the IR limit, the lattice charges do commute and flow to the anomalous U(1)$^W \times$ U(1)$^M$ symmetry of the compact free boson. We will further discuss the Onsager algebra and its relation to lattice T-duality in Section 5.

---

[20]Gauging the $\mathbb{Z}_2^C$ symmetry of the XX model leads to the model $H_{\text{Ising}^2} = \sum_{j=1}^L (Z_j Z_{j+2} - X_j)$, which flows to the Ising$^2$ CFT when $L = 0$ mod 4. The XX model's U(1) symmetries generated by $Q^M$ and $Q^W$ become non-invertible symmetries under this gauging map. These non-invertible symmetries are the lattice counterpart to the non-invertible "cosine" symmetries of the Ising$^2$ CFT [31].

Even though $[Q^M, Q^W] \neq 0$, the symmetry operators $e^{i\phi Q^M}$ and $e^{i\theta Q^W}$ can still commute for particular values of $\phi$ and $\theta$. For example, $e^{i\pi Q^M}$ and $e^{i\theta Q^W}$ commute for all values of $\theta$. However, $e^{i\phi Q^M}$ and $e^{i\pi Q^W}$ only commute for $\phi = \pi$. Therefore, unlike in the compact free boson, the momentum and winding symmetries in the XX model are on different footing.

We can also identify a lattice winding symmetry of the $\mathbb{Z}_2^M$-gauged XX model (48) using the unitary operator $U_T$:

$$\widehat{Q}^W = U_T^{-1} Q^M U_T = \frac{1}{2} \sum_{n=1}^{L/2} (Y_{2n} Z_{2n+1} - Z_{2n} Y_{2n+1}). \tag{65}$$

This flows to the winding charge of the compact free boson at $R = 1/\sqrt{2}$. In fact, denoting by $Q^W_{/\mathbb{Z}_2^M}$ the image of $Q^W$ under the $\mathbb{Z}_2^M$-gauging map (50), it satisfies

$$Q^W_{/\mathbb{Z}_2^M} = \frac{1}{2} \widehat{Q}^W \implies Q^M = U_T \left( 2 Q^W_{/\mathbb{Z}_2^M} \right) U_T^{-1}. \tag{66}$$

The factor of $1/2$ matches the corresponding result in the compact free boson (see (34)). Furthermore, the charge conjugation symmetry $C = \prod_{j=1}^{L} X_j$ of the XX model in this basis becomes

$$\widehat{C} = U_T^{-1} C U_T = \prod_{j=1}^{L} [X_j]^{j+1}. \tag{67}$$

This is a modulated $\mathbb{Z}_2$ symmetry that satisfies $T\widehat{C}T^{-1} = e^{i\pi\widehat{Q}^W}\widehat{C}$.[21] Therefore, the $\widehat{C}$ and $e^{i\pi\widehat{Q}^W}$ symmetry operators of $H_{XX/\mathbb{Z}_2^M}$ form a $\mathbb{Z}_2$ dipole symmetry. This dipole symmetry arises after $\mathbb{Z}_2^M$ gauging due to the Lieb-Schultz-Mattis (LSM) anomaly between lattice translations, $C$ and $e^{i\pi Q^M} \equiv \eta$ in the XX model (see Refs. 70–72 for more discussion on the relationship between dipole symmetries and LSM anomalies via gauging).

Therefore, the unitary transformation implemented by $U_T$ is lattice version of the T-duality map in the XX model since it relates the XX model Hamiltonian *and* its lattice momentum and winding charges to their $\mathbb{Z}_2^M$-gauged versions (*i.e.*, Eqs. (60) and (66)), just as T-duality does in the compact free boson. We summarize this relation in Fig. 1. As we will discuss in Section 3.3, this lattice T-duality is equivalent to the existence of a non-invertible symmetry of the XX model.

The unitary $U_T$ satisfies $U_T^5 = 1$. Since T-duality in the compact free boson maps the radius $R$ CFT to radius $1/R$, it is sometimes said that T-duality has a finite order. However, dualities of quantum field theories do not have an order: they are maps between classical Lagrangians that yield the same quantum field theory. More precisely, what is meant is that at the self-dual point $R = 1$, the T-duality map becomes a $\mathbb{Z}_4$ symmetry of the quantum field theory [73]. It is the self-duality symmetry that has a finite order.

### 3.2.3 An unquantized winding charge

The conserved winding charge $Q^W$ is integer quantized and its relation to $Q^M$ gives rise to lattice T-duality in the XX model. It, however, does not commute with $Q^M$. Here we note that there is another conserved charge

$$\widetilde{Q}^W = \frac{1}{2}(Q^W + TQ^W T^{-1}) = \frac{1}{8} \sum_{j=1}^{L} (X_j Y_{j+1} - Y_j X_{j+1}), \tag{68}$$

---

[21]Because lattice translation $T$ of the XX model flows to $e^{i\pi(\mathcal{Q}^M + \mathcal{Q}^W)}$ in $R = \sqrt{2}$ compact free boson [22, 42], $T$ in the $\mathbb{Z}_2^M$-gauged XX model flows to $e^{i\pi(2\mathcal{Q}^M + \frac{1}{2}\mathcal{Q}^W)} = e^{i\frac{\pi}{2}\mathcal{Q}^W}$ in the $R = 1/\sqrt{2}$ compact free boson. Denoting by $\mathcal{C}$ the IR limit of $\widehat{C}$, the dipole symmetry algebra $T\widehat{C}T^{-1} = e^{i\pi\widehat{Q}^W}\widehat{C}$ on the lattice becomes $e^{i\frac{\pi}{2}\mathcal{Q}^W}\mathcal{C}e^{-i\frac{\pi}{2}\mathcal{Q}^W} = e^{i\pi\mathcal{Q}^W}\mathcal{C}$ in the IR, describing the action of charge conjugation on the winding symmetry.

which does commute with $Q^{\mathrm{M}}$. This conserved charge was considered in Ref. 74. Since the diagonal part of $\mathbb{Z}_2^{\mathcal{M}} \times \mathbb{Z}_2^{\mathcal{W}}$ emanates from lattice translations, $\widetilde{Q}^{\mathrm{W}}$ also flows to the winding symmetry charge in the IR. Furthermore, $\widetilde{Q}^{\mathrm{W}}$ can be written as

$$\widetilde{Q}^{\mathrm{W}} = -\frac{\mathrm{i}}{2\pi} \sum_{j=1}^{L} g_j^{-1} \left( \frac{g_{j+1} - g_{j-1}}{2} \right), \qquad g_j = \frac{\sqrt{\pi}}{2}(X_j + \mathrm{i}Y_j), \tag{69}$$

which, with the identification of $g_j$ with $\mathrm{e}^{\mathrm{i}\Phi}$, makes $\widetilde{Q}^{\mathrm{W}}$ the most straightforward lattice regularization of the winding charge $\mathcal{Q}^{\mathcal{W}} = -\frac{\mathrm{i}}{2\pi} \int \mathrm{e}^{-\mathrm{i}\Phi} \partial_x \mathrm{e}^{\mathrm{i}\Phi} \, \mathrm{d}x$ in the compact free boson.

While $\widetilde{Q}^{\mathrm{W}}$ is appealing as a naïve lattice regularization, it does not relate to the momentum charge $Q^{\mathrm{M}}$ under the lattice T-duality $U_{\mathrm{T}}$. It becomes further less appealing when viewed as generating a lattice symmetry. Indeed, it is tempting to view the unitary operator $\mathrm{e}^{\mathrm{i}\lambda\widetilde{Q}^{\mathrm{W}}}$ as an $\mathbb{R}$ symmetry operator ($\widetilde{Q}^{\mathrm{W}}$ does not have integer-quantized eigenvalues). For $\lambda \sim \mathcal{O}(L^0)$ and large $L$, $\mathrm{e}^{\mathrm{i}\lambda\widetilde{Q}^{\mathrm{W}}}$ transforms a local operator acting in a neighborhood of site $j$ to a quasi-local operator localized around site $j$. For $\lambda \sim \mathcal{O}(L)$, however, it transforms such a local operator to a non-local operator, i.e. one that acts on all sites in a non-localized fashion [39]. Therefore, $\mathrm{e}^{\mathrm{i}\lambda\widetilde{Q}^{\mathrm{W}}}$ is not a locality-preserving unitary. This lack of locality disqualifies $\mathrm{e}^{\mathrm{i}\lambda\widetilde{Q}^{\mathrm{W}}}$ from being interpreted as a symmetry operator on the lattice. In the IR limit, however, $\widetilde{Q}^{\mathrm{W}}$ becomes integer-quantized and $\mathrm{e}^{\mathrm{i}\lambda\widetilde{Q}^{\mathrm{W}}}$ becomes a well-behaved symmetry operator, namely the winding symmetry operator.

## 3.3 Non-invertible symmetry and lattice T-duality

T-duality in the compact free boson implies a non-invertible symmetry when $R^2 \in \mathbb{Z}_{>0}$ (see Section 2.3). Because the XX model flows to the compact free boson at $R = \sqrt{2}$ and has a lattice version of T-duality, there is a corresponding lattice non-invertible symmetry. This symmetry implements the $\mathbb{Z}_2^{\mathrm{M}}$-gauging map (50) followed by the lattice T-duality map (57), and its action on $\mathbb{Z}_2^{\mathrm{M}}$-symmetric operators follows from

$$\begin{pmatrix} Z_{2n-1} \\ Z_{2n} \\ X_{2n-1}X_{2n} \\ X_{2n}X_{2n+1} \end{pmatrix} \xrightarrow{\text{Gauge } \mathbb{Z}_2^{\mathrm{M}}} \begin{pmatrix} -Z_{2n-1}Z_{2n} \\ Z_{2n}Z_{2n+1} \\ X_{2n} \\ X_{2n+1} \end{pmatrix} \xrightarrow{U_{\mathrm{T}}} \begin{pmatrix} X_{2n-1}Y_{2n} \\ -Y_{2n}X_{2n+1} \\ X_{2n}X_{2n+1} \\ Y_{2n}Y_{2n+1} \end{pmatrix}. \tag{70}$$

These transformations are implemented by an operator $\mathsf{D}$ that satisfies

$$\mathsf{D}Z_j = \begin{cases} (X_jY_{j+1})\mathsf{D}, & j \text{ odd}, \\ (-Y_jX_{j+1})\mathsf{D}, & j \text{ even}, \end{cases} \qquad \mathsf{D}X_jX_{j+1} = \begin{cases} (X_{j+1}X_{j+2})\mathsf{D}, & j \text{ odd}, \\ (Y_jY_{j+1})\mathsf{D}, & j \text{ even}. \end{cases} \tag{71}$$

This is a non-invertible operator because $\mathsf{D}\eta = \mathsf{D}$, so it has a nontrivial kernel spanned by states $|\psi\rangle$ for which $\eta|\psi\rangle = -|\psi\rangle$. Furthermore, using (71), we find

$$\mathsf{D}Y_jY_{j+1} = \begin{cases} (X_jX_{j+1})\mathsf{D}, & j \text{ odd}, \\ (Y_{j+1}Y_{j+2})\mathsf{D}, & j \text{ even}, \end{cases} \tag{72}$$

which makes it clear that $\mathsf{D}$ commutes with the XX model Hamiltonian, as expected.

The non-invertible symmetry operator $\mathsf{D}$ satisfies

$$\mathsf{D}Q^{\mathrm{M}} = 2Q^{\mathrm{W}}\mathsf{D}, \qquad \mathsf{D}Q^{\mathrm{W}} = \frac{1}{2}Q^{\mathrm{M}}\mathsf{D}. \tag{73}$$

The action of $\mathsf{D}$ on these symmetry charges is the same as the Kramers-Wannier symmetries' action (34) on the momentum and winding symmetry currents in the compact free boson!

Therefore, any Hamiltonian that commutes with $Q^M$, $Q^W$, and D has the lattice T-duality implemented by $U_T$. Indeed, by the construction of D, any Hamiltonian commuting with D will be unitarily equivalent by $U_T$ to its $\mathbb{Z}_2^M$ gauged version. In fact, as we argue in Section 6.2, any many-qubit, local Hamiltonian that commutes with $Q^M$ and $Q^W$ must also commute with D.

Using the action of D on the $\eta$-symmetric operators and that $L$ is even, we find that D satisfies the operator algebra[22]

$$
\begin{aligned}
D^2 &= (1+\eta)\, T\, e^{-i\frac{\pi}{2}Q^M}\,, \qquad D\,\eta = \eta\, D = D\,, \qquad C D = D C\,, \\
T D T^{-1} &= e^{i\frac{\pi}{2}Q^M} e^{i\pi Q^W} D\,, \qquad D^\dagger = D\, T^{-1}\, e^{i\frac{\pi}{2}Q^M}\,.
\end{aligned}
\tag{74}
$$

This operator algebra, which mixes with lattice translation, bears many similarities to other Kramers-Wannier type symmetries [75–81] (see [13,71,82–99] for discussion related to other non-invertible symmetry operators in quantum spin chains and [100–107] for those in higher-dimensional quantum spin models). However, something notable about this non-invertible symmetry operator is that it does not commute with lattice translations. Therefore, it is a modulated non-invertible symmetry [71]. This is also evident from its action on $\eta$-symmetry operators (71) depending on the position of the qubits. Nevertheless, while $T$ does not commute with D, lattice translations by two sites $T^2$ does commute.

### 3.3.1 Continuous families of non-invertible symmetries

Using the invertible symmetry operators discussed earlier in this section, we can construct two families of non-invertible symmetries

$$
D_{+,\phi,\theta} = e^{i\phi Q^M}\, e^{i\theta Q^W}\, D\,, \qquad D_{-,\phi,\theta} = C\, e^{i\phi Q^M}\, e^{i\theta Q^W}\, D\,.
\tag{75}
$$

These operators act on the momentum and winding charges by

$$
\begin{aligned}
D_{\pm,\phi,\theta}Q^M &= (\pm 2 e^{\pm i\phi Q^M} Q^W e^{\mp i\phi Q^M}) D_{\pm,\phi,\theta}\,, \\
D_{\pm,\phi,\theta}Q^W &= \big(\pm \frac{1}{2} e^{\pm i\phi Q^M} e^{\pm i\theta Q^W} Q^M e^{\mp i\theta Q^W} e^{\mp i\phi Q^M}\big) D_{\pm,\phi,\theta}\,.
\end{aligned}
\tag{76}
$$

The only operator $D_{\pm,\phi,\theta}$ whose action on $Q^M$ and $Q^W$ exactly matches with that in the compact free boson is $D_{+,0,0} = D$. Furthermore, using the operator algebra for D, it is straightforward to find the operator algebra obeyed by $D_{\pm,\phi,\theta}$. For example, $D_{\pm,\phi,\theta}$ satisfies

$$
(D_{\pm,\phi,\theta})^2 = (1+\eta)\, e^{\pm i\phi Q^M}\, e^{i(2\phi\pm\theta)Q^W}\, e^{\frac{i}{2}(\theta-\pi)Q^M}\, T\,,
\tag{77}
$$

$$
T D_{\pm,\phi,\theta} T^{-1} = D_{\pm,\,\phi+\pi/2,\,\theta+\pi}\,.
\tag{78}
$$

Therefore, $D_{\pm,\phi,\theta}$ is always a modulated non-invertible operator that squares to something involving more than just $(1+\eta)T$.

Similar to other non-invertible Kramers-Wannier symmetry operators in $1+1$D quantum spin chains, the non-invertible symmetries in the CFT describing the IR arise from $D_{\pm,\phi,\theta}$. Let us denote by $\mathcal{D}_{\pm,\phi,\theta}$ the IR limit of $D_{\pm,\phi,\theta}$. Then, since $Q^M$ and $Q^W$ flow to the momentum and winding charges $\mathcal{Q}^\mathcal{M}$ and $\mathcal{Q}^\mathcal{M}$ of the compact free boson, respectively, and since $T$ flows to $e^{i\pi(\mathcal{Q}^\mathcal{M}+\mathcal{Q}^\mathcal{W})}$ [22,42], the IR operator $\mathcal{D}_{\pm,\phi,\theta}$ satisfies

$$
(\mathcal{D}_{\pm,\phi,\theta})^2 = (1 + e^{i\pi\mathcal{Q}^\mathcal{M}}) e^{\frac{i}{2}(\theta\pm 2\phi-\pi)(\mathcal{Q}^\mathcal{W}\pm 2\mathcal{Q}^\mathcal{W})}\,.
\tag{79}
$$

---

[22]It is easy to see that Eqs. (73) and (74) are consistent with one another because $Q^W$ is a modulated operator that satisfies (64). Similarly, it follows from Eq. (64) that $T^2 D = D\, T^2$.

When $\theta = \pi \mp 2\phi$, this non-invertible symmetry operator obeys the $\mathbb{Z}_2$ Tambara-Yamagami fusion algebra

$$(\mathcal{D}_{\pm,\phi,\pi\mp2\phi})^2 = 1 + e^{i\pi\mathcal{Q}^{\mathcal{M}}} \,. \tag{80}$$

Therefore, the non-invertible symmetry operators $\mathsf{D}_{\pm,\phi,\pi\mp2\phi}$ of the XX model flow to the $\mathbb{Z}_2$ Tambara-Yamagami fusion category symmetries of the compact free boson at $R = \sqrt{2}$ [31]. In particular, these symmetries are described by the $\mathbb{Z}_2$ Tambara-Yamagami fusion category whose Frobenius-Schur indicator is $\epsilon = 1$. Furthermore, while not all $\mathcal{D}_{\pm,\phi,\theta}$ are described by this fusion category symmetry, they all act on the IR momentum and winding charges as

$$\mathcal{D}_{\pm,\phi,\theta}\mathcal{Q}^{\mathcal{M}} = \pm2\mathcal{Q}^{\mathcal{W}}\mathcal{D}_{\pm,\phi,\theta} \,, \qquad \mathcal{D}_{\pm,\phi,\theta}\mathcal{Q}^{\mathcal{W}} = \pm\frac{1}{2}\mathcal{Q}^{\mathcal{M}}\mathcal{D}_{\pm,\phi,\theta} \,. \tag{81}$$

This follows from the expression (76) and the fact that the IR momentum and winding charges commute.

### 3.3.2 Matrix product operator expression

Thus far, our discussion of the non-invertible symmetry D related to lattice T-duality in the XX model did not use an explicit expression for D. Here, we will construct a matrix product operator (MPO) expression for D. This is most easily done by first rotating to a basis in which D is simpler, constructing the MPO expression, and then rotating back.

Consider the unitary operator

$$U = \prod_{n=1}^{L/2} e^{i\frac{\pi}{4}(3Y_{2n-1}+Y_{2n})}e^{-i\frac{\pi}{4}Z_{2n}} \,, \tag{82}$$

which implements the transformation

$$U X_j U^{-1} = \begin{cases} -Z_j \,, & j \text{ odd,} \\ +Y_j \,, & j \text{ even,} \end{cases} \qquad U Y_j U^{-1} = \begin{cases} +Y_j \,, & j \text{ odd,} \\ -Z_j \,, & j \text{ even.} \end{cases} \tag{83}$$

The XX model under this transformation becomes

$$UH_{XX}U^{-1} = -\sum_{j=1}^{L}(Z_jY_{j+1} + Y_jZ_{j+1}) \,. \tag{84}$$

This Hamiltonian commutes with the operator $\mathsf{D}_U$ that satisfies

$$\mathsf{D}_U X_j = Z_j Z_{j+1}\mathsf{D}_U \,, \qquad \mathsf{D}_U Z_j Z_{j+1} = -X_{j+1}\mathsf{D}_U \,. \tag{85}$$

Since we assume $L$ is even, states charged under the symmetry $\prod_{j=1}^{L} X_j$ of (84) span the kernel of $\mathsf{D}_U$, so $\mathsf{D}_U$ a non-invertible symmetry operator of (84). Using its action on $\eta$-symmetric local operators, we find that the non-invertible symmetry $\mathsf{D}_U$ once rotated back into the basis of the XX model Hamiltonian becomes

$$\mathsf{D} = U^{-1}\mathsf{D}_U U \,. \tag{86}$$

The transformation (85) implemented by $\mathsf{D}_U$ is similar to that of the non-invertible symmetry operator $\mathsf{D}_{KW}$ in the critical Ising chain. Using that $\mathsf{D}_{KW}$ satisfies $\mathsf{D}_{KW}X_j = Z_jZ_{j+1}\mathsf{D}_{KW}$ and $\mathsf{D}_{KW}Z_jZ_{j+1} = X_{j+1}\mathsf{D}_{KW}$, these two operators are related by

$$\mathsf{D}_U = e^{2\pi i\frac{L}{4}} e^{i\pi Q^{\mathsf{M}}} \mathsf{D}_{KW} \,. \tag{87}$$

The advantage of this is that the MPO expression for $D_{KW}$ is well-known [76,80,108],[23] and using it allows us to easily find the MPO expression for D using that $D = e^{2\pi i \frac{L}{4}} U^{-1} e^{i\pi Q^M} D_{KW} U$. Doing so, we find that

$$D = \text{Tr}\left(\prod_{j=1}^{L} \mathbb{D}^{(j)}\right) \equiv \boxed{\mathbb{D}^{(1)}} - \boxed{\mathbb{D}^{(2)}} - \cdots - \boxed{\mathbb{D}^{(L)}} \tag{89}$$

where the trace is over the virtual Hilbert space and the MPO tensors are

$$\mathbb{D}^{(j)} \equiv -\boxed{\mathbb{D}^{(j)}}- = \begin{cases} \dfrac{1}{\sqrt{8}}\begin{pmatrix} \mathbf{1}-Z_j+X_j+iY_j & \mathbf{1}+Z_j+X_j-iY_j \\ -\mathbf{1}-Z_j+X_j-iY_j & \mathbf{1}-Z_j-X_j-iY_j \end{pmatrix}, & j \text{ odd,} \\[3ex] \dfrac{i}{\sqrt{8}}\begin{pmatrix} \mathbf{1}+Z_j-iX_j-Y_j & -\mathbf{1}+Z_j-iX_j+Y_j \\ \mathbf{1}-Z_j-iX_j+Y_j & \mathbf{1}+Z_j+iX_j+Y_j \end{pmatrix}, & j \text{ even.} \end{cases} \tag{90}$$

D is a modulated operator because its MPO tensors differ between even and odd sites. The MPO tensors satisfy

$$T\mathbb{D}^{(j)}T^{-1} = -Y_{j+1}e^{i\frac{\pi}{4}Z_{j+1}}\mathbb{D}^{(j+1)}X_{j+1}e^{i\frac{\pi}{4}Z_{j+1}}. \tag{91}$$

The explicit MPO expression of D is useful since various identities involving D can be derived explicitly using it. Denoting by $\mathbb{X}$, $\mathbb{Y}$, and $\mathbb{Z}$ the Pauli matrix tensors, the matrix tensors $\mathbb{D}^{(j)}$ satisfy

$$\begin{aligned} Z_j\mathbb{D}^{(j)} &= (-1)^j\mathbb{X}\mathbb{D}^{(j)}\mathbb{X}, & \mathbb{D}^{(j)}Z_j &= (-1)^j\mathbb{Z}\mathbb{D}^{(j)}\mathbb{Z}, \\ X_j\mathbb{D}^{(j)} &= \begin{cases} \mathbb{Z}\mathbb{D}^{(j)}, & j \text{ odd,} \\ \mathbb{Y}\mathbb{D}^{(j)}\mathbb{X}, & j \text{ even,} \end{cases} & \mathbb{D}^{(j)}X_j &= \begin{cases} \mathbb{D}^{(j)}\mathbb{X}, & j \text{ odd,} \\ \mathbb{Z}\mathbb{D}^{(j)}\mathbb{Y}, & j \text{ even.} \end{cases} \end{aligned} \tag{92}$$

Using these expression, for instance, we find that the $\mathbb{Z}_2^M$ symmetry operator $\eta = \prod_{j=1}^{L}(-1)^j Z_j$ acts on $\mathbb{D}^{(j)}$ by

$$\eta\,\mathbb{D}^{(j)}\eta^{-1} = \mathbb{Y}\mathbb{D}^{(j)}\mathbb{Y}^{-1}. \tag{93}$$

Therefore, the $\mathbb{Z}_2^M$ symmetry operator acts as $\mathbb{Y}$ in the virtual Hilbert space. Furthermore, using (92), we find that

$$\mathbb{D}^{(2n-1)}\mathbb{D}^{(2n)}(Z_{2n-1}) = -\mathbb{Z}\mathbb{D}^{(2n-1)}\mathbb{Z}\mathbb{D}^{(2n)} = (X_{2n-1}Y_{2n})\mathbb{D}^{(2n-1)}\mathbb{D}^{(2n+2)},$$

$$\mathbb{D}^{(2n)}\mathbb{D}^{(2n+1)}(Z_{2n}) = \mathbb{Z}\mathbb{D}^{(2n)}\mathbb{Z}\mathbb{D}^{(2n+1)} = (-Y_{2n}X_{2n+1})\mathbb{D}^{(2n)}\mathbb{D}^{(2n+1)},$$

$$\mathbb{D}^{(2n-1)}\mathbb{D}^{(2n)}\mathbb{D}^{(2n+1)}(X_{2n-1}X_{2n}) = -i\mathbb{D}^{(2n-1)}\mathbb{Y}\mathbb{D}^{(2n)}\mathbb{Y}\mathbb{D}^{(2n+1)} = (X_{2n}X_{2n+1})\mathbb{D}^{(2n-1)}\mathbb{D}^{(2n)}\mathbb{D}^{(2n+1)},$$

$$\mathbb{D}^{(2n)}\mathbb{D}^{(2n+1)}(X_{2n}X_{2n+1}) = \mathbb{Z}\mathbb{D}^{(2n)}\mathbb{Y}\mathbb{D}^{(2n+1)}\mathbb{X} = (Y_{2n}Y_{2n+1})\mathbb{D}^{(2n)}\mathbb{D}^{(2n+1)},$$

from which the action (71) of D on $Z_j$ and $X_jX_{j+1}$ straightforwardly follows.

---

[23]Written in terms of the Pauli operators, the MPO expression for $D_{KW}$ is

$$D_{KW} = \text{Tr}\left(\prod_{j=1}^{L} \mathbb{D}_{KW}^{(j)}\right), \qquad \mathbb{D}_{KW}^{(j)} = \frac{1}{\sqrt{8}}\begin{pmatrix} \mathbf{1}+Z_j+X_j+iY_j & \mathbf{1}+Z_j-X_j-iY_j \\ -\mathbf{1}+Z_j+X_j-iY_j & \mathbf{1}-Z_j+X_j-iY_j \end{pmatrix}. \tag{88}$$

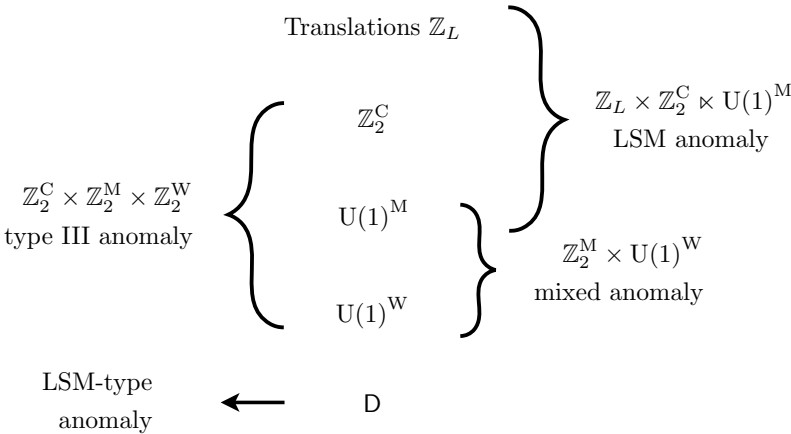

Figure 2: Symmetries of the XX model and their anomalies. When considering only lattice translation, the $\mathbb{Z}_2^{\mathrm{C}}$ symmetry generated by $C = \prod_{j=1}^{L} X_j$, and the spin rotation symmetry $\mathrm{U}(1)^{\mathrm{M}}$ generated by $Q^{\mathrm{M}} = \frac{1}{2}\sum_{j=1}^{L} Z_j$, the only anomaly is the well-known LSM anomaly [22, 40–44]. Once the winding symmetry (59) and non-invertible symmetry formed by D (71) are included, three new types of anomalies arise.

## 4 't Hooft anomalies in the XX model

In Section 3, we identified a lattice T-duality in the XX model and related it to the lattice winding and non-invertible symmetries that flow to well-known symmetries of the compact free boson at $R = \sqrt{2}$ in its IR limit. The compact free boson, however, also has various 't Hooft anomalies. In fact, as we reviewed in Section 2.1, all of its anomalous invertible symmetries involve the winding symmetry. Furthermore, the mixed anomaly between the momentum and winding symmetries enforces any symmetric theory to be gapless.

Having found a lattice winding symmetry in the XX model, it is natural to wonder how its anomalies compare to those in the compact free boson. However, it only makes sense to compare anomalies of symmetries whose symmetry operators form the same group/algebra. Since the U(1) momentum and winding symmetry operators do not commute on the lattice but do in the compact free boson, their *total* symmetry groups are different. Nonetheless, some of the IR anomalies can still be matched by those on the lattice because there are *sub-symmetries* that obey the same group/algebra on the lattice and in the continuum. In particular, the lattice symmetry operators $\eta$, $C$, and $\mathrm{e}^{\mathrm{i}\theta Q^{\mathrm{W}}}$ satisfy

$$\eta\, C = C\eta\,, \qquad \eta\, \mathrm{e}^{\mathrm{i}\theta Q^{\mathrm{W}}} = \mathrm{e}^{\mathrm{i}\theta Q^{\mathrm{W}}}\eta\,, \qquad C\, \mathrm{e}^{\mathrm{i}\theta Q^{\mathrm{W}}} = \mathrm{e}^{-\mathrm{i}\theta Q^{\mathrm{W}}}C\,, \tag{94}$$

which give the group multiplication law for $\mathbb{Z}_2^{\mathrm{M}} \times \mathbb{Z}_2^{\mathrm{C}} \ltimes \mathrm{U}(1)^{\mathrm{W}}$ (recall that we assume $L$ is even). This is the same as the $\mathbb{Z}_2^{\mathcal{M}} \times \mathbb{Z}_2^{\mathcal{C}} \ltimes \mathrm{U}(1)^{\mathcal{W}}$ symmetry in the compact free boson, so we can compare their anomalies.

In this section, we will discuss the 't Hooft anomalies in the XX model involving the lattice winding symmetry identified in Section 3.2, as well as the non-invertible symmetry from 3.3. We will emphasize their connection to corresponding anomalies of the compact free boson. In particular, we first show how the $\mathbb{Z}_2^{\mathrm{M}} \times \mathrm{U}(1)^{\mathrm{W}}$ and $\mathbb{Z}_2^{\mathrm{M}} \times \mathbb{Z}_2^{\mathrm{W}} \times \mathbb{Z}_2^{\mathrm{C}}$ symmetries of the XX model are anomalous, and how these anomalies can be diagnosed using symmetry defects (just as in the IR). The various anomalies present in the XX model are summarized in Fig. 2. Our discussion follows [22], which discusses various anomalies of the XX model (and the more general XXZ model) that do not involve the lattice winding symmetry generated by $Q^{\mathrm{W}}$. Then, we prove that any quantum spin chain with both the $\mathrm{U}(1)^{\mathrm{M}}$ and $\mathrm{U}(1)^{\mathrm{W}}$ symmetries must be gapless, reminiscent of the 't Hooft anomaly of $\mathrm{U}(1)^{\mathcal{M}} \times \mathrm{U}(1)^{\mathcal{W}}$ in the compact free boson.

## 4.1 The mixed anomaly of momentum and winding symmetries

In the compact free boson, there is a mixed anomaly between $U(1)^{\mathcal{M}}$ and $U(1)^{\mathcal{W}}$ that manifests through spectral flow (*i.e.*, charge pumping in the corresponding $2+1D$ SPT). Here, we will show how this anomaly and related spectral flow arise from the $\mathbb{Z}_2^M \times U(1)^W$ symmetry in the XX model using symmetry defects. We only consider the $\mathbb{Z}_2^M \subset U(1)^M$ sub-symmetry since other $U(1)^M$ symmetry operators do not commute with $U(1)^W$ symmetry operators.

Let us first insert an $\eta$-symmetry defect. Since $\mathbb{Z}_2^M$ is an internal symmetry whose symmetry operator $\eta$ is unitary, we can do so by first transforming all operators by $\eta_{I,J} = \prod_{j=I}^J (-1)^j Z_j$ (*i.e.*, $\eta$ truncated to sites $I \leq j \leq J$). By having $|I-J| \sim \mathcal{O}(L)$, this modifies (quasi-)local, $\eta$-symmetric operators in localized neighborhoods around sites $I$ and $J$. These two localized modifications are $\eta$-symmetry defects, which we take to reside at the links $\langle I-1, I \rangle$ and $\langle J, J+1 \rangle$, respectively. To find the corresponding operator with only one symmetry defect at $\langle I-1, I \rangle$, we then remove the modifications near site $J$ by hand.

Using this procedure, we find that the XX Hamiltonian with a $\eta$-defect is

$$H_\eta^{\langle I-1,I \rangle} = H_{XX} - 2(X_{I-1}X_I + Y_{I-1}Y_I), \tag{95}$$

which differs from the original XX Hamiltonian in the signs of the terms on the $\langle I-1, I \rangle$ link.

Because the Hamiltonian $H_\eta^{\langle I-1,I \rangle}$ differs from $H_{XX}$, the symmetry operators of the XX model will generally also be modified (that is if they remain symmetries, of course).[24] Indeed, this procedure also modifies the lattice translation operator $T$ to $T_\eta = Z_I T$.[25] This twisted lattice translation satisfies the twisted periodic boundary conditions $[T_\eta]^L = e^{2\pi i \frac{L}{4}} \eta$. Furthermore, the winding symmetry charge $Q^W$ in the presence of this defect becomes

$$Q_\eta^W = \begin{cases} Q^W + \frac{1}{2}Y_{I-1}X_I, & I \text{ odd}, \\ Q^W - \frac{1}{2}X_{I-1}Y_I, & I \text{ even}. \end{cases} \tag{96}$$

The symmetry defect can be moved from $\langle I-1, I \rangle$ to $\langle I, I+1 \rangle$ (and vice versa) by conjugating observables (*e.g.*, the defect Hamiltonian and symmetry operators) with the unitary operator $Z_I$. For instance, $Z_I H_\eta^{\langle I-1,I \rangle} Z_I^{-1} = H_\eta^{\langle I,I+1 \rangle}$. Therefore, the spectrum of $H_\eta^{\langle I-1,I \rangle}$ does not depend on $I$, and the symmetry defect is referred to as a topological defect for this reason.

While $Q^W$ has integer eigenvalues, the eigenvalues of $Q_\eta^W$ are integer plus a half. This is because $Q^W$ commutes with $Y_{I-1}X_I$ and $X_{I-1}Y_I$, both of which have integer eigenvalues. Consequentially, after inserting the $\eta$-defect,

$$Q_\eta^W \in \mathbb{Z} + \frac{1}{2} \quad \Rightarrow \quad e^{2\pi i Q_\eta^W} = -1. \tag{97}$$

This matches the spectral flow formula in (26) (with $\phi = \pi$) in the continuum compact boson theory. As discussed in Section 2, this is a manifestation of a mixed anomaly between the $\mathbb{Z}_2^M$ and $U(1)^W$ symmetries.

---

[24]Using the truncated symmetry operator $\eta_{I,J}$ is a systematic way of finding how inserting an $\eta$-symmetry defect modifies symmetry operators. However, when the defect Hamiltonian does not commute with local operators, there is a practical and often simpler way to find the modified symmetry operators after inserting a symmetry defect in $1+1D$. In particular, they can be found by first observing how (and if) the defect-free symmetry operators fail to commute with the defect-Hamiltonian. If they no longer commute, we then modify them by operators localized near the defect in order to make them commute. This will always be possible if the symmetry remains after inserting the defect. For example, the defect-free translation operator $T$ satisfies $T H_\eta^{\langle I-1,I \rangle} T^{-1} = H_\eta^{\langle I,I+1 \rangle}$ and does not commute with $H_\eta^{\langle I-1,I \rangle}$. However, since $Z_I$ is the defect movement operator, we modify $T$ to $Z_I T \equiv T_\eta$ which does commute with $H_\eta^{\langle I-1,I \rangle}$. This simplified procedure returns the correct operator up to an overall phase factor. However, this phase does not affect the symmetry operators' algebra or their action on states.

[25]To avoid cluttered notation, we will drop the $I$ dependency of the twisted symmetry operator and label them just by the symmetry defect.

Having seen a manifestation of the mixed anomaly after inserting a $\mathbb{Z}_2^{\text{M}}$ defect, let us now insert a $\theta \in \text{U}(1)^{\text{W}}$ symmetry defect instead. For truncating $\text{e}^{\text{i}\theta Q^{\text{W}}}$, it is convenient to first rewrite

$$Q^{\text{W}} = \sum_{j=1}^{L} q_{j,j+1}^{\text{W}}, \qquad q_{j,j+1}^{\text{W}} = -\frac{1}{4} Y_j X_{j+1} (-Z_j Z_{j+1})^j. \tag{98}$$

The $\text{U}(1)^{\text{W}}$ symmetry operator $\text{e}^{\text{i}\theta Q^{\text{W}}}$ can then be rewritten as $\text{e}^{\text{i}\theta Q^{\text{W}}} = \prod_{j=1}^{L} \text{e}^{\text{i}\theta q_{j,j+1}^{\text{W}}}$ because the charge density operators $q_{j,j+1}^{\text{W}}$ all mutually commute (*i.e.*, $q_{j,j+1}^{\text{W}} q_{\ell,\ell+1}^{\text{W}} = q_{\ell,\ell+1}^{\text{W}} q_{j,j+1}^{\text{W}}$). In this form, we now introduce the truncated $\text{U}(1)^{\text{W}}$ symmetry operator $[\text{e}^{\text{i}\theta Q^{\text{W}}}]_{I,J} = \prod_{j=I}^{J} \text{e}^{\text{i}\theta q_{j,j+1}^{\text{W}}}$, which inserts a $\theta$-symmetry defect at $\langle I-1, I \rangle$ and a $(-\theta)$-symmetry defect at $\langle J, J+1 \rangle$. Using this, we find that the defect Hamiltonian with a $\theta \in \text{U}(1)^{\text{W}}$ defect inserted at $\langle I-1, I \rangle$ is

$$\widetilde{H}_\theta^{\langle I-1,I \rangle} = \begin{cases} H_{\text{XX}} + (Y_{I-1}Y_I + Y_I Y_{I+1})(\text{e}^{-\frac{\text{i}\theta}{2} X_I Y_{I+1}} - 1), & I \text{ odd}, \\ H_{\text{XX}} + (X_{I-1}X_I + X_I X_{I+1})(\text{e}^{\frac{\text{i}\theta}{2} Y_I X_{I+1}} - 1), & I \text{ even}. \end{cases} \tag{99}$$

The unitary operator moving the defect from $\langle I-1, I \rangle$ to $\langle I, I+1 \rangle$ is $\text{e}^{-\text{i}\theta q_{I,I+1}^{\text{W}}}$. Furthermore, because the $\text{U}(1)^{\text{W}}$ and $\text{U}(1)^{\text{M}}$ symmetry operators fail to commute, inserting the $\theta$-symmetry defect explicitly breaks $\text{U}(1)^{\text{M}}$ down to its $\mathbb{Z}_2^{\text{M}}$ subgroup. The surviving $\mathbb{Z}_2^{\text{M}}$ symmetry of $\widetilde{H}_\theta^{\langle I-1,I \rangle}$, however, is still generated by $\eta$ since $\eta$ commutes with the truncated operator $[\text{e}^{\text{i}\theta Q^{\text{W}}}]_{I,J}$.

There is something perhaps surprising about the defect Hamiltonian $\widetilde{H}_\theta^{\langle I-1,I \rangle}$: the factor of 2 dividing $\theta$ in (99) makes $\widetilde{H}_\theta^{\langle I-1,I \rangle}$ $4\pi$-periodic in $\theta$ instead of $2\pi$-periodic. Therefore, the fusion rules of the $\theta$-defect are described by a lift of $\text{U}(1)^{\text{W}}$. However, The Hamiltonians at $\theta$ and $\theta + 2\pi$ are unitarily equivalent:

$$\widetilde{H}_{\theta+2\pi}^{\langle I-1,I \rangle} = \begin{cases} X_I \widetilde{H}_\theta^{\langle I-1,I \rangle} X_I, & I \text{ odd}, \\ Y_I \widetilde{H}_\theta^{\langle I-1,I \rangle} Y_I, & I \text{ even}, \end{cases} \tag{100}$$

so the lift can be trivialized using a unitary operator. This is something that can generically happen when inserting symmetry defects [22]. In particular, using the $\theta$-dependent unitary operator

$$V_I(\theta) = \begin{cases} \text{e}^{\frac{\text{i}\theta}{4} X_I}, & I \text{ odd}, \\ \text{e}^{-\frac{\text{i}\theta}{4} Y_I}, & I \text{ even}, \end{cases} \tag{101}$$

we rotate the Hilbert space into a basis where the defect-Hamiltonian becomes

$$H_\theta^{\langle I-1,I \rangle} = V_I(\theta) \widetilde{H}_\theta^{\langle I-1,I \rangle} V_I^{-1}(\theta) = \begin{cases} H_{\text{XX}} + (Y_{I-1}Y_I + Y_I Y_{I+1})(\text{e}^{-\text{i}\theta X_I \frac{1+Y_{I+1}}{2}} - 1), & I \text{ odd}, \\ H_{\text{XX}} + (X_{I-1}X_I + X_I X_{I+1})(\text{e}^{\text{i}\theta Y_I \frac{1+X_{I+1}}{2}} - 1), & I \text{ even}. \end{cases} \tag{102}$$

In this basis, the defect movement operator is still $\text{e}^{-\text{i}\theta q_{I,I+1}^{\text{W}}}$, and the Hamiltonian is now $2\pi$ periodic in $\theta$.

The unitary operator $V_I$ is not $\mathbb{Z}_2^{\text{M}}$-symmetric, and the $\mathbb{Z}_2^{\text{M}}$ symmetry operator $\eta$ in the above basis becomes

$$\eta(\theta) = V_I(\theta) \eta V_I^{-1}(\theta) = \begin{cases} \eta\,\text{e}^{-\frac{\text{i}\theta}{2} X_I}, & I \text{ odd}, \\ \eta\,\text{e}^{\frac{\text{i}\theta}{2} Y_I}, & I \text{ even}. \end{cases} \tag{103}$$

While $\eta(\theta)$ is still a $\mathbb{Z}_2$ operator, it is not $2\pi$ periodic in $\theta$. Instead, $\eta(\theta)$ satisfies

$$\eta(\theta + 2\pi) = -\eta(\theta). \tag{104}$$

However, this is precisely spectral flow and a manifestation of the 't Hooft anomaly. In particular, inserting a $2\pi$ winding symmetry defect by adiabatically tuning $\theta \to \theta + 2\pi$ causes $\mathbb{Z}_2^M$ even (odd) states to become $\mathbb{Z}_2^M$ odd (even) (*i.e.*, pumps a $\mathbb{Z}_2^M$ symmetry charge). Again, this exactly matches the spectral flow formula (27) in the continuum with the identification that $\eta$ flows to $e^{i\pi Q^M}$.

A consequence of the anomaly between $\mathbb{Z}_2^M$ and $U(1)^W$ is that the symmetry operators $e^{i\phi Q^M}$ and $e^{i\theta Q^W}$ can not simultaneously be made onsite. For instance, consider the $\mathbb{Z}_2^M \times \mathbb{Z}_2^W$ sub-symmetry generated by $\eta = e^{i\phi Q^M}$ and $e^{i\pi Q^W}$. Using the lattice T-duality unitary operator $U_T$, we can rotate to a basis where[26]

$$U_T^{-1} e^{i\pi Q^M} U_T = e^{i\pi \widehat{Q}^W} = \prod_{j=1}^{L} X_j \,, \tag{105}$$

$$U_T^{-1} e^{i\pi Q^W} U_T = e^{i\pi \widehat{Q}^M} = \prod_{j=1}^{L} CZ_{j,j+1} \,. \tag{106}$$

Therefore, the $\mathbb{Z}_2$ symmetry operator $e^{i\pi Q^M} e^{i\pi Q^W}$ is unitarily equivalent to $\prod_{j=1}^{L} CZ_{j,j+1} \prod_{k=1}^{L} X_k$, which is a well-known non-onsiteable ("CZX") symmetry operator whose anomaly is classified by the generator of $H^3(\mathbb{Z}_2, U(1)) \simeq \mathbb{Z}_2$ [109]. Therefore, since $e^{i\pi Q^M}$ and $e^{i\pi Q^W}$ cannot be made simultaneously onsite, $e^{i\phi Q^M}$ and $e^{i\theta Q^W}$ cannot as well.

## 4.2 Type III anomaly of $\mathbb{Z}_2^M \times \mathbb{Z}_2^W \times \mathbb{Z}_2^C$

There is an 't Hooft anomaly of $\mathbb{Z}_2^{\mathcal{M}} \times \mathbb{Z}_2^{\mathcal{W}} \times \mathbb{Z}_2^{\mathcal{C}}$ in the compact free boson manifested by the projective representations formed by two of these $\mathbb{Z}_2$ symmetries in the defect Hilbert space of the third $\mathbb{Z}_2$ symmetry (*e.g.*, $\mathbb{Z}_2^{\mathcal{M}} \times \mathbb{Z}_2^{\mathcal{W}}$ realizing a projective representation in the nontrivial $\mathbb{Z}_2^{\mathcal{C}}$ defect Hilbert space). It is often called a type III anomaly [49, 50] due to the form of its SPT theory (28).

In the XX model, the lattice $\mathbb{Z}_2^M \times \mathbb{Z}_2^W \times \mathbb{Z}_2^C$ symmetry is generated by the unitary operators[27]

$$\eta = \prod_{j=1}^{L} (-1)^j Z_j \,, \qquad e^{i\pi Q^W} = \prod_{n=1}^{L/2} e^{i\frac{\pi}{4} X_{2n-1} Y_{2n}} e^{-i\frac{\pi}{4} Y_{2n} X_{2n+1}} \,, \qquad C = \prod_{j=1}^{L} X_j \,. \tag{108}$$

The winding symmetry operator $e^{i\pi Q^W}$ is relatively complicated, but it acts on the Pauli oper-

---

[26]In deriving this expression for $e^{i\pi \widehat{Q}^M}$, we used that for even $L$, $\widehat{Q}^M = \frac{1}{4} \sum_{j=1}^{L} (-1)^j Z_j Z_{j+1}$ can be written as $-\frac{1}{2} \sum_{j=1}^{L} (-1)^j CZ_{j,j+1}$.

[27]Using the unitary operator $U_T^{-1}$ whose action on the Pauli operators is (56), we can change basis to where the $\mathbb{Z}_2^M \times \mathbb{Z}_2^W \times \mathbb{Z}_2^C$ symmetry operators become

$$e^{i\pi \widehat{Q}^W} = \prod_{j=1}^{L} X_j \,, \qquad e^{i\pi \widehat{Q}^M} = \prod_{j=1}^{L} CZ_{j,j+1} \,, \qquad \widehat{C} = \prod_{j=1}^{L} [X_j]^{j+1} \,. \tag{107}$$

These are known to generate a $\mathbb{Z}_2 \times \mathbb{Z}_2 \times \mathbb{Z}_2$ symmetry with a type III anomaly (see, *e.g.*, Ref. 90). This anomaly falls into a class of anomalous symmetries formed by $G$ symmetry operators and a $G$-SPT entangler. In particular, $e^{i\pi \widehat{Q}^M}$ is the SPT entangler for an SPT—the cluster state—protected by the $G = \mathbb{Z}_2 \times \mathbb{Z}_2$ symmetry generated by $e^{i\pi \widehat{Q}^W}$ and $\widehat{C}$. Indeed, these three operators are symmetries of the Hamiltonian (48), which is the transition point between the paramagnet and cluster Hamiltonians.

ators as

$$e^{i\pi Q^W} X_j e^{-i\pi Q^W} = \begin{cases} X_j, & j \text{ odd}, \\ X_{j-1}X_j X_{j+1}, & j \text{ even}, \end{cases} \tag{109}$$

$$e^{i\pi Q^W} Y_j e^{-i\pi Q^W} = \begin{cases} Y_{j-1}Y_j Y_{j+1}, & j \text{ odd}, \\ Y_j, & j \text{ even}. \end{cases} \tag{110}$$

In what follows, we will see the projective algebras arising from inserting each of the $\mathbb{Z}_2$ defects in the context of the XX model and confirm the type III anomaly of $\mathbb{Z}_2^M \times \mathbb{Z}_2^W \times \mathbb{Z}_2^C$.

We first insert an $\eta$-symmetry defect at the link $\langle I-1, I \rangle$. As was worked out in the previous section, the defect Hamiltonian is (95) and the winding symmetry charge becomes (96). The modified winding symmetry charge causes the $\mathbb{Z}_2^W$ symmetry operator to become

$$e^{i\pi Q_\eta^W} = \begin{cases} i\, e^{i\pi Q^W} Y_{I-1}X_I, & I \text{ odd}, \\ -i\, e^{i\pi Q^W} X_{I-1}Y_I, & I \text{ even}. \end{cases} \tag{111}$$

On the other hand, the charge conjugation symmetry operator $C$ does not change: $C_\eta = C$. This can be seen from the fact that $C$ still commutes with the $\eta$-defect Hamiltonian (95). After inserting the $\eta$-symmetry defect, the $\mathbb{Z}_2^W$ and $\mathbb{Z}_2^C$ symmetry operators obey the projective algebra

$$C_\eta\, e^{i\pi Q_\eta^W} = -e^{i\pi Q_\eta^W}\, C_\eta. \tag{112}$$

This nontrivial projectivity is precisely the expected manifestation of the type III anomaly for the $\mathbb{Z}_2^M \times \mathbb{Z}_2^W \times \mathbb{Z}_2^C$ symmetry.

Let us next check that inserting a $e^{i\pi Q^W}$ symmetry defect causes $\mathbb{Z}_2^M \times \mathbb{Z}_2^C$ to realize a projective representation. In the previous section, we inserted a $e^{i\theta Q^W}$ symmetry defect at $\langle I-1, I \rangle$ for general $\theta$. The defect Hamiltonian (102) for $\theta = \pi$ simplifies to

$$H_W^{\langle I-1, I \rangle} = \begin{cases} H_{XX} - (Y_{I-1}Y_I + Y_I Y_{I+1})(Y_{I+1} + 1), & I \text{ odd}, \\ H_{XX} - (X_{I-1}X_I + X_I X_{I+1})(X_{I+1} + 1), & I \text{ even}. \end{cases} \tag{113}$$

After inserting this symmetry defect, the $\mathbb{Z}_2^M$ and $\mathbb{Z}_2^C$ symmetry operators become

$$\eta_W = \begin{cases} -i\,\eta X_I, & I \text{ odd}, \\ i\,\eta Y_I, & I \text{ even}, \end{cases} \qquad C_W = \begin{cases} X_I C, & I \text{ odd}, \\ C, & I \text{ even}. \end{cases} \tag{114}$$

The modified $\mathbb{Z}_2^M$ symmetry operator for general $\theta$ is given by (103), and we arrive at the above expression by setting $\theta = \pi$. Furthermore, we found the modified $\mathbb{Z}_2^C$ symmetry operator by modifying $C$ such that it commutes with $H_W^{\langle I-1, I \rangle}$. After inserting the $e^{i\pi Q^W}$ symmetry defect, the $\mathbb{Z}_2^M \times \mathbb{Z}_2^C$ symmetry operators satisfy the projective algebra

$$\eta_W C_W = -C_W \eta_W. \tag{115}$$

Again, this is the expected signature of a type III anomaly for the $\mathbb{Z}_2^M \times \mathbb{Z}_2^W \times \mathbb{Z}_2^C$ symmetry.

Lastly, the Hamiltonian with a $C$ defect at link $\langle I-1, I \rangle$ is

$$H_C^{\langle I-1, I \rangle} = H_{XX} - 2Y_{I-1}Y_I. \tag{116}$$

The $\mathbb{Z}_2^M$ and $\mathbb{Z}_2^W$ symmetry operators in the presence of the $C$-defect become

$$\eta_C = \eta, \qquad e^{i\pi Q_C^W} = \begin{cases} X_I\, e^{i\pi Q^W}, & I \text{ odd}, \\ X_{I-1}\, e^{i\pi Q^W}, & I \text{ even}. \end{cases} \tag{117}$$

The $\mathbb{Z}_2^M$ symmetry operator $\eta$ still commutes with this defect Hamiltonian and, therefore, is not modified by the symmetry defect. The $\mathbb{Z}_2^W$ symmetry operator $e^{i\pi Q^W}$ does not commute with $H_C^{\langle I-1, I\rangle}$, and the above expression is found by modifying $e^{i\pi Q^W}$ to make it commute. The $\mathbb{Z}_2^M$ and $\mathbb{Z}_2^W$ symmetry operators after inserting the $C$-symmetry defect form the projective algebra

$$\eta_C \, e^{i\pi Q_C^W} = -e^{i\pi Q_C^W} \eta_C \,. \tag{118}$$

Once again, this matches the expectation from the type III anomaly for the $\mathbb{Z}_2^M \times \mathbb{Z}_2^W \times \mathbb{Z}_2^C$ symmetry.

All three $\mathbb{Z}_2$ symmetries obey these projective algebras upon inserting respective defects. Therefore, we find that the $\mathbb{Z}_2^M \times \mathbb{Z}_2^W \times \mathbb{Z}_2^C$ symmetry of the XX model has a type III anomaly, which is the same as the type III anomaly of $\mathbb{Z}_2^{\mathcal{M}} \times \mathbb{Z}_2^{\mathcal{W}} \times \mathbb{Z}_2^{\mathcal{C}}$ in the compact free boson.

### 4.3 Anomaly of the non-invertible symmetry

Just as invertible symmetries can have 't Hooft anomalies, non-invertible symmetries can as well [110–115]. When discussing 't Hooft anomalies of non-invertible symmetries, it is useful to define an 't Hooft anomaly as an obstruction to a Symmetry-Protected Topological (SPT) phase (*i.e.*, a phase with a symmetric, non-degenerate gapped ground state on all spatial manifolds) [32,110,111,115,116]. This definition is motivated by 't Hooft anomaly matching and avoids having to define the gauging of non-invertible symmetries. We will now argue that the non-invertible symmetry D arising from lattice T-duality is anomalous in this sense.

To do so, we first restrict ourselves to spin chains for which the number of sites $L = 0 \mod 4$. For such system sizes, we can define the unitary operator

$$U_4 = \prod_{n=1}^{L/4} Z_{4n+1} X_{4n+2} Y_{4n+3} \,. \tag{119}$$

Using this unitary operator and the unitary $U$ given by (82), we can rotate the Hilbert space to a basis in which

$$(U_4 U) H_{XX} (U_4 U)^{-1} = \sum_{j=1}^{L} (Z_j Y_{j+1} - Y_j Z_{j+1}) \,, \tag{120}$$

$$(U_4 U) D (U_4 U)^{-1} = D_{KW} \,. \tag{121}$$

Eq. (121) follows from (86) and (87). $D_{KW}$ is the non-invertible symmetry operator of the critical Ising chain, which satisfies

$$D_{KW} X_j = Z_j Z_{j+1} D_{KW} \,, \qquad D_{KW} Z_j Z_{j+1} = X_{j+1} D_{KW} \,. \tag{122}$$

The benefit of this basis transformation is that the operator D becomes the well-known non-invertible Kramers-Wannier symmetry operator $D_{KW}$, which has been shown to be anomalous [76,117]. Therefore, because D and $D_{KW}$ are related by the finite-depth local unitary $U_4 U$ when $L = 0 \mod 4$, the symmetry generated by D is also anomalous when $L = 0 \mod 4$. However, the symmetry generated by D does not change when $L = 2 \mod 4$. In particular, its operator algebra is the same for all even $L$. (This is to be contrasted with odd $L$, for which D no longer commutes with $H_{XX}$.) Therefore, this obstruction to an SPT is expected to persist for all even $L$, and the symmetry generated by D has an 't Hooft anomaly. The 't Hooft anomaly for $L = 2 \mod 4$ can likely be argued more rigorously by generalizing the arguments from Refs. 117 and 76, but it is not something we will pursue here.

## 4.4 Anomaly enforced gaplessness

Thus far, we have seen how manifestations of 't Hooft anomalies involving the momentum and winding symmetries in the XX model are the same as those in the compact free boson. In this subsection, we will show that any local Hamiltonian (defined on the same Hilbert space as the XX model) that is both $U(1)^M$ and $U(1)^W$ symmetric must be gapless. To do this, we will use a theorem from Ref. 39 proven in the fermionized counterpart of the XX model.

Fermionizing the XX model on $L$ sites gives rise to a local fermionic Hamiltonian on $L$ sites, where each site $j$ carries a two-dimensional Hilbert space $\mathcal{H}^f_j$ acted on by the complex fermion operators $c_j$ and $c_j^\dagger$. It is implemented by a map that relates $\eta$-symmetric operators of the XX model to $(-1)^F \equiv (-1)^{\sum_{j=1}^L (c_j^\dagger c_j - \frac{1}{2})}$ symmetric operators of the fermionic model. For the fermionization map we consider, it is convenient to rewrite the complex fermion creation operator $c_j^\dagger$ acting on $\mathcal{H}^f_j$ as $c_j^\dagger = \frac{1}{2}(a_j - \mathrm{i} b_j)$, where the real fermion operators $a_j$ and $b_j$ satisfy

$$\{a_j, b_{j'}\} = 0, \qquad \{a_j, a_{j'}\} = 2\delta_{j,j'}, \qquad \{b_j, b_{j'}\} = 2\delta_{j,j'}. \tag{123}$$

In terms of these real fermion operators, the fermionization map is specified by

$$Z_j \to \mathrm{i} a_j b_j, \qquad X_j X_{j+1} \to \begin{cases} -\mathrm{i} a_j a_{j+1}, & j \text{ odd}, \\ -\mathrm{i} b_j b_{j+1}, & j \text{ even}. \end{cases} \tag{124}$$

We refer the reader to Appendix B.2.3 for a detailed derivation.

The map (124) fully specifies the fermionization because any $\eta$-symmetric operator can be constructed from $Z_j$ and $X_j X_{j+1}$. For example, the $\eta$-symmetric operator $Y_j Y_{j+1}$ appearing in the XX model Hamiltonian is a product of the above operators. Under fermionization, it becomes

$$Y_j Y_{j+1} = Z_j X_j X_{j+1} Z_{j+1} \to \begin{cases} -\mathrm{i} b_j b_{j+1}, & j \text{ odd}, \\ -\mathrm{i} a_j a_{j+1}, & j \text{ even}. \end{cases} \tag{125}$$

Using this, we find that fermionizing the XX model Hamiltonian yields the two-flavor Majorana chain Hamiltonian

$$H_{\text{2maj}} = -\mathrm{i} \sum_{j=1}^L (a_j a_{j+1} + b_j b_{j+1}), \tag{126}$$

with periodic boundary conditions. Furthermore, the $U(1)^M$ and $U(1)^W$ symmetry charges $Q^M$ and $Q^W$ become

$$Q^M \to \frac{1}{2} \sum_{j=1}^L \mathrm{i} a_j b_j \equiv Q^V, \qquad Q^W \to \frac{1}{4} \sum_{j=1}^L \mathrm{i} a_j b_{j+1} \equiv \frac{1}{2} Q^A. \tag{127}$$

Here, $Q^V$ and $Q^A$ denote the vector and axial symmetry charges of the Hamiltonian (126) discussed in Ref. 39. 

Fermionizing any local bosonic Hamiltonian commuting with $Q^M$ and $Q^W$ yields a local fermionic Hamiltonian commuting with $Q^V$ and $Q^A$. In fact, there is a one-to-one correspondence of such bosonic and fermionic models. In Ref. 39, it was proven that any $Q^V$ and $Q^A$ symmetric deformation to (126) never gaps out the theory. In other words, the axial and vector lattice symmetries enforce gaplessness. While fermionization changes the global spectrum on a closed chain, it does not change the scaling behavior of the energy gap. Therefore, the theorem proved in Ref. 39 for the fermion models implies that the lattice charges $Q^M$ and $Q^W$ enforce gaplessness in any symmetric spin chain model. This is reminiscent of the consequence of the anomaly in the continuum compact boson theory.

It is interesting to perform the the unitary transformation $(X_{2n}, Y_{2n}) \rightarrow (Y_{2n}, X_{2n})$, which implements the transformations

$$Q^{\mathrm{M}} \rightarrow -\frac{1}{2} \sum_{j=1}^{L} Z_j, \qquad Q^{\mathrm{W}} \rightarrow -\frac{1}{4} \sum_{j=1}^{L} (-1)^j X_j X_{j+1}. \tag{128}$$

Therefore, the gapless constraint induced by $Q^{\mathrm{M}}$ and $Q^{\mathrm{W}}$ implies that any even-length quantum spin chain whose Hamiltonian commutes with $\sum_{j=1}^{L} Z_j$ and $\sum_{j=1}^{L} (-1)^j X_j X_{j+1}$ is gapless. When $L = 0 \bmod 4$, the unitary transformation $(X_j, Y_j) \rightarrow (-X_j, -Y_j)$ for $j = 0, 1 \bmod 4$ further implies that any Hamiltonian commuting with $\sum_{j=1}^{L} Z_j$ and $\sum_{j=1}^{L} X_j X_{j+1}$ must be gapless. In fact, the operators $\sum_{j=1}^{L} Z_j$ and $\sum_{j=1}^{L} X_j X_{j+1}$ were the ones consider by Onsager in Ref. 69 as generators of the now-called Onsager algebra (see Section 5.1).

### 4.4.1 Proof

In the remainder of this subsection, we review the proof from Ref. 39 that $Q^{\mathrm{V}}$ and $Q^{\mathrm{A}}$ symmetric Hamiltonians are always gapless. The U(1)$^{\mathrm{V}}$ and U(1)$^{\mathrm{A}}$ symmetries act on the Majorana fermions as

$$
\begin{aligned}
Q^{\mathrm{V}} &= \frac{\mathrm{i}}{2} \sum_{j=1}^{L} a_j b_j, & e^{\mathrm{i}\varphi Q^{\mathrm{V}}} \begin{pmatrix} a_j \\ b_j \end{pmatrix} e^{-\mathrm{i}\varphi Q^{\mathrm{V}}} &= \begin{pmatrix} \cos\varphi \; a_j + \sin\varphi \; b_j \\ \cos\varphi \; b_j - \sin\varphi \; a_j \end{pmatrix}, \\
Q^{\mathrm{A}} &= \frac{\mathrm{i}}{2} \sum_{j=1}^{L} a_j b_{j+1}, & e^{\mathrm{i}\varphi Q^{\mathrm{A}}} \begin{pmatrix} a_j \\ b_j \end{pmatrix} e^{-\mathrm{i}\varphi Q^{\mathrm{A}}} &= \begin{pmatrix} \cos\varphi \; a_j + \sin\varphi \; b_{j+1} \\ \cos\varphi \; b_j - \sin\varphi \; a_{j-1} \end{pmatrix}.
\end{aligned}
\tag{129}
$$

Therefore, the operator $e^{-\mathrm{i}\frac{\pi}{2}Q^{\mathrm{A}}} e^{\mathrm{i}\frac{\pi}{2}Q^{\mathrm{V}}}$ acts as

$$e^{-\mathrm{i}\frac{\pi}{2}Q^{\mathrm{A}}} e^{\mathrm{i}\frac{\pi}{2}Q^{\mathrm{V}}} \begin{pmatrix} a_j \\ b_j \end{pmatrix} e^{-\mathrm{i}\frac{\pi}{2}Q^{\mathrm{V}}} e^{\mathrm{i}\frac{\pi}{2}Q^{\mathrm{A}}} = \begin{pmatrix} a_{j-1} \\ b_{j+1} \end{pmatrix}. \tag{130}$$

This action on the real fermion operators is the same as the action from $T_b T_a^{-1}$, where $T_a$ and $T_b$ are Majorana translation operators satisfying

$$T_a a_j T_a^{-1} = a_{j+1}, \qquad T_a b_j T_a^{-1} = b_j, \qquad T_b a_j T_b^{-1} = a_j, \qquad T_b b_j T_b^{-1} = b_{j+1}. \tag{131}$$

Thus, any Hamiltonian commuting with $Q^{\mathrm{V}}$ and $Q^{\mathrm{A}}$ also commutes with $T_b T_a^{-1}$.

Repeated actions of $T_b T_a^{-1}$ makes any operators constructed from both $a$ and $b$ operators increasingly non-local. Therefore, any local Hamiltonian must include terms made of only $a$ or only $b$ real fermion operators. Consider $2\ell$-fermion operators involving only $a$ or $b$ fermions. For $\ell = 1$, these take the form $a_j a_{j+n}$ and $b_j b_{j+n}$. However, the U(1)$^{\mathrm{V}}$ symmetry requires that they appear in combinations $a_j a_{j+n} + b_j b_{j+n}$. A general allowed linear combination of such operators for a fixed $n$ is

$$\sum_{j=1}^{L} g_{j,n} \left( \mathrm{i} a_j a_{j+n} + \mathrm{i} b_j b_{j+n} \right), \tag{132}$$

which under the action of $T_b T_a^{-1}$ becomes

$$\sum_{j=1}^{L} g_{j,n} \left( \mathrm{i} a_{j-1} a_{j+n-1} + \mathrm{i} b_{j+1} b_{j+n+1} \right) = \sum_{j=1}^{L} (g_{j+1,n} \mathrm{i} a_j a_{j+n} + g_{j-1,n} \mathrm{i} b_j b_{j+n}). \tag{133}$$

However, $T_b T_a^{-1}$ invariance requires $g_{j+1,n} = g_{j,n}$ for all $j$. Therefore, we drop the subscript $j$ in $g_{j,n}$ and write the above general quadratic $Q^V$ and $Q^A$ symmetric deformation as

$$\sum_{j=1}^{L} g_n \left( i a_j a_{j+n} + i b_j b_{j+n} \right). \tag{134}$$

Having worked out the $\ell = 1$ case, we next consider $2\ell$-fermion operators with $\ell \geq 2$. These are all of the form

$$O_{2\ell}^a(j_1, \ldots, j_{2\ell}) = \prod_{m=1}^{2\ell} a_{j_m}, \qquad O_{2\ell}^b(j_1, \ldots, j_{2\ell}) = \prod_{m=1}^{2\ell} b_{j_m}. \tag{135}$$

An infinitesimal $e^{i\varphi Q^V}$ transformation acts on these operators as

$$O_{2\ell}^a(j_1, \ldots, j_{2\ell}) \to O_{2\ell}^a(j_1, \ldots, j_{2\ell}) + \varphi \sum_{m=1}^{2\ell} a_{j_1} \cdots a_{j_{m-1}} b_{j_m} a_{j_{m+1}} \cdots a_{j_{2\ell}} + \mathcal{O}(\varphi^2),$$

$$O_{2\ell}^b(j_1, \ldots, j_{2\ell}) \to O_{2\ell}^b(j_1, \ldots, j_{2\ell}) - \varphi \sum_{m=1}^{2\ell} b_{j_1} \cdots b_{j_{m-1}} a_{j_m} b_{j_{m+1}} \cdots b_{j_{2\ell}} + \mathcal{O}(\varphi^2). \tag{136}$$

Each $\mathcal{O}(\varphi)$ deformation of $O_{2\ell}^a$ consists of $2\ell - 1$ of the $a$ fermions and one $b$ fermion, while that of $O_{2\ell}^b$ consists of $2\ell - 1$ of the $b$ fermions and one $a$ fermion. Therefore, these deformations must be linearly independent when $\ell \geq 2$. Consequently, there is no linear combination of $O_{2\ell}^a$ and $O_{2\ell}^b$ with $\ell \geq 2$ that preserve the $U(1)^V$ symmetry.

From the above arguments, we have found that all $Q^V$ and $Q^A$ symmetric local Hamiltonians are constructed from (134) and of the form

$$H_{V,A} = -\sum_{j=1}^{L} \left( \sum_{n=1}^{N} g_n \left( i a_j a_{j+n} + i b_j b_{j+n} \right) \right) = -2i \sum_{j=1}^{L} \left( \sum_{n=1}^{N} g_n \left( c_j^\dagger c_{j+n} + c_j c_{j+n}^\dagger \right) \right). \tag{137}$$

In order for $H_{V,A}$ to be local, we assume that $N/L \to 0$ in the $L \to \infty$ limit. This is a gapless Hamiltonian for all choices of $g_n$, which has been discussed in, for example, Ref. 118. Indeed, performing the Fourier transformation

$$c_j = \frac{1}{\sqrt{L}} \sum_{-L/2 \leq k < L/2} e^{\frac{2\pi i k j}{L}} \gamma_k, \tag{138}$$

$H_{V,A}$ in momentum space becomes

$$H_{V,A} = \sum_{-L/2 \leq k < L/2} \left( \sum_{n=1}^{N} 4 g_n \sin \frac{2\pi k n}{L} \right) \gamma_k^\dagger \gamma_k. \tag{139}$$

For any non-trivial choice of the parameters $\{g_n\}$, this Hamiltonian has an algebraically vanishing spectral gap in the $L \to \infty$ limit.

## 5 The Onsager algebra

As we saw in section 3.2, unlike in the compact free boson, the $U(1)^M$ and $U(1)^W$ symmetries in the XX model do not commute. Instead, they generate an extensively-large Lie algebra known as the Onsager algebra, which was demonstrated in Refs. [36, 37, 39]. In this section, we will further explore the interplay of the conserved charges forming the Onsager algebra with lattice T-duality and the various related non-invertible symmetries of the XX model. In particular, we will see that the non-invertible and translation symmetries in the XX model have a nontrivial interplay with the conserved Onsager charges.

### 5.1 Onsager charges from bosonization

Before getting into this rich interplay, we first review the conserved Onsager charges in the XX model. The Onsager charges are simplest to introduce in the fermionized version (126) of the XX model. In terms of the real fermion operators $a_j$ and $b_j$, the conserved Onsager charges of the two-Majorana chain (126) are [39]

$$Q_n^{\mathrm{f}} = \frac{\mathrm{i}}{2} \sum_{j=1}^{L} a_j b_{j+n}, \qquad G_n^{\mathrm{f}} = \frac{\mathrm{i}}{2} \sum_{j=1}^{L} (a_j a_{j+n} - b_j b_{j+n}). \tag{140}$$

Notice that the axial and vector charges (127) are contained within these expressions, given by $Q_0^{\mathrm{f}} = Q^{\mathrm{V}}$ and $Q_1^{\mathrm{f}} = Q^{\mathrm{A}}$. The Onsager charges get their name from the fact that they satisfy the Onsager algebra [69]

$$[Q_n^{\mathrm{f}}, Q_m^{\mathrm{f}}] = \mathrm{i} G_{m-n}^{\mathrm{f}}, \qquad [G_n^{\mathrm{f}}, G_m^{\mathrm{f}}] = 0, \qquad [Q_n^{\mathrm{f}}, G_m^{\mathrm{f}}] = 2\mathrm{i}(Q_{n-m}^{\mathrm{f}} - Q_{n+m}^{\mathrm{f}}). \tag{141}$$

From these relations, it is clear that $Q_0^{\mathrm{f}} = Q^{\mathrm{V}}$ and $Q_1^{\mathrm{f}} = Q^{\mathrm{A}}$ are generators of this algebra.

To relate this fermion model and its conserved operators to the XX model, we implement the bosonization map

$$\mathrm{i} a_j b_j \to Z_j, \qquad \mathrm{i} a_j b_{j+1} \to \begin{cases} X_j Y_{j+1}, & j \text{ odd}, \\ -Y_j X_{j+1}, & j \text{ even}. \end{cases} \tag{142}$$

This bosonization is the "inverse" of the fermionization map (124).[28] Indeed, using that the bilinears $\mathrm{i} a_j a_{j+1} = \mathrm{i}(\mathrm{i} a_j b_{j+1})(\mathrm{i} a_{j+1} b_{j+1})$ and $\mathrm{i} b_j b_{j+1} = \mathrm{i}(\mathrm{i} a_j b_j)(\mathrm{i} a_j b_{j+1})$, their image under this bosonization is

$$\mathrm{i} a_j a_{j+1} \to \begin{cases} -X_j X_{j+1}, & j \text{ odd}, \\ -Y_j Y_{j+1}, & j \text{ even}, \end{cases} \tag{143}$$

$$\mathrm{i} b_j b_{j+1} \to \begin{cases} -Y_j Y_{j+1}, & j \text{ odd}, \\ -X_j X_{j+1}, & j \text{ even}. \end{cases} \tag{144}$$

We refer the reader to Appendix B.2.1 for further discussion and a detailed derivation of this bosonization procedure.

Let us denote by $Q_n$ and $G_n$ the image of $Q_n^{\mathrm{f}}$ and $G_n^{\mathrm{f}}$ under this bosonization map. Just like $Q_n^{\mathrm{f}}$ and $G_n^{\mathrm{f}}$, they also satisfy the Onsager algebra

$$[Q_n, Q_m] = \mathrm{i} G_{m-n}, \qquad [G_n, G_m] = 0, \qquad [Q_n, G_m] = 2\mathrm{i}(Q_{n-m} - Q_{n+m}). \tag{145}$$

It is straightforward to find the explicit, albeit cumbersome, expressions for $Q_n$ and $G_n$ in terms of the Pauli operators using (142) (see Appendix C). For example, we find that

$$Q_0 = Q^{\mathrm{M}}, \qquad Q_1 = 2Q^{\mathrm{W}}, \qquad G_1 = \frac{1}{2} \sum_n (-1)^j (X_j X_{j+1} - Y_j Y_{j+1}), \tag{146}$$

which is consistent with (127).

There are simpler and more enlightening expressions for $Q_n$ and $G_n$ in terms of the momentum and winding charges $Q^{\mathrm{M}}$ (37) and $Q^{\mathrm{W}}$ (59). To derive them, we introduce the images of

---

[28]Importantly, the bosonization map is not an invertible map on the entire Hilbert space. More precisely, these bosonization and fermionization maps are inverses of each other when restricted to the $\eta$ and $(-1)^F$ even local operators and states of the XX and the two-flavor Majorana models, respectively.

the Majorana translation operators $T_a$ and $T_b$ under bosonization by $\mathsf{D}_a$ and $\mathsf{D}_b$, respectively. Since $T_a$ and $T_b$ satisfy

$$T_a Q_n^{\mathrm{f}} = Q_{n-1}^{\mathrm{f}} T_a, \qquad T_b Q_n^{\mathrm{f}} = Q_{n+1}^{\mathrm{f}} T_b, \qquad T_a G_n^{\mathrm{f}} = G_n^{\mathrm{f}} T_a, \qquad T_b G_n^{\mathrm{f}} = G_n^{\mathrm{f}} T_b, \qquad (147)$$

the operators $\mathsf{D}_a$ and $\mathsf{D}_b$ satisfy

$$\mathsf{D}_a Q_n = Q_{n-1} \mathsf{D}_a, \qquad \mathsf{D}_b Q_n = Q_{n+1} \mathsf{D}_b, \qquad \mathsf{D}_a G_n = G_n \mathsf{D}_a, \qquad \mathsf{D}_b G_n = G_n \mathsf{D}_b. \qquad (148)$$

The first two expressions are particularly suggestive that $Q_n$ can be solved for recursively with the "initial conditions" $Q_0 = Q^{\mathrm{M}}$ and $Q_1 = 2Q^{\mathrm{W}}$. Then, the related expressions for $G_n$ would follow from the Onsager algebra relation $[Q_n, Q_m] = \mathrm{i} G_{m-n}$.

To do so, however, we need to know a bit more about the operators $\mathsf{D}_a$ and $\mathsf{D}_b$. In particular, since $T_a$ and $T_b$ satisfy

$$T_a(\mathrm{i} a_j b_j) = (\mathrm{i} a_{j+1} b_j) T_a, \quad T_a(\mathrm{i} b_j b_{j+1}) = (\mathrm{i} b_j b_{j+1}) T_a, \quad T_a(\mathrm{i} a_j a_{j+1}) = (\mathrm{i} a_{j+1} a_{j+2}) T_a,$$
$$T_b(\mathrm{i} a_j b_j) = (\mathrm{i} a_j b_{j+1}) T_b, \quad T_b(\mathrm{i} b_j b_{j+1}) = (\mathrm{i} b_{j+1} b_{j+2}) T_b, \quad T_b(\mathrm{i} a_j a_{j+1}) = (\mathrm{i} a_j a_{j+1}) T_b,$$

we can deduce from the bosonization map (142) that

$$\mathsf{D}_a Z_j = \begin{cases} (-Y_j X_{j+1}) \mathsf{D}_a, & j \text{ odd}, \\ (X_j Y_{j+1}) \mathsf{D}_a, & j \text{ even}, \end{cases} \qquad \mathsf{D}_a(X_j X_{j+1}) = \begin{cases} (Y_{j+1} Y_{j+2}) \mathsf{D}_a, & j \text{ odd}, \\ (X_j X_{j+1}) \mathsf{D}_a, & j \text{ even}, \end{cases}$$
$$\mathsf{D}_b Z_j = \begin{cases} (X_j Y_{j+1}) \mathsf{D}_b, & j \text{ odd}, \\ (-Y_j X_{j+1}) \mathsf{D}_b, & j \text{ even}, \end{cases} \qquad \mathsf{D}_b(X_j X_{j+1}) = \begin{cases} (X_j X_{j+1}) \mathsf{D}_b, & j \text{ odd}, \\ (Y_{j+1} Y_{j+2}) \mathsf{D}_b, & j \text{ even}. \end{cases} \qquad (149)$$

These expressions are useful because they are similar to the action of $\mathsf{D}$ on $\eta$-even operators given by (71). In fact, from their actions on these Pauli operators, we find that

$$\mathsf{D}_a = \mathrm{e}^{\mathrm{i}\frac{\pi}{2}Q^{\mathrm{M}}} \mathsf{D}, \qquad \mathsf{D}_b = \mathrm{e}^{\mathrm{i}\pi Q^{\mathrm{W}}} \mathsf{D}. \qquad (150)$$

In particular, these expressions make it clear that $\mathsf{D}_a, \mathsf{D}_b$ are non-invertible (since $\mathsf{D}$ is). Indeed, bosonization turns the invertible Majorana translation operators $T_a, T_b$ into non-invertible operators [75, 119].

Equipped with Eq. (150), the first two equations of (148) can be written as

$$\mathsf{D} Q_n = \left( \mathrm{e}^{-\mathrm{i}\frac{\pi}{2}Q^{\mathrm{M}}} Q_{n-1} \mathrm{e}^{\mathrm{i}\frac{\pi}{2}Q^{\mathrm{M}}} \right) \mathsf{D}, \qquad \mathsf{D} Q_n = \left( \mathrm{e}^{-\mathrm{i}\pi Q^{\mathrm{W}}} Q_{n+1} \mathrm{e}^{\mathrm{i}\pi Q^{\mathrm{W}}} \right) \mathsf{D}. \qquad (151)$$

These can now be used to recursively solve for $Q_n$ since we know that $\mathsf{D}$ acts on $Q^{\mathrm{M}}$ and $Q^{\mathrm{W}}$ as (73). For example, setting $n = 1$ and using $\mathsf{D} Q^{\mathrm{W}} = \frac{1}{2} Q^{\mathrm{M}} \mathsf{D}$, we find that

$$\mathsf{D}(2Q^{\mathrm{W}}) = \left( \mathrm{e}^{-\mathrm{i}\pi Q^{\mathrm{W}}} Q_2 \mathrm{e}^{\mathrm{i}\pi Q^{\mathrm{W}}} \right) \mathsf{D} \implies Q_2 = \mathrm{e}^{\mathrm{i}\pi Q^{\mathrm{W}}} Q^{\mathrm{M}} \mathrm{e}^{-\mathrm{i}\pi Q^{\mathrm{W}}}, \qquad (152)$$

thereby expressing $Q_2$ in terms of $Q^{\mathrm{M}}$ and $Q^{\mathrm{W}}$. Using this expression for $Q_2$, we can find the expression for $Q_3$ using (151) and continue on to find $Q_n$ for all $n > 0$. Similarly, setting $n = 0$ and using $\mathsf{D} Q^{\mathrm{M}} = 2Q^{\mathrm{W}} \mathsf{D}$, we find that

$$\mathsf{D} Q^{\mathrm{M}} = \left( \mathrm{e}^{-\mathrm{i}\frac{\pi}{2}Q^{\mathrm{M}}} Q_{-1} \mathrm{e}^{\mathrm{i}\frac{\pi}{2}Q^{\mathrm{M}}} \right) \mathsf{D} \implies Q_{-1} = 2 \mathrm{e}^{\mathrm{i}\frac{\pi}{2}Q^{\mathrm{M}}} Q^{\mathrm{W}} \mathrm{e}^{-\mathrm{i}\frac{\pi}{2}Q^{\mathrm{M}}}. \qquad (153)$$

Again, using this we can find $Q_{-2}$ in terms of $Q^{\mathrm{M}}$ and $Q^{\mathrm{W}}$, and eventually $Q_n$ for all $n < 0$. The general expression for $Q_n$ using this recursive solution is[29]

$$Q_n = \begin{cases} 2 S_n Q^{\mathrm{W}} S_n^{-1}, & n \text{ odd}, \\ S_n Q^{\mathrm{M}} S_n^{-1}, & n \text{ even}, \end{cases} \qquad (154)$$

---

[29]This derivation of $Q_n$ given by (154) is ambiguous up to multiplying $Q_n$ by the operator $\eta$ since $\eta \mathsf{D} = \mathsf{D}\eta = \mathsf{D}$. However, the expression (154) is correct since it satisfies a crucial locality property obeyed by the explicit expressions from Appendix C: each $Q_n$ is a sum of operators $q_n^{(j)}$ acting within a size $\sim n$ neighborhood of each site $j$. Replacing $Q_n$ by $Q_n \eta$ or $\eta Q_n$ would spoil this property. Furthermore, while conjugating $Q_n$ by $\eta$ preserves this locality, each $Q_n$ commutes with $\eta$, so it does not affect their expressions.

where the unitary operator $S_n$ is an alternating product of $\mathrm{e}^{\mathrm{i}\pi Q^{\mathrm{W}}}$ and $\mathrm{e}^{\mathrm{i}\frac{\pi}{2}Q^{\mathrm{M}}}$, given by

$$S_n = \begin{cases} \prod_{k=1}^{|n|-1} s_k, & n \geq 0, \\ \mathrm{e}^{\mathrm{i}\frac{\pi}{2}Q^{\mathrm{M}}} \prod_{k=1}^{|n|-1} s_k, & n < 0, \end{cases} \qquad s_k = \begin{cases} \mathrm{e}^{\mathrm{i}\pi Q^{\mathrm{W}}}, & k \text{ odd}, \\ \mathrm{e}^{\mathrm{i}\frac{\pi}{2}Q^{\mathrm{M}}}, & k \text{ even}. \end{cases} \tag{155}$$

For example,

$$S_0 = S_1 = 1, \qquad S_2 = \mathrm{e}^{\mathrm{i}\pi Q^{\mathrm{W}}}, \qquad S_3 = \mathrm{e}^{\mathrm{i}\pi Q^{\mathrm{W}}} \mathrm{e}^{\mathrm{i}\frac{\pi}{2}Q^{\mathrm{M}}}, \qquad S_4 = \mathrm{e}^{\mathrm{i}\pi Q^{\mathrm{W}}} \mathrm{e}^{\mathrm{i}\frac{\pi}{2}Q^{\mathrm{M}}} \mathrm{e}^{\mathrm{i}\pi Q^{\mathrm{W}}}. \tag{156}$$

The operators $S_n$ with $n < 0$ are related to those with $n > 0$ by $S_{-|n|} = \mathrm{e}^{\mathrm{i}\frac{\pi}{2}Q^{\mathrm{M}}} S_{|n|}$. These $S_n$ operators implement the pivoting procedure for the Onsager algebra [120, 121].

The Onsager charges in the form (154) are useful because they are directly related to generators $Q^{\mathrm{M}}$ and $Q^{\mathrm{W}}$ of the Onsager algebra (145). Firstly, since they are unitarily related to the quantized $Q^{\mathrm{M}}$ and $Q^{\mathrm{W}}$, they have quantized eigenvalues. Secondly, since they are constructed from only $Q^{\mathrm{M}}$ and $Q^{\mathrm{W}}$, which commute with the XX model Hamiltonian, it is clear that each $Q_n$ also commutes with the XX model Hamiltonian. Furthermore, in this form, it is clear what the Onsager charges in the T-dual model (48) become: conjugating $Q_n$ by $U_{\mathrm{T}}^{-1}$ amounts to replacing each $Q^{\mathrm{M}}$ with $\widehat{Q}^{\mathrm{W}}$ and each $Q^{\mathrm{W}}$ with $\widehat{Q}^{\mathrm{M}}$. Last but not least, the expression (154) for $Q_n$ makes it clear what $Q_n$ flows to in the IR limit — the compact free boson. Indeed, since $Q^{\mathrm{M}}$ and $Q^{\mathrm{W}}$ become the commuting operators $\mathcal{Q}^{\mathcal{M}}$ and $\mathcal{Q}^{\mathcal{W}}$, the IR limit of $S_n$ will commute with $\mathcal{Q}^{\mathcal{M}}$ and $\mathcal{Q}^{\mathcal{W}}$, too. Therefore, $Q_n$ in the IR becomes

$$Q_n \xrightarrow{\text{IR limit}} \begin{cases} 2\mathcal{Q}^{\mathcal{W}}, & n \text{ odd}, \\ \mathcal{Q}^{\mathcal{M}}, & n \text{ even}. \end{cases} \tag{157}$$

## 5.2 Symmetry actions on the Onsager charges

In deriving the expression (154) for the Onsager charges $Q_n$, we constructed the operators $\mathsf{D}_a$ and $\mathsf{D}_b$ that have a nontrivial action (148) on $Q_n$. From Eq. (150), $\mathsf{D}_a$ and $\mathsf{D}_b$ are non-invertible symmetry operators of the XX model. They are particular instances of the family of operators $\mathsf{D}_{\pm,\phi,\theta}$ (75), given by $\mathsf{D}_a \equiv \mathsf{D}_{+,\pi/2,0}$ and $\mathsf{D}_b \equiv \mathsf{D}_{+,0,\pi}$, and satisfying

$$\mathsf{D}_a \mathsf{D}_b = \mathsf{D}_b \mathsf{D}_a = \mathsf{D}\mathsf{D}, \qquad T\mathsf{D}_a T^{-1} = \mathsf{D}_b, \qquad T\mathsf{D}_b T^{-1} = \mathsf{D}_a. \tag{158}$$

Given that $\mathsf{D}_a$ and $\mathsf{D}_b$ have interesting actions on $Q_n$, it is natural to wonder whether other symmetry operators of the XX model act nontrivially on $Q_n$ as well. In what follows, we will deduce the action of the translation operator $T$ and $\mathsf{D}$ on $Q_n$. We summarize these actions in Fig. 3.

Let us first discuss the action of $T$ on $Q_n$. To do so, we first recall that $T$ commutes with $Q^{\mathrm{M}}$ but not $Q^{\mathrm{W}}$. In fact, from Eq. (64), $T$ acts on $Q^{\mathrm{W}}$ the same as $\mathrm{e}^{\mathrm{i}\frac{\pi}{2}Q^{\mathrm{M}}}$. Using this, we can rewrite the action of $T$ on $S_n$ as

$$T S_n T^{-1} = \mathrm{e}^{\mathrm{i}\frac{\pi}{2}Q^{\mathrm{M}}} S_n \mathrm{e}^{-\mathrm{i}\frac{\pi}{2}Q^{\mathrm{M}}}. \tag{159}$$

With this relationship, the action of $T$ on $Q_n$ is straightforward to deduce using $S_{-|n|} = \mathrm{e}^{\mathrm{i}\frac{\pi}{2}Q^{\mathrm{M}}} S_{|n|}$, and we find that

$$T Q_n T^{-1} = \mathrm{e}^{\mathrm{i}\frac{\pi}{2}Q^{\mathrm{M}}} Q_n \mathrm{e}^{-\mathrm{i}\frac{\pi}{2}Q^{\mathrm{M}}} = Q_{-n}. \tag{160}$$

Furthermore, because $[Q_n, Q_m] = \mathrm{i}G_{m-n}$, translation acts on $G_n$ as

$$T G_n T^{-1} = G_{-n} = -G_n. \tag{161}$$

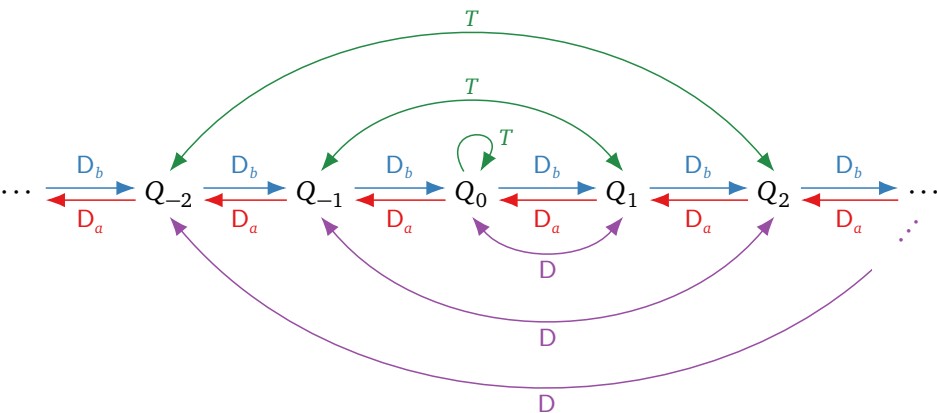

Figure 3: The symmetry operators $T$, $D_a$, $D_b$, and D of the XX model act nontrivially on the conserved Onsager charges $Q_n$. The color-coded arrows in this diagram describe the action on $Q_n$ by the symmetry operators labeling them.

Therefore, the Onsager charges $Q_n$ for $n \neq 0$ are all modulated operators. Since $e^{i\pi(\mathcal{Q}^\mathcal{M}+\mathcal{Q}^\mathcal{W})}$ arises from $T$ in the IR limit, and $Q_n$ and $Q_{-n}$ both flow to either $\mathcal{Q}^\mathcal{M}$ or $\mathcal{Q}^\mathcal{W}$, Eq. (160) in the IR becomes trivially satisfied because $\mathcal{Q}^\mathcal{M}$ and $\mathcal{Q}^\mathcal{W}$ commute.

Having found how $D_a$, $D_b$, and $T$ act on $Q_n$, we can next deduce how D acts on $Q_n$. In particular, using that $D_a = e^{i\frac{\pi}{2}Q^\mathrm{M}}D = e^{-i\frac{\pi}{2}Q^\mathrm{M}}D$ and plugging this into $D_a Q_n = Q_{n-1} D_a$, we find that

$$DQ_n = \left( e^{i\frac{\pi}{2}Q^\mathrm{M}} Q_{n-1} e^{-i\frac{\pi}{2}Q^\mathrm{M}} \right) D. \tag{162}$$

However, we can further simplify this using (160), which says that $e^{i\frac{\pi}{2}Q^\mathrm{M}}$ acts on $Q_n$ by changing the sign of $n$. Therefore, we find that

$$DQ_n = Q_{1-n}D, \tag{163}$$

which implies that

$$DG_n = G_{-n}D = -G_nD. \tag{164}$$

The existence of a non-invertible transformation sending $Q_n \to Q_{1-n}$ was also discussed in Ref. 121. Here, in the context of symmetries of the XX model, we find that D implements said transformation, and that the non-invertible symmetry formed by D and the Onsager algebra formed by $Q_n$ have a nontrivial interplay from D acting on $Q_n$. Furthermore, we see that Eq. (73) is a particular instance of (163), realizing the special cases in which $n = 0$ and $n = 1$. In the IR limit, (163) always flows to the continuum equation (34) since if $Q_n$ becomes $\mathcal{Q}^\mathcal{M}$ $(2\mathcal{Q}^\mathcal{W})$, then $Q_{1-n}$ will become $2\mathcal{Q}^\mathcal{W}$ $(\mathcal{Q}^\mathcal{M})$.

## 6 T-duality in other spin chains

In Section 3, we identified lattice T-duality in the XX model. In this section, we will discuss the fate of lattice T-duality upon deforming the XX model Hamiltonian and explore the phase diagrams of the resulting spin models.

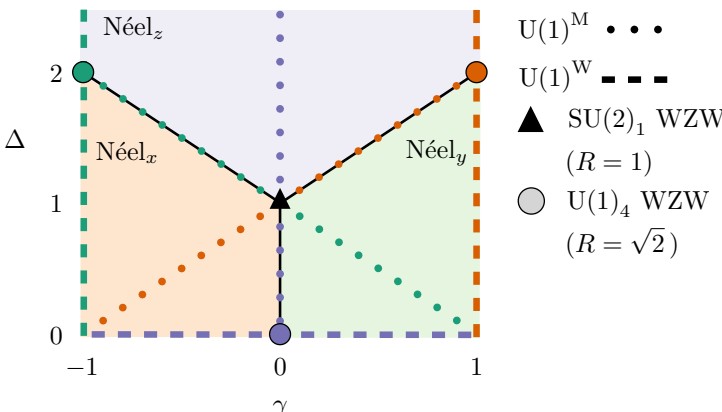

Figure 4: The XYZ model (165) with $-1 \leq \gamma \leq 1$ and $\Delta > 0$ as three distinct phases, all of which are gapped, Néel ordered phases with two ground states. We color these three phases orange, green, and purple, respectively. The black solid lines separating them denote phase transitions, which are described by the compact free boson CFT at various radii $1 \leq R \leq \sqrt{2}$. The three $R = \sqrt{2}$ points are described by three unitarily related XX models, all of which have their own lattice momentum and winding U(1) symmetries. These symmetries do not generally persist away from the XX model points, and we denote by dashed/dotted colored lines where such respective symmetries exist in the XYZ model.

## 6.1 The XYZ model

Let us first contextualize the discussion of lattice T-duality and corresponding symmetries to the XYZ model

$$H_{\mathrm{XYZ}}(\gamma, \Delta) = \sum_{j=1}^{L} \left( (1-\gamma) X_j X_{j+1} + (1+\gamma) Y_j Y_{j+1} + \Delta Z_j Z_{j+1} \right). \tag{165}$$

We will restrict our discussion of this model to parameters $-1 \leq \gamma \leq 1$ and $\Delta \geq 0$.

The phase diagram of the XYZ model has been studied both numerically and analytically [122–124], which we show in Fig. 4. The Hamiltonian is translation invariant and has the $\mathbb{Z}_2^{\mathrm{M}} \times \mathbb{Z}_2^{\mathrm{C}}$ symmetry generated by $\eta = \prod_{j=1}^{L} (-1)^j Z_j$ and $C = \prod_{j=1}^{L} X_j$. It has three gapped phases, all of which have two ground states that spontaneously break $\mathbb{Z}_2^{\mathrm{M}} \times \mathbb{Z}_2^{\mathrm{C}} \times \mathbb{Z}_L$. We denote these three phases as Néel$_x$, Néel$_y$, and Néel$_z$ phases. The spontaneous symmetry-breaking patterns associated with them are

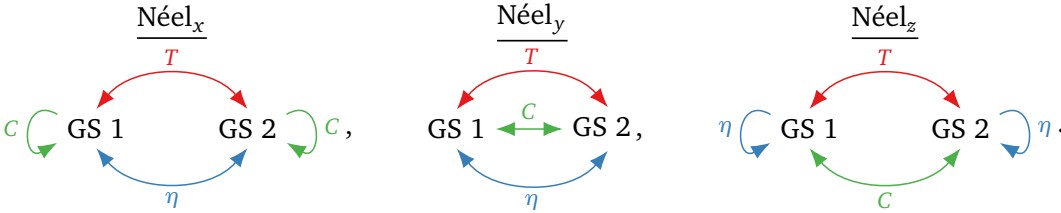

As their name suggests, these are all Néel antiferromagnet phases since the spin rotation symmetry $\mathbb{Z}_2^{\mathrm{M}} \times \mathbb{Z}_2^{\mathrm{C}}$ and lattice translations are spontaneously broken such that each spin rotation symmetry can be composed with $T$ to act trivially on the ground states. Consequently, the coarse-grained magnetization vanishes (*i.e.*, it is an antiferromagnet). We use the notation that the subscript on Néel describes the trivial spin flip symmetry without using translations

(*i.e.*, Néel$_y$ means that $\prod_{j=1}^{L} Y_j \equiv C\,\eta$ acts trivially on the ground states). The transitions between these various Néel phases are deconfined quantum critical points [125] described by the compact free boson CFT.

When $\gamma = \Delta = 0$, the XYZ Hamiltonian reduces to the XX Hamiltonian (36). Furthermore, it is unitarily equivalent to the XX model when $(\gamma, \Delta) = (-1, 2)$ and $(\gamma, \Delta) = (1, 2)$. Therefore, there are three sets of momentum and winding U(1) symmetries respective to these three XX model points. These XX model points and their symmetries are related to one another by the unitary operators

$$\mathsf{H}_{zx} = \prod_{j=1}^{L}\left(\frac{Z_j + X_j}{\sqrt{2}}\right), \qquad \mathsf{H}_{yz} = \prod_{j=1}^{L}\left(\frac{Y_j + Z_j}{\sqrt{2}}\right), \qquad \mathsf{H}_{xy} = \prod_{j=1}^{L}\left(\frac{X_j + Y_j}{\sqrt{2}}\right). \tag{166}$$

In fact, these unitaries relate the XYZ model Hamiltonians (up to a multiplicative constant) at different parameters. For example, using that $\mathsf{H}_{yz}$ acts on the Pauli operators by exchanging each $Y_j$ and $Z_j$, we find that

$$\mathsf{H}_{zx}\, H_{\mathrm{XYZ}}(\gamma, \Delta)\, \mathsf{H}_{zx}^{-1} = \frac{1 + \Delta + \gamma}{2}\, H_{\mathrm{XYZ}}\left(\frac{1 - \Delta + \gamma}{1 + \Delta + \gamma}, \frac{2 - 2\gamma}{1 + \Delta + \gamma}\right). \tag{167}$$

The Hamiltonian is invariant under this transformation when $\Delta = 1 - \gamma$. Similar expressions hold for $\mathsf{H}_{yz}$ and $\mathsf{H}_{xy}$:

$$\mathsf{H}_{yz}\, H_{\mathrm{XYZ}}(\gamma, \Delta)\, \mathsf{H}_{yz}^{-1} = \frac{1 + \Delta - \gamma}{2}\, H_{\mathrm{XYZ}}\left(\frac{-1 + \Delta + \gamma}{1 + \Delta - \gamma}, \frac{2 + 2\gamma}{1 + \Delta - \gamma}\right), \tag{168}$$

$$\mathsf{H}_{xy}\, H_{\mathrm{XYZ}}(\gamma, \Delta)\, \mathsf{H}_{xy}^{-1} = H_{\mathrm{XYZ}}(-\gamma, \Delta), \tag{169}$$

the invariant lines of which are $\Delta = 1 + \gamma$ and $\gamma = 0$, respectively. These invariant lines coincide with the critical lines of the model, and the above unitaries also relate the three Néel phases and $c = 1$ phase transitions to one another.

Perturbing away from an XX model point in the XYZ model explicitly breaks most of the symmetries discussed in Section 3. Consequently, without both momentum and winding symmetries simultaneously present, there fails to be a lattice T-duality. For example, the XYZ Hamiltonian with $\Delta = 0$ becomes the XY model [53]

$$H_{\mathrm{XY}}(\gamma) = \sum_{j=1}^{L}\left((1 - \gamma)X_j X_{j+1} + (1 + \gamma)Y_j Y_{j+1}\right). \tag{170}$$

When $\gamma \neq 0$, the U(1) momentum symmetry (38) is explicitly broken and there is no lattice T-duality. In fact, this must be the case in order for the phase diagram to be consistent with Section 4.4: any Hamiltonian with both the U(1) momentum and winding symmetries is gapless, but the XY model at $\gamma \neq 0$ is gapped. The winding symmetry, on the other hand, is preserved for all $\gamma$. Therefore, using the unitaries (166), there are also winding symmetries at $\gamma = \pm 1$ corresponding to XX model points at $(\gamma, \Delta) = (\pm 1, 2)$, as depicted by the dashed lines in Fig. 4. However, away from their critical points—their XX model points—these winding symmetries do not play an important role in understanding the phase diagram.

Similarly, the XYZ Hamiltonian when $\gamma = 0$ becomes the the XXZ chain

$$H_{\mathrm{XXZ}}(\Delta) = \sum_{j=1}^{L}(X_j X_{j+1} + Y_j Y_{j+1} + \Delta\, Z_j Z_{j+1}). \tag{171}$$

When $\Delta \neq 0$, this Hamiltonian fails to preserve with the U(1) winding symmetry (59). Therefore, without the winding symmetry, there is no lattice T-duality. Again, this must be the case

for the phase diagram to be consistent with Section 4.4 since the XXZ Hamiltonian is gapped for $\Delta > 1$, undergoing a Berezinskii–Kosterlitz–Thouless (BKT) transition at $\Delta = 1$ [126, 127].

The XXZ Hamiltonian, however, still has the U(1) momentum symmetry. Using the unitaries (166), there two other U(1) momentum symmetries at $\Delta = 1 \pm \gamma$ corresponding to the XX model points at $(\gamma, \Delta) = (\pm 1, 2)$. This U(1) symmetry play an important role in the phase diagram: together with lattice translations [22], they protect the gapless phase of the XXZ model. For instance, in the XXZ model (171), it is in a gapless phase for $-1 < \Delta \leq 1$ and flows to the compact free boson at radius [55, 56]

$$R_{\text{XXZ}}(\Delta) = \sqrt{\frac{\pi}{\pi - \arccos(\Delta)}} \,. \tag{172}$$

Even though the deformation $Z_j Z_{j+1}$ explicitly breaks the lattice winding symmetry, there is an emergent U(1) winding symmetry in the continuum limit for this range of $\Delta$. Indeed, the continuum counterpart of the $Z_j Z_{j+1}$ deformation is an exactly marginal current-current deformation of the compact boson CFT, which changes the radius $R$. This deformation, unlike its lattice counterpart, preserves both the continuum momentum and winding U(1) symmetries.

The three U(1) momentum symmetries along $\Delta = 0, 1 \pm \gamma$ all coexist at the BKT transition point $(\gamma, \Delta) = (0, 1)$. At this point, the three U(1) momentum symmetries are embedded into an enlarged SO(3) spin rotation symmetry and the XYZ chain becomes the XXX chain, which is described in the IR by the compact free boson at radius $R = 1$. This is the $\text{SU}(2)_1 = \text{U}(1)_2$ WZW CFT, which is the WZW model whose associated 2 + 1D Chern-Simons theory is $\text{SU}(2)_1 = \text{U}(1)_2$.

## 6.2 U(1)$^{\text{M}}$ and U(1)$^{\text{W}}$ symmetric spin chains

In the XYZ chain (165), the XX model points were the only points in parameter space with both momentum and winding U(1) symmetries and the related lattice T-duality. While the XYZ deformations of the XX model could break the momentum and winding symmetries, alternative symmetric deformations exist. Here, we will discuss the general class of qubit chains with both the momentum and winding U(1) symmetries.

We can construct the Hamiltonian of such a U(1)$^{\text{M}}$ and U(1)$^{\text{W}}$ symmetric model by bosonizing the fermionic model (137). The model (137) is the most general fermion model commuting with the vector and axial charges $Q^{\text{V}}$ and $Q^{\text{A}}$, which are defined by (127). Under bosonization, these conserved charges become the momentum and winding charges $Q^{\text{M}}$ (37) and $Q^{\text{W}}$ (59). Therefore, the image of the fermionic model (137) under bosonization will yield the most general qubit model commuting with $Q^{\text{M}}$ and $Q^{\text{W}}$. This qubit Hamiltonian is[30]

$$H_{\text{M,W}}(g_n) = \sum_{j=1}^{L} \sum_{n=1}^{N} g_n H_j^{(n)}, \qquad H_j^{(n)} = \begin{cases} \left(X_j X_{j+n} + Y_j Y_{j+n}\right) \prod_{k=1}^{n-1} Z_{j+k}, & n \text{ odd}, \\ \left(Y_j X_{j+n} - X_j Y_{j+n}\right) \prod_{k=1}^{n-1} Z_{j+k}, & n \text{ even}. \end{cases} \tag{173}$$

By construction, $H_{\text{M,W}}(g_n)$ commutes with $Q^{\text{M}}$ and $Q^{\text{W}}$, which can be checked explicitly. However, it also commutes with the non-invertible symmetry operator D (71). Indeed, its fermionized counterpart (137) commutes with $\text{e}^{-\text{i}\frac{\pi}{2}Q^{\text{V}}} T_a$, and under bosonization this becomes $\text{e}^{-\text{i}\frac{\pi}{2}Q^{\text{M}}} D_a = \text{D}$. Because D commutes with (173), any Hamiltonian commuting with both $Q^{\text{M}}$ and $Q^{\text{W}}$ has the non-invertible symmetry formed by D. Therefore, any $Q^{\text{M}}$ and $Q^{\text{W}}$ symmetric Hamiltonian has lattice T-duality that exchanges the momentum and winding charges.

---

[30]The parameters $g_n$ in the bosonic Hamiltonian (173) are the same as those in the fermionic model (137) up to possible multiplicative signs. In particular, they are related by $g_{2\ell}^{\text{bosonic}} = (-1)^{\ell+1} g_{2\ell}^{\text{fermionic}}$ and $g_{2\ell+1}^{\text{bosonic}} = (-1)^{\ell} g_{2\ell+1}^{\text{fermionic}}$.

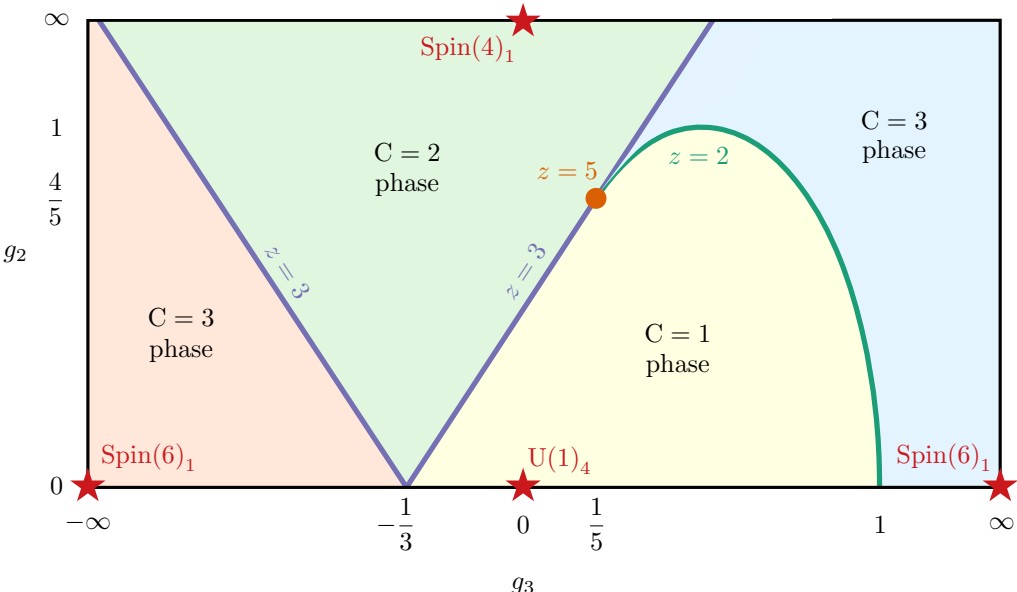

Figure 5: Phase diagram of the quantum spin Hamiltonian (174), which is gapless for all values of $g_2$ and $g_3$. The gapless phases are incommensurate and labeled by the number of bosonized Dirac fermions C. Furthermore, the dark red stars denote points in the phase diagram for which the IR of (174) is described by a CFT. At these CFT points, C corresponds to the central charge. The phase transitions are labeled by their dynamical critical exponent $z$, all of which have $z > 1$. The $z = 2$ critical line shown in green occurs at $g_2 = \sqrt{1 - (2g_3 - 1)^2}$ for $g_3 > 1/5$. The $z = 3$ line is shown in purple and occurs at $g_2 = \frac{1}{2}|3g_3 + 1|$. These two critical lines intersect at a multi-critical point with $z = 5$ at $(g_2, g_3) = (\frac{4}{5}, \frac{1}{5})$.

Because the Hamiltonian (173) commutes with both $Q^{\mathrm{M}}$ and $Q^{\mathrm{W}}$, it is gapless for all $g_n$ (see Section 4.4). When $g_n = \delta_{n,k}$ for some positive integer $k$, the IR is described by the $\mathrm{Spin}(2k)_1$ WZW CFT. This is the diagonal, bosonic WZW model based on the Lie algebra $\mathfrak{so}(2k)$ whose associated $2 + 1$D Chern-Simons theory is $\mathrm{Spin}(2k)_1$. For example, when $k = 1$, this is the $\mathrm{Spin}(2)_1 = \mathrm{U}(1)_4$ WZW model, which is the $R = \sqrt{2}$ compact free boson. To see why this CFT describes the IR for general $k$, we use that the IR of the fermionized model (137) with $g_n = \delta_{n,k}$ is described by the $\mathrm{Dirac}^k$ CFT—$k$ decoupled copies of the Dirac CFT. This is because fermions on different sites $j \bmod k$ are decoupled from one another, and there is a free, massless Dirac fermion in the IR for each $j \bmod k$. It then follows that the $\mathrm{Spin}(2k)_1$ WZW CFT describes the IR of the bosonic model since the $\mathrm{Dirac}^k$ CFT under bosonization becomes the $\mathrm{Spin}(2k)_1$ WZW CFT [128]. It would be interesting to understand how the Onsager algebra maps into the IR current algebra at these $\mathrm{Spin}(2k)_1$ points, but we leave this for future work. When $g_n \neq \delta_{n,k}$, the Hamiltonian (173) remains gapless, but its IR is generally described by a non-relativistic quantum field theory, not a CFT. At the $\mathrm{Spin}(2k)_1$ point $g_n = \delta_{n,k}$, turning on $g_{n \neq k}$ corresponds in the fermionized IR theory to changing the relative velocities of left and right-moving fermions. This is a marginal, but nonzero conformal spin, non-relativistic deformation of the $\mathrm{Spin}(2k)_1$ WZW model.

Let us now specialize (173) to the case where $N = 3$. In this case, the Hamiltonian becomes

$$H(g_2, g_3) = \sum_{j=1}^{L} \Big( X_j X_{j+1} + Y_j Y_{j+1} + g_2 \left( Y_{j-1} Z_j X_{j+1} - X_{j-1} Z_j Y_{j+1} \right)$$

$$+ g_3 \left( X_{j-2} Z_{j-1} Z_j X_{j+1} + Y_{j-2} Z_{j-1} Z_j Y_{j+1} \right) \Big), \qquad (174)$$

where we have set $g_1 = 1$. The $g_2$ term is the simplest deformation of the XX model that preserves both the momentum and winding U(1) symmetries on the lattice. Using the fermionized Hamiltonian (137) with $N = 3$, we can exactly solve for the spectrum of (174).

We can deduce the phase diagram of $H(g_2, g_3)$ using the single-particle dispersion of the fermionized model, which is given by

$$\frac{1}{4}\,\omega_k = \sin\left(\frac{2\pi k}{L}\right) + g_2 \sin\left(\frac{4\pi k}{L}\right) + g_3 \sin\left(\frac{6\pi k}{L}\right). \tag{175}$$

Because $H(g_2, g_3)$ is unitarily related to $H(-g_2, g_3)$ by $X_j \leftrightarrow Y_j$, we consider only positive values of $g_2$. Then, using this single-particle dispersion, we find the phase diagram shown in Fig. 5. At the points $(g_2, g_3) = (0, 0)$, $(0, \infty)$, and $(\pm\infty, 0)$ in the phase diagram, the IR of the Hamiltonian is described by the $\mathrm{U}(1)_4$, $\mathrm{Spin}(4)_1$, and $\mathrm{Spin}(6)_1$ WZW CFTs, respectively. Away from these special CFT points, however, the IR of its gapless phases is described by a non-relativistic quantum field theory. Furthermore, these gapless phases are incommensurate gapless phases in the sense of Ref. 78. Namely, there are points in the phases where the many-body spectrum is gapless in a dense subset of the Brillouin zone, which follows from the zeros of $\omega_k$ occurring at irrational values of $\frac{2\pi k}{L}$ mod $2\pi$. The fact that the phase transitions between these incommensurate gapless phases have $z > 1$ agrees with a conjecture made in Ref. 78.

# 7 Outlook

In this paper, we have shown how dualities of quantum field theories can also arise in quantum lattice models of qubits. We focused on the T-duality of the compact boson CFT at radius $R = \sqrt{2}$ and its relationship to a duality of the XX spin chain. However, there is much more to reveal in understanding the general relationship of dualities in quantum field theories and dualities of quantum lattice models. In particular, what is the deeper relationship between unitary transformations on the lattice and field theory dualities in their IR limit? For one, it would be interesting to explore how the T-duality of the compact boson CFT at $R \neq \sqrt{2}$ can appear in quantum spin chains (especially at radii $R^2 \notin \mathbb{Z}_{>0}$). Related work towards this direction was carried out in [129]. Furthermore, exploring other dualities, such as other T-dualities and S-dualities, would be interesting. In particular, S-duality of U(1) gauge theory in $3 + 1$D, which, like T-duality, is related to a non-invertible symmetry [32]. There are known three-dimensional quantum spin models known to flow to U(1) gauge theory (*e.g.*, quantum spin ice models [130–132]) and it would be exciting to explore the possibility of them realizing a lattice S-duality.

Additionally, this paper considered spin chains described in the IR by CFTs with Abelian current algebras. Of course, there are various well-known CFTs with non-Abelian current algebras, such as $\mathfrak{su}(n)$ current algebras. It would be interesting to investigate a generalization of our results to such non-Abelian algebras and consider lattice counterparts of non-Abelian continuous symmetries and their anomalies.

## Acknowledgments

We would like to thank Ömer Aksoy, Nick Jones, Michael Levin, Greg Moore, Nathan Seiberg, Sahand Seifnashri, T. Senthil, Zhengyan Darius Shi, Xiao-Gang Wen, Yichen Xu, and Zeqi Zhang for interesting discussions. We further thank Pranay Gorantla, Nick Jones, Igor Klebanov, Sahand Seifnashri, Ryan Thorngren, and Yifan Wang for their comments on the draft.

**Funding information** SDP is supported by the National Science Foundation Graduate Research Fellowship under Grant No. 2141064. AC is supported by NSF DMR-2022428 and by the Simons Collaboration on Ultra-Quantum Matter, which is a grant from the Simons Foundation (651446, Wen). SHS is also supported by the Simons Collaboration on Ultra-Quantum Matter, which is a grant from the Simons Foundation (651444, SHS).

# A  Review of bosonization and fermionization in the continuum

In this appendix, we briefly review bosonization and fermionization in $1+1$D continuum field theory,[31] following recent discussions [136–142]. See Ref. 29 for a more detailed review. In the following discussion, we consider Euclidean continuum field theories defined on an arbitrary smooth orientable Riemannian 2-manifold $M$ with genus $g$ (*i.e.*, a Riemann surface). For the IR limit of the lattice models discussed in the main text, it is enough to consider $M$ to be the torus $T^2$. However, in this appendix, our treatment applies for arbitrary $M$. For simplicity, we assume there is no gravitational anomaly, in which case bosonization can be implemented by gauging the fermion number parity.[32]

## A.1  Fermionic field theories and the Arf invariant

Defining spinors in a fermionic field theory on an arbitrary Riemann surface $M$ requires a choice of spin structure on $M$. Abstractly, an oriented Riemannian $n$-manifold admits a spin structure if the transition group $\mathrm{SO}(n)$ of its oriented frame bundle can be lifted to $\mathrm{Spin}(n)$ while preserving the cocycle condition on each triple-overlap of coordinate charts. Every oriented $n < 4$-manifold admits a spin structure, and inequivalent lifts of $\mathrm{SO}(n)$ correspond to different spin structures. Furthermore, if a manifold admits a spin structure, it is called a spin manifold.

In practice, a spin structure on a genus $g \neq 0$ Riemann surface $M$ corresponds to a choice of signs picked up by a spinor when parallel transported around non-trivial 1-cycles of $M$. For general $M$, there are $2^{2g}$ such choices. For $M = T^2$, there are $2^2 = 4$ such choices, which correspond to the different choices of periodic and anti-periodic boundary conditions along the two canonical non-trivial 1-cycles. These boundary conditions are referred to as (NS,NS), (NS,R), (R,NS), and (R,R) in the string theory literature, where NS and R are anti-periodic (Neveu-Schwarz) and periodic (Ramond) boundary conditions, respectively.

---

[31]The terms bosonization and fermionization sometimes refer to an exact duality between two presentations of a QFT, one of which is expressed in terms of bosonic fields and the other in terms of fermionic fields [128, 133]. In this context, the QFT can be either fermionic or bosonic (i.e., it may or may not depend on a spin structure and have a fermion number parity symmetry). For fermionic QFTs, the spin structure that affects the spinor fields in the fermionic presentation arises after bosonization in topological terms of the bosonic presentations [134–136]. For bosonic QFTs, there is no dependence on a spin structure (by definition), and the fermion number parity in the fermion presentation is a gauge redundancy, not a global symmetry. In this paper, we refer to bosonization/fermionization as maps between bosonic and fermionic QFTs. Therefore, our usage of "bosonization" and "fermionization" does not refer to exact dualities. Instead, the bosonization and fermionization maps are implemented by summing over various background fields (e.g., bosonization is implemented by summing over spin structures).

[32]Bosonizing a $1+1$D fermionic QFT is generally implemented by summing over its spin structures, which requires its gravitational anomaly $n \in (I_{\mathbb{Z}}\Omega^{\mathrm{Spin}})^4(\mathrm{pt}) = \mathbb{Z}$ to be 0 mod 16. In this case, gauging fermion number parity causes the spin structures to be summed over and bosonizes the theory. When the gravitational anomaly is $n \neq 0$ mod 16, the spin structure cannot be summed over and the QFT cannot be bosonized. When $n = 8$ mod 16, fermion number parity can still be gauged, but doing so maps the fermionic QFT to a different fermionic QFT. For example, gauging $(-1)^F$ in a QFT with eight chiral fermions with $c_L = 4$ and $c_R = 0$ maps the fermionic QFT to another fermionic QFT. We refer the reader to [143, 144] for additional discussion.

Given a Riemann surface $M$ with spin structure $\rho$, there is a bordism, mod 2 invariant known as the Arf invariant. It depends on the spin structure $\rho$ and is characterized by the relation [145]

$$\mathrm{Arf}[\rho] = \begin{cases} 1, & \rho \text{ is odd}, \\ 0, & \rho \text{ is even}, \end{cases} \tag{A.1}$$

where $\rho$ is even/odd if the associated chiral Dirac operator $\slashed{D}_\rho$ has an even/odd number of zero-modes. Of the $2^{2g}$ allowed spin structures on $M$, $2^{g-1}(2^g+1)$ are even and the remaining $2^{g-1}(2^g-1)$ are odd. See, *e.g.*, Ref. [146] for a pedagogical discussion of the Arf invariant from a physics perspective. The Arf invariant plays a key role in the classification of invertible fermionic phases. In $1+1$D, there are two invertible fermionic phases whose low-energy partition functions differ by the invertible TFT $(-1)^{\mathrm{Arf}[\rho]}$ [147].

The difference between two spin structures is given by an element of $H^1(M, \mathbb{Z}_2)$. Importantly, there is no canonically trivial spin structure, so individual spin structures themselves are *not* classified by $H^1(M, \mathbb{Z}_2)$.[33] This is to be contrasted to the space of $\mathbb{Z}_2$ gauge fields on $M$, which is described by $H^1(M, \mathbb{Z}_2)$ and has a trivial gauge field. In fact, because both spin structures and $\mathbb{Z}_2$ gauge fields are related to $H^1(M, \mathbb{Z}_2)$, there is a natural action of $\mathbb{Z}_2$ gauge fields, $a \in H^1(M, \mathbb{Z}_2)$, on the spin structures, which we denote as $a + \rho$. Two important identities using this action, which we will make use of below, are[34]

$$\mathrm{Arf}[(a+b)+\rho] = \mathrm{Arf}[a+\rho] + \mathrm{Arf}[b+\rho] + \mathrm{Arf}[\rho] + \int_M a \cup b, \tag{A.2}$$

$$e^{i\pi\mathrm{Arf}[b+\rho]} = \frac{1}{2^g} \sum_{a \in H^1(M, \mathbb{Z}_2)} e^{i\pi(\mathrm{Arf}[a+\rho] + \mathrm{Arf}[\rho] + \int_M a \cup b)}, \tag{A.3}$$

where $a, b \in H^1(M, \mathbb{Z}_2)$ and $\cup$ denotes the cup product.[35]

## A.2 Bosonizing a fermionic field theory

Every fermionic field theory $\mathcal{F}$ has a $\mathbb{Z}_2^{\mathcal{F}}$ symmetry generated by the fermion parity operator $(-1)^F$. This symmetry flips the sign of every fermion field and distinguishes a bosonic local operator from a fermionic one. Therefore, activating a nontrivial background for $\mathbb{Z}_2^{\mathcal{F}}$ amounts to changing the spin structure. We denote the partition function of $\mathcal{F}$ with spin structure $\rho$ as $Z_{\mathcal{F}}[\rho]$.

Bosonization of $\mathcal{F}$ is implemented by gauging the $\mathbb{Z}_2^{\mathcal{F}}$ symmetry, which is performed by summing over different spin structures via making the background $\mathbb{Z}_2^{\mathcal{F}}$ gauge field dynamical. Doing so yields a bosonic field theory $\mathcal{B}$ whose partition function is

$$Z_{\mathcal{B}}[A] = \frac{1}{2^g} \sum_{b \in H^1(M, \mathbb{Z}_2^{\mathcal{F}})} Z_{\mathcal{F}}[b+\rho] \, e^{i\pi(\int b \cup A + \mathrm{Arf}[A+\rho] + \mathrm{Arf}[\rho])}. \tag{A.5}$$

---

[33]When Euclidean spacetime is $T^2$, people sometimes make a *non-canonical* choice of calling the spin structure associated with the (R,R) boundary condition the trivial spin structure. With respect to such a non-canonical choice, the remaining spin structures are then classified by $H^1(M, \mathbb{Z}_2)$. However, the general and most invariant perspective is that spin structures are classified by an $H^1(M, \mathbb{Z}_2)$-torsor, which has no unique trivial element.

[34]We will drop the subscript $M$ going forward to make the notation less cumbersome.

[35]The reader may be familiar with cup products via Poincaré duality. The cup product is related to the intersection number of 1-cycles via

$$\int_M a \cup b = \langle \gamma_a, \gamma_b \rangle. \tag{A.4}$$

Here, $\gamma_a, \gamma_b \in H_1(M, \mathbb{Z}_2)$ are the Poincaré dual 1-cycles associated with $a$ and $b$, respectively, and $\langle \gamma_a, \gamma_b \rangle$ is their intersection number modulo 2.

Here $A \in H^1(M, \mathbb{Z}_2)$ is the background gauge field for the dual $\mathbb{Z}_2$ global symmetry of $\mathcal{B}$. The factor $e^{i\pi(\mathrm{Arf}[A+\rho]+\mathrm{Arf}[\rho])}$ is designed such that the left-hand side does not depend on the choice of $\rho$, as it shouldn't for a bosonic field theory.

There is an alternative way to bosonize the fermionic theory $\mathcal{F}$. In this way, we first stacking the Arf invariant $e^{i\pi\mathrm{Arf}[b+\rho]}$ onto $\mathcal{F}$ before summing over $b$. This "twisted" gauging of the fermion parity symmetry produces a second bosonization map that produces the bosonic theory

$$
\begin{aligned}
Z_{\mathcal{B}^\vee}[A] &= \frac{1}{2^g} \sum_{b \in H^1(M, \mathbb{Z}_2^{\mathcal{F}})} Z_{\mathcal{F}}[b+\rho]\, e^{i\pi\left(\mathrm{Arf}[b+\rho]+\int b \cup A + \mathrm{Arf}[A+\rho]+\mathrm{Arf}[\rho]\right)} \\
&= \frac{1}{2^g} \sum_{b \in H^1(M, \mathbb{Z}_2^{\mathcal{F}})} Z_{\mathcal{F}}[b+\rho]\, e^{i\pi\mathrm{Arf}[(b+A)+\rho]}.
\end{aligned}
\tag{A.6}
$$

These two bosonization producers are related. Indeed. one can check that $\mathcal{B}$ and $\mathcal{B}^\vee$ are related by orbifolding,

$$
Z_{\mathcal{B}^\vee}[A] = \frac{1}{2^g} \sum_{a \in H^1(M, \mathbb{Z}_2)} Z_{\mathcal{B}}[a] e^{i\pi \int a \cup A}.
\tag{A.7}
$$

## A.3 Fermionizing a bosonic field theory

We now consider a bosonic field theory $\mathcal{B}$ with a non-anomalous $\mathbb{Z}_2^{\mathcal{B}}$ global symmetry. Since all Riemann surfaces are spin, we can equip spacetime with a spin structure $\rho$ and fermionize the bosonic theory by gauging $\mathbb{Z}_2^{\mathcal{B}}$ in a particular way involving this spin structure that gives rise to a dual fermion number parity symmetry.

One way to fermionize is to stack $\mathcal{B}$ with $(-1)^{\mathrm{Arf}[a+\rho]+\mathrm{Arf}[\rho]}$ before summing over $a$. This yields a fermionic theory $\mathcal{F}$ coupled to a background fermion parity background gauge field $B$ whose partition function is

$$
Z_{\mathcal{F}}[B+\rho] = \frac{1}{2^g} \sum_{a \in H^1(M, \mathbb{Z}_2^{\mathcal{B}})} Z_{\mathcal{B}}[a]\, e^{i\pi\left(\mathrm{Arf}[a+\rho]+\mathrm{Arf}[\rho]+\int a \cup B\right)}.
\tag{A.8}
$$

We refer to $\mathcal{F}$ as a fermionization of $\mathcal{B}$. It is straightforward to verify that bosonizing this fermionic theory using (A.5) returns the bosonic theory we started with.

Instead of fermionizing as above, one can first orbifold the $\mathbb{Z}_2^{\mathcal{B}}$ symmetry to obtain theory $\mathcal{B}^\vee$ and then perform the above fermionization map with respect to the dual $\mathbb{Z}_2^{\mathcal{B}^\vee}$ symmetry. This provides an alternative fermionized theory $\mathcal{F}^\vee$,

$$
\begin{aligned}
Z_{\mathcal{F}^\vee}[B+\rho] &= \frac{1}{2^g} \sum_{a^\vee \in H^1(M, \mathbb{Z}_2^{\mathcal{B}^\vee})} Z_{\mathcal{B}^\vee}[a^\vee]\, e^{i\pi\left(\mathrm{Arf}[a^\vee+\rho]+\mathrm{Arf}[\rho]+\int a^\vee \cup B\right)} \\
&= \frac{1}{2^{2g}} \sum_{a^\vee \in H^1(M, \mathbb{Z}_2^{\mathcal{B}^\vee})} \sum_{a \in H^1(M, \mathbb{Z}_2^{\mathcal{B}})} Z_{\mathcal{B}}[a]\, e^{i\pi\left(\mathrm{Arf}[a^\vee+\rho]+\mathrm{Arf}[\rho]+\int a^\vee \cup B + \int a \cup a^\vee\right)} \\
&= e^{i\pi\mathrm{Arf}[B+\rho]} Z_{\mathcal{F}}[B+\rho].
\end{aligned}
\tag{A.9}
$$

In going from the first line to the second, we used the definition of orbifolding (cf. (A.7)), and in going from the second line to the third, we used the identities (A.2) and (A.3).

From (A.9), we learn that the two possible ways to fermionize $\mathcal{B}$, leading to fermionic theories $\mathcal{F}$ and $\mathcal{F}^\vee$, differ by an invertible TFT $(-1)^{\mathrm{Arf}}$. There is no canonically preferred choice between these two ways to fermionize, but only a relative difference. In our lattice discussion in Appendix B, we will see a similar feature.

The fermionization and bosonization maps of $1+1$D continuum field theories discussed in Appendices A.2 and A.3 are summarized in Figure 6.

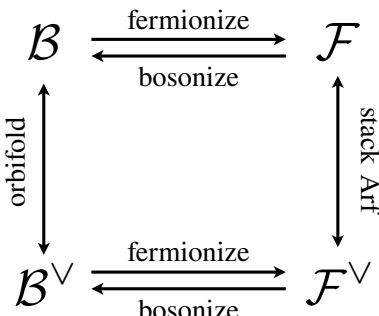

Figure 6: A summary of the different transformations relating bosonic and fermionic field theories discussed in Appendix A. The diagram commutes.

# B  Lattice bosonization and fermionization done globally

In this Appendix, we discuss how bosonization and fermionization of $1+1$D lattice Hamiltonians can be implemented within a unified gauging framework. In Appendix B.1, we illustrate this approach by bosonizing the Kitaev chain and fermionizing the transverse field Ising model. Afterwards, we apply the same approach to the XX model (36) and the two-Majorana chain (126) in Appendix B.2, the results of which are used in the main text. The reader is encouraged to refer to Refs. [70, 75, 87, 148, 149] for closely related discussions.

## B.1  Ising model $\longleftrightarrow$ Kitaev chain

In this appendix, we discuss a lattice perspective on the bosonization and fermionization maps discussed in Appendix A. In B.1.2 and B.1.4, we focus on developing a unified gauging language for the maps between fermionic and bosonic lattice models, closely paralleling the continuum treatment. We illustrate this approach using the examples of the Ising model and the Kitaev chain [150] for concreteness.

Let us define the transverse field Ising model on a periodic ring of qubits with $L$ sites,

$$H_{\text{TFIM}}(g) = -\sum_{j=1}^{L} \left( g^{-1} Z_j Z_{j+1} + g\, X_j \right). \tag{B.1}$$

Unlike in the rest of this paper, in this subsection we let $L$ be any positive integer, even or odd. The Hamiltonian has a $\mathbb{Z}_2$ symmetry generated by $\xi \equiv \prod_{j=1}^{L} X_j$. For $g < 1$, this symmetry is spontaneously broken evidenced by the doubly degenerate ground state in the thermodynamic limit. This is known as the ferromagnet phase. For $g > 1$, $\xi$ is unbroken with a unique ground state. This is known as the paramagnet phase.

Next, consider the Kitaev chain of $2L$ Majorana whose Hamiltonian is given by [150]:

$$H_{\text{Kitaev}}(t) = -\sum_{j=1}^{L} \left( \mathrm{i}\, t\, \chi_{2j-1} \chi_{2j} + \mathrm{i}\, t^{-1} \chi_{2j} \chi_{2j+1} \right). \tag{B.2}$$

The Majorana fermions $\chi_\ell$ (with $\ell = 1, 2, \ldots, 2L$) satisfy the anti-commutation algebra

$$\{\chi_\ell, \chi_{\ell'}\} = 2\delta_{\ell, \ell'}. \tag{B.3}$$

Below we consider both the periodic ($v = 0$) and anti-periodic ($v = 1$) boundary conditions on a closed chain:[36]

$$\chi_{\ell+2L} = (-1)^v \chi_\ell \,. \tag{B.5}$$

### B.1.1 Fermion parity

The Hamiltonian $H_{\text{Kitaev}}$ has a fermion parity symmetry, implemented by the operator

$$(-1)^F = \mathrm{i}^L \prod_{\ell=1}^{2L} \chi_\ell = \mathrm{i}^L \chi_1 \chi_2 \cdots \chi_{2L} \,, \tag{B.6}$$

where the factor of $\mathrm{i}^L$ ensures that the operator squares to 1. For $t \neq 1$, the ground state of this Hamiltonian is unique and gapped. The two regimes, $t < 1$ and $t > 1$ correspond to distinct invertible fermionic phases, differing in the continuum by the Arf invariant. On the lattice, we can choose a tensor product factorization of the full Hilbert space as

$$\mathcal{H} = \bigotimes_{j=1}^{L} \mathcal{H}_{2j-1,2j}^{\mathrm{f}} \,, \tag{B.7}$$

where $\mathcal{H}_{2j-1,2j}^{\mathrm{f}} \cong \mathbb{C}^2$ is the local Hilbert space acted on by $\chi_{2j-1}$ and $\chi_{2j}$. With this choice, one finds that the ground state in the $t \to \infty$ limit is an unentangled product state, while that in the $t \to 0^+$ limit has non-trivial entanglement. We could alternatively choose a tensor product factorization of the full Hilbert space as

$$\mathcal{H} = \bigotimes_{j=1}^{L} \mathcal{H}_{2j,2j+1}^{\mathrm{f}} \,, \tag{B.8}$$

where $\mathcal{H}_{2j,2j+1}^{\mathrm{f}} \cong \mathbb{C}^2$ is the local Hilbert space acted on by $\chi_{2j}$ and $\chi_{2j+1}$. With this second choice of tensor product factorization, we find that the ground state in the $t \to 0^+$ limit is now a product state while that in the $t \to \infty$ limit realizing an entangled state. We will make a non-canonical choice to call the $t < 1$ phase non-trivial in the following discussion, which amounts to choosing the tensor product factorization (B.7).

The Majorana translation operator $T_{\text{maj}}$, which acts as[37]

$$T_{\text{maj}} \chi_\ell T_{\text{maj}}^{-1} = \chi_{\ell+1} \,, \tag{B.9}$$

transforms the coupling constant $t$ of the Kitaev Hamiltonian as

$$T_{\text{maj}} H_{\text{Kitaev}}(t) T_{\text{maj}}^{-1} = H_{\text{Kitaev}}(t^{-1}) \,. \tag{B.10}$$

As a result, $T_{\text{maj}}$ interchanges the two phases of the Kitaev chain. The fermion parity $(-1)^F$ is transformed by $T_{\text{maj}}$ as [75, 151, 152]

$$T_{\text{maj}} (-1)^F T_{\text{maj}}^{-1} = \mathrm{i}^L \chi_2 \chi_3 \cdots \chi_{2L+1} = (-1)^{v+1} (-1)^F \,. \tag{B.11}$$

---

[36]Due to the (anti-)periodic boundary conditions (B.5), operators can depend on $v$ when written in terms of $\chi_\ell$ with $\ell = 1, 2, \ldots, 2L$. For example, using that $\chi_{2L+1} = (-1)^v \chi_1$, the explicit dependence on $v$ for the Kitaev Hamiltonian (B.2) is

$$-\sum_{j=1}^{L} \mathrm{i} t \chi_{2j-1} \chi_{2j} - \sum_{j=1}^{L-1} \mathrm{i} t^{-1} \chi_{2j} \chi_{2j+1} - (-1)^v \mathrm{i} t^{-1} \chi_{2L} \chi_1 \,. \tag{B.4}$$

To avoid making expressions too cumbersome, however, we will keep this dependence implicit.

[37]The Majorana translation operator $T_{\text{maj}}$ implicitly depends on $v$ such that $T_{\text{maj}} \chi_{2L} T_{\text{maj}}^{-1} = (-1)^v \chi_1$. See Ref. [75] for the explicit expressions for the Majorana translation operators under these two boundary conditions.

In the continuum limit, $T_{\text{maj}}$ implements a chiral fermion parity transformation that equivalently stacks the Arf invariant. At $t = 1$, this algebra was related to the 't Hooft anomaly of the chiral fermion parity of the continuum Majorana CFT in Ref. [75].

For anti-periodic boundary conditions ($\nu = 1$), the ground states in both the $t < 1$ and $t > 1$ have the same eigenvalue under $(-1)^F$. This follows from (B.11) since these two ground states are related by $T_{\text{maj}}$. In fact, we have fixed the normalization of $(-1)^F$ in (B.6) such that both eigenvalues are $+1$. However, for periodic boundary conditions ($\nu = 0$), the ground states of the two phases carry different $(-1)^F$ quantum numbers. Therefore, there is no natural way to determine the overall minus sign of $(-1)^F$ for all $t$ under periodic boundary conditions. The different responses to changing boundary conditions shows that these phases are distinct invertible phases, and their low-energy partition functions differ by an Arf invariant.

### B.1.2 Bosonizing by gauging

We will gauge this fermion parity symmetry of the Hamiltonian $H_{\text{Kitaev}}$ in two different ways to bosonize the Kitaev chain and find the transverse-field Ising model at two different $g$'s related by the Kramers-Wannier transformation.

To gauge the fermion parity symmetry, we first rewrite the symmetry operator (B.6) in a manifestly onsite manner. One way of doing so is

$$(-1)^F = \prod_{j=1}^{L} \mathrm{i}\,\chi_{2j-1}\chi_{2j}\,, \tag{B.12}$$

which is onsite with respect to the tensor product factorization (B.7). We will refer to the local Hilbert spaces $\mathcal{H}^{\text{f}}_{2j-1,2j}$ as site Hilbert spaces, with sites indexed by $j = 1, \ldots, L$. We introduce gauge field qubits on links; the qubit on link $\langle j, j+1 \rangle$ is acted on by the Pauli operators $X_{j,j+1}, Z_{j,j+1}$. We impose the Gauss law $G_j = 1$ for each site $j$, where the Gauss operators are defined as

$$G_j = Z_{j-1,j}\left(\mathrm{i}\,\chi_{2j-1}\chi_{2j}\right)Z_{j,j+1}\,. \tag{B.13}$$

These Gauss operators appropriately satisfy $\prod_{j=1}^{L} G_j = (-1)^F$. Next, we couple the Kitaev Hamiltonian (B.2) minimally to the gauge fields so that it commutes with each $G_j$:

$$H_{\text{mc}} = -\mathrm{i}\,t\sum_{j=1}^{L}\chi_{2j-1}\chi_{2j} - \mathrm{i}\,t^{-1}\sum_{j=1}^{L}\chi_{2j}X_{j,j+1}\,\chi_{2j+1}\,. \tag{B.14}$$

Then we perform the unitary transformation

$$\begin{aligned}
\chi_{2j-1} &\to Z_{j-1,j}\,\chi_{2j-1}\,, & \chi_{2j} &\to \chi_{2j}\,Z_{j,j+1}\,, \\
X_{j,j+1} &\to \mathrm{i}\,\chi_{2j}X_{j,j+1}\,\chi_{2j+1}\,, & Z_{j,j+1} &\to Z_{j,j+1}\,,
\end{aligned} \tag{B.15}$$

which transforms

$$G_j \to \mathrm{i}\,\chi_{2j-1}\chi_{2j}\,, \qquad H_{\text{mc}} \to -t\sum_{j=1}^{L}Z_{j-1,j}Z_{j,j+1}(\mathrm{i}\,\chi_{2j-1}\chi_{2j}) - t^{-1}\sum_{j=1}^{L}X_{j,j+1}\,. \tag{B.16}$$

In this unitary frame, we project to the Gauss law invariant sector by setting $\mathrm{i}\,\chi_{2j-1}\chi_{2j} = 1$, decoupling the fermions. Upon this projection and a shift of the qubits by a half-lattice translation $\langle j-1, j \rangle \to j$, we find the bosonized Hamiltonian,

$$H_{\text{b}} = -\sum_{j=1}^{L}\left(t\,Z_jZ_{j+1} + t^{-1}X_j\right) = H_{\text{TFIM}}(t^{-1})\,, \tag{B.17}$$

which is the transverse field Ising model (B.1) with coupling $g = t^{-1}$. The trivial ($t > 1$) and non-trivial ($t < 1$) phases of the Kitaev chain get mapped to the ferromagnet and paramagnet phases of the Ising model, respectively. This gauging procedure can be summarized by a gauging map, which acts on the generators of $(-1)^F$-even local operator algebra as follows:

$$i\chi_{2j-1}\chi_{2j} \to Z_j Z_{j+1}, \qquad i\chi_{2j}\chi_{2j+1} \to X_{j+1}. \tag{B.18}$$

Given that there is an alternate factorization of the Hilbert space (B.8), it is also natural to consider the onsite operator

$$\prod_{j=1}^{L} i\chi_{2j}\chi_{2j+1} = (-1)^{\nu+1}(-1)^F. \tag{B.19}$$

For anti-periodic boundary condition ($\nu = 1$), this gives the fermion parity operator, whereas for periodic boundary condition ($\nu = 0$), it differs from $(-1)^F$ by an overall minus sign. However, as we discussed before, there is no canonical way to determine the overall minus of the fermion parity operator under the periodic boundary condition.

This operator is obtained by applying $T_{\text{maj}}$ on the fermion parity operator (B.12):

$$T_{\text{maj}} (-1)^F T_{\text{maj}}^{-1} = \prod_{j=1}^{L} i\chi_{2j}\chi_{2j+1}. \tag{B.20}$$

This mirrors the continuum discussion in Appendix A. There, the second bosonization map was implemented in the continuum by stacking with the Arf invariant followed by gauging fermion parity. On the lattice, this stacking corresponds to acting with the entangler for the non-trivial Kitaev phase. This entangler is nothing but $T_{\text{maj}}$, which exchanges the two phases (see discussion around (B.10)).

For the second bosonization map, we therefore define the Gauss operators in line with the factorization in (B.19),

$$G_j^{\vee} = T_{\text{maj}} G_j T_{\text{maj}}^{-1} = Z_{j-1,j}\left(i\chi_{2j}\chi_{2j+1}\right)Z_{j,j+1}. \tag{B.21}$$

The minimally coupled Hamiltonian commuting with all $G_j$ is

$$H_{\text{mc}}^{\vee} = -i t \sum_{j=1}^{L} X_{j-1,j}\chi_{2j-1}\chi_{2j} - i t^{-1} \sum_{j=1}^{L} \chi_{2j}\chi_{2j+1}. \tag{B.22}$$

This time, we perform the unitary transformation

$$\begin{aligned}
\chi_{2j-1} &\to Z_{j-1,j}\,\chi_{2j-1}, & \chi_{2j} &\to Z_{j-1,j}\,\chi_{2j}, \\
X_{j-1,j} &\to X_{j-1,j}\,i\chi_{2j-1}\chi_{2j}, & Z_{j-1,j} &\to Z_{j-1,j},
\end{aligned} \tag{B.23}$$

which transforms

$$G_j^{\vee} \to i\chi_{2j}\chi_{2j+1}, \qquad H_{\text{mc}}^{\vee} \to -t \sum_{j=1}^{L} X_{j-1,j} - t^{-1} \sum_{j=1}^{L} Z_{j-1,j}Z_{j,j+1}(i\chi_{2j}\chi_{2j+1}). \tag{B.24}$$

In this unitary frame, we project to the Gauss law invariant sector by setting $i\chi_{2j}\chi_{2j+1} = 1$, decoupling the fermions. Upon this projection and a shift of the qubits by a half-lattice translation $\langle j-1, j\rangle \to j$, we find the bosonized Hamiltonian,

$$H_{\text{b}}^{\vee} = -\sum_{j=1}^{L}\left(t^{-1}Z_j Z_{j+1} + t X_j\right) = H_{\text{TFIM}}(t). \tag{B.25}$$

which is the transverse field Ising model (B.1) with coupling $g = t$. The trivial ($t > 1$) and non-trivial ($t < 1$) phases of the Kitaev chain get mapped to the paramagnet and ferromagnet phases of the Ising model, respectively. This gauging procedure can be summarized by a gauging map, whose action on the generators of the algebra of $(-1)^F$-symmetric local operators is given by

$$\mathrm{i}\chi_{2j-1}\chi_{2j} \to X_j, \qquad \mathrm{i}\chi_{2j}\chi_{2j+1} \to Z_j Z_{j+1}. \tag{B.26}$$

Let us note that $H_{\mathrm{b}}$ (B.17) and $H_{\mathrm{b}}^{\vee}$ (B.25) are related by orbifolding, *i.e.* gauging the $\mathbb{Z}_2$ symmetry generated by $\prod_j X_j$ (also known as Kramers-Wannier transformation), which parallels the continuum discussion.

### B.1.3 Jordan-Wigner transformation

Here we compare bosonization and the Jordan-Wigner transformation. We consider both periodic ($\nu = 0$) and anti-periodic ($\nu = 1$) boundary conditions for the Majorana fermions defining the Kitaev chain, as above.

Starting with the Majorana operators, we define the Pauli operators ($j = 1, \ldots, L$):

$$\sigma_j^z = \left(\prod_{k=1}^{j-1} \mathrm{i}\chi_{2k-1}\chi_{2k}\right)\chi_{2j-1}, \qquad \sigma_j^y = \left(\prod_{k=1}^{j-1} \mathrm{i}\chi_{2k-1}\chi_{2k}\right)\chi_{2j}, \tag{B.27}$$

which satisfy the usual Pauli operator algebra. This is known as the Jordan-Wigner transformation. From this, it follows that

$$\mathrm{i}\chi_{2j-1}\chi_{2j} = \sigma_j^x, \qquad \mathrm{i}\chi_{2j}\chi_{2j+1} = \begin{cases} \sigma_j^z \sigma_{j+1}^z, & j \neq L, \\ (-1)^{\nu+1}(\prod_{k=1}^L \sigma_k^x)\sigma_L^z \sigma_1^z, & j = L. \end{cases} \tag{B.28}$$

Using this transformation, we rewrite the Kitaev Hamiltonian (B.2) as:

$$H_{\mathrm{Kitaev}} = -t \sum_{j=1}^L \sigma_j^x - t^{-1} \sum_{j=1}^{L-1} \sigma_j^z \sigma_{j+1}^z - t^{-1}(-1)^{\nu+1}\left(\prod_{k=1}^L \sigma_k^x\right)\sigma_L^z \sigma_1^z. \tag{B.29}$$

Although expressed in terms of bosonic variables, this Hamiltonian is an exact rewriting of the Kitaev Hamiltonian: it has exactly the same spectrum and phase diagram. In terms of the spin variables, this Hamiltonian is non-local, because of the last term.

However this is different from the transverse field Ising model we obtained via gauging $(-1)^F$ in (B.17) or (B.25). In particular, (B.29) has a different spectrum and phase diagram from the transverse field Ising Hamiltonian on a closed periodic chain. While the structure of the phase diagram—in particular, the location of the critical point—is the same, the ground state degeneracies of the gapped phases as well as the conformal field theories describing the critical points are different.

The Jordan-Wigner transformation provides an alternative presentation for bosonization of the Kitaev chain. We focus on the first gauging while the discussion for the second one is similar. Written in terms of the Pauli variables, the Gauss operators (B.13) and the minimally coupled Hamiltonian (B.14) become

$$G_j = Z_{j-1,j}\,\sigma_j^x\, Z_{j,j+1},$$

$$H_{\mathrm{mc}} = -t \sum_{j=1}^L \sigma_j^x - t^{-1} \sum_{j=1}^{L-1} \sigma_j^z X_{j,j+1}\sigma_{j+1}^z - t^{-1}(-1)^{\nu+1}\left(\prod_{k=1}^L \sigma_k^x\right)\sigma_L^z X_{L,1}\sigma_1^z. \tag{B.30}$$

Gauss's law $G_j = 1$ in particular implies that $\prod_{k=1}^{L} \sigma_k^x = 1$. Furthermore, the sign $(-1)^{\nu+1}$ can be removed by a unitary transformation that flips the sign of $X_{L,1}$. Therefore, up to this unitary transformation, the minimally coupled Hamiltonian becomes

$$H_{\mathrm{mc}} = -t \sum_{j=1}^{L} \sigma_j^x - t^{-1} \sum_{j=1}^{L} \sigma_j^z X_{j,j+1} \sigma_{j+1}^z \,. \tag{B.31}$$

This is nothing but the Ising Hamiltonian $H_{\mathrm{TFIM}}(t)$ with its $\mathbb{Z}_2$ symmetry gauged. Performing the unitary transformation $\sigma_j^x \to Z_{j-1,j} \sigma_j^x Z_{j,j+1}$ and $X_{j,j+1} \to \sigma_j^z X_{j,j+1} \sigma_{j+1}^z$ causes the new $\sigma$ qubit in the $G_j = 1$ subspace to decouple and $H_{\mathrm{mc}}$ to become the Ising Hamiltonian at the opposite coupling, *i.e.*, $H_{\mathrm{TFIM}}(t^{-1})$. This gives the same result as in (B.17) from bosonization.

### B.1.4 Fermionizing by gauging

In this appendix, we will fermionize the Ising model (B.1), *i.e.*, gauge the $\mathbb{Z}_2$ symmetry $\xi$ by introducing fermionic degrees of freedom on links. The fermionic Hilbert space on the link $\langle j, j+1 \rangle$ is acted on by the Majorana operators $a_{j,j+1}, b_{j,j+1}$. We will impose periodic boundary conditions on the fermion degrees of freedom. To gauge $\xi$, we define the Gauss operators,

$$G_j = \mathrm{i}\, b_{j-1,j} X_j a_{j,j+1} \,. \tag{B.32}$$

Next, we minimally couple the fermions to (B.1) so that it commutes with all $G_j$,

$$H_{\mathrm{mc}} = -\sum_{j=1}^{L} \left( g^{-1} Z_j \mathrm{i}\, a_{j,j+1} b_{j,j+1} Z_{j+1} + g X_j \right) . \tag{B.33}$$

To simplify the Gauss operator, we do a change of basis using a unitary transformation that acts on the qubits and the fermions as

$$X_j \to \mathrm{i}\, b_{j-1,j} X_j a_{j,j+1} \,, \qquad Z_j \to Z_j \,, \qquad a_{j,j+1} \to Z_j a_{j,j+1} \,, \qquad b_{j,j+1} \to b_{j,j+1} Z_{j+1} \,. \tag{B.34}$$

This transforms the Gauss operator as $G_j \to X_j$, so that the Gauss law can be enforced by projecting to the $X_j = 1$ eigenspace, decoupling the qubits entirely. The unitary transformation also changes the minimally coupled Hamiltonian,

$$H_{\mathrm{mc}} \to -\sum_{j=1}^{L} \left( g^{-1} \mathrm{i}\, a_{j,j+1} b_{j,j+1} + g X_j \mathrm{i}\, b_{j-1,j} a_{j,j+1} \right) , \tag{B.35}$$

which upon projecting to the $X_j = 1$ subspace, followed by the renaming the Majoranas as

$$a_{j,j+1} = \chi_{2j-1} \,, \qquad b_{j,j+1} = \chi_{2j} \,. \tag{B.36}$$

With this renaming, (B.35) becomes the periodic Kitaev chain Hamiltonian (B.2) at $t = g^{-1}$,

$$H_{\mathrm{f}} = -\sum_{j=1}^{L} \left( \mathrm{i}\, g^{-1} \chi_{2j-1} \chi_{2j} + \mathrm{i}\, g\, \chi_{2j} \chi_{2j+1} \right) . \tag{B.37}$$

If we had chosen the Majorana fermions on the links to have anti-periodic boundary conditions, the fermionized Hamiltonian would be the Kitaev chain with anti-periodic boundary conditions.

The fermionization map transforms the $\mathbb{Z}_2$ symmetry of the qubits,

$$\xi = \prod_{j=1}^{L} X_j \to \prod_{j=1}^{L} \mathrm{i}\, \chi_{2j-2} \chi_{2j-1} = -(-1)^F \,, \tag{B.38}$$

and maps the ferromagnet ($g < 1$) and paramagnet ($g > 1$) phases of the Ising Hamiltonian to the trivial and non-trivial phases of the Kitaev chain, respectively. Furthermore, the action of this fermionization map on the algebra of $\xi$-symmetric local operators is given by

$$Z_j Z_{j+1} \to i\chi_{2j-1}\chi_{2j}, \qquad X_j \to i\chi_{2j-2}\chi_{2j-1}. \tag{B.39}$$

As in the continuum, we could also first gauge the $\mathbb{Z}_2$ symmetry $\xi$, before performing the fermionization map relative to the dual $\mathbb{Z}_2$ symmetry, to implement another bosonization map. Gauging $\xi$ amounts to implementing the Kramers-Wannier transformation on the $\xi$-symmetric local operators,

$$Z_j Z_{j+1} \to \widetilde{X}_{j+1}, \qquad X_j \to \widetilde{Z}_j \widetilde{Z}_{j+1}. \tag{B.40}$$

Furthermore, gauging $\xi$ trivializes the symmetry. However, the gauged model has a dual $\mathbb{Z}_2$ symmetry generated by $\widetilde{\xi} \equiv \prod_{j=1}^{L} \widetilde{X}_j$. Now, if we perform the fermionization map relative to the symmetry $\widetilde{\xi}$, the operator map (B.39) acts as

$$\widetilde{X}_{j+1} \to i\chi_{2j}\chi_{2j+1}, \qquad \widetilde{Z}_j \widetilde{Z}_{j+1} \to i\chi_{2j-1}\chi_{2j}. \tag{B.41}$$

Composing (B.40) and (B.41), we have the following transformation of the generators of the $\xi$-even local operator algebra:

$$Z_j Z_{j+1} \to i\chi_{2j}\chi_{2j+1}, \qquad X_j \to i\chi_{2j-1}\chi_{2j}, \tag{B.42}$$

hence the fermionized Hamiltonian is the Kitaev chain Hamiltonian (B.2) at coupling $t = g$,

$$H_f^\vee = -\sum_{j=1}^{L} \left( i g^{-1} \chi_{2j}\chi_{2j+1} + i g\, \chi_{2j-1}\chi_{2j} \right). \tag{B.43}$$

This fermionization map transforms the $\mathbb{Z}_2$ symmetry generator $\xi$ of the Ising Hamiltonian as

$$\xi = \prod_{j=1}^{L} X_j \to \prod_{j=1}^{L} i\chi_{2j-1}\chi_{2j} = (-1)^F, \tag{B.44}$$

while the $\mathbb{Z}_2$ symmetry-breaking ferromagnet ($g < 1$) and symmetric paramagnet ($g > 1$) phases are mapped to the non-trivial and trivial phases of the Kitaev chain, respectively. Note that the two fermionized Hamiltonians (B.37) and (B.43) are related by the unitary $T_{\text{maj}}$, which implements the lattice analog of stacking with the fermionic invertible TFT. This parallels the relationship between the two bosonized theories in the continuum, as summarized in (A.6).

## B.2 XX model $\longleftrightarrow$ two-Majorana chain

In this appendix, we follow the approach outlined in Appendix B.1 to bosonize the XX model Hamiltonian,

$$H_{\text{XX}} = \sum_{j=1}^{L} \left( X_j X_{j+1} + Y_j Y_{j+1} \right), \tag{B.45}$$

to the two-Majorana chain Hamiltonian,

$$H_{\text{2maj}} = -i \sum_{j=1}^{L} (a_j a_{j+1} + b_j b_{j+1}), \tag{B.46}$$

as well as fermionize $H_{\text{2maj}}$ to recover the XX model. As we carry out either step, we will keep track of how the symmetries of each theory are modified by the respective transformations.

We will assume $L$ is an even integer and that both (B.45) and (B.46) are defined with periodic boundary conditions, *i.e.*, $X_{j+L} = X_j, Y_{j+L} = Y_j$ for the former and $a_{j+L} = a_j$ and $b_{j+L} = b_j$ for the latter. Having already demonstrated multiple ways of fermionizing and bosonizing on the lattice and the continuum in preceding appendices, in this appendix we will make a non-canonical choice for the transformation in each direction.

### B.2.1 Bosonizing by gauging

Similar to the Kitaev chain (see Appendix B.1.2), bosonizing the two-Majorana chain model (B.46) on a periodic chain is implemented by gauging the fermion number parity symmetry,

$$(-1)^F = \prod_{j=1}^{L} \mathrm{i} a_j b_j\,. \tag{B.47}$$

We gauge this symmetry by introducing a qubit onto each link $\langle j, j+1 \rangle$, which is acted on by the Pauli operators $X_{j,j+1}$ and $Z_{j,j+1}$, and specifying a Gauss law that implements the gauging. Noting that $(-1)^F$ can be written as $(-1)^F = -\prod_{j=1}^{L} \mathrm{i} a_j b_{j+1}$, we enforce the Gauss law $G_j = 1$ where

$$G_j = \begin{cases} X_{j-1,j}\,(\mathrm{i} a_j b_{j+1})\,Y_{j,j+1}\,, & j \text{ odd,} \\ -Y_{j-1,j}\,(\mathrm{i} a_j b_{j+1})\,X_{j,j+1}\,, & j \text{ even.} \end{cases} \tag{B.48}$$

The fermion number parity operator satisfies $(-1)^F = (-1)^{L/2+1} \prod_{j=1}^{L} G_j$, so enforcing $G_j = 1$ projects the Hilbert space into the $(-1)^F = (-1)^{L/2+1}$ subspace—the $(-1)^F$ odd (even) subspace when $L = 0 \bmod 4$ ($L = 2 \bmod 4$). When $L = 0 \bmod 4$, this choice of gauging corresponds in the IR field theory to gauging $(-1)^F$ after stacking with the nontrivial Arf invariant.

Minimally coupling the new qubits, the two-Majorana chain model (B.46) becomes

$$H_{\mathrm{mc}} = -\mathrm{i} \sum_{j=1}^{L} \left( Z_{j,j+1}\, a_j a_{j+1} + Z_{j-1,j}\, b_j b_{j+1} \right), \tag{B.49}$$

and the vector and axial charges become

$$Q_{\mathrm{mc}}^{\mathrm{V}} = \frac{\mathrm{i}}{2} \sum_{j=1}^{L} Z_{j-1,j}\, a_j b_j\,, \qquad Q_{\mathrm{mc}}^{\mathrm{A}} = \frac{\mathrm{i}}{2} \sum_{j=1}^{L} a_j b_{j+1}\,. \tag{B.50}$$

We next perform a unitary transformation

$$a_j \to \begin{cases} -X_{j-1,j}\, a_j\,, & j \text{ odd,} \\ Y_{j-1,j}\, a_j\,, & j \text{ even,} \end{cases} \qquad b_j \to \begin{cases} -X_{j-1,j}\, b_j\,, & j \text{ odd,} \\ -Y_{j-1,j}\, b_j\,, & j \text{ even,} \end{cases}$$

$$X_{j-1,j} \to \begin{cases} X_{j-1,j}\,, & j \text{ odd,} \\ X_{j-1,j}\,(\mathrm{i} a_j b_j)\,, & j \text{ even,} \end{cases} \qquad Z_{j-1,j} \to (-1)^{j-1} Z_{j-1,j}\,(\mathrm{i} a_j b_j)\,. \tag{B.51}$$

In this unitary frame, the Gauss law becomes $G_j = \mathrm{i} a_j b_{j+1} = 1$, which decouples the original fermions from the system. Shifting the qubits from links to sites, the minimally coupled Hamiltonian (B.49) in the physical subspace of this unitary frame becomes the XX model (B.45). Similarly, the vector and axial charges become

$$(Q^{\mathrm{V}})^{\vee} = \frac{1}{2} \sum_{j=1}^{L} Z_j = Q^{\mathrm{M}}\,, \qquad (Q^{\mathrm{A}})^{\vee} = \frac{1}{2} \sum_{n=1}^{L/2} (X_{2n-1} Y_{2n} - Y_{2n} X_{2n+1}) = 2 Q^{\mathrm{W}}\,. \tag{B.52}$$

This is consistent with bosonizing the free Dirac fermion CFT [35, 139].

The above gauging procedure gives rise to a gauging map—a bosonization map—that relates the fermionic model (B.46) and its $(-1)^F$-even operators to the XX model and its $\eta$-even operators. The algebra of $(-1)^F$-even operators are generated by the fermion bilinears $\mathrm{i}a_j b_j$ and $\mathrm{i}a_j b_{j+1}$ for all sites $j$. Under the above gauging procedure, these operators are mapped to

$$\mathrm{i}a_j b_j \to Z_j, \qquad \mathrm{i}a_j b_{j+1} \to \begin{cases} X_j Y_{j+1}, & j \text{ odd}, \\ -Y_j X_{j+1}, & j \text{ even}. \end{cases} \tag{B.53}$$

Using that $\mathrm{i}a_j a_{j+1} = \mathrm{i}(\mathrm{i}a_j b_{j+1})(\mathrm{i}a_{j+1}b_{j+1})$ and $\mathrm{i}b_j b_{j+1} = \mathrm{i}(\mathrm{i}a_j b_j)(\mathrm{i}a_j b_{j+1})$, their image under this bosonization is

$$\mathrm{i}a_j a_{j+1} \to \begin{cases} -X_j X_{j+1}, & j \text{ odd}, \\ -Y_j Y_{j+1}, & j \text{ even}, \end{cases} \qquad \mathrm{i}b_j b_{j+1} \to \begin{cases} -Y_j Y_{j+1}, & j \text{ odd}, \\ -X_j X_{j+1}, & j \text{ even}. \end{cases} \tag{B.54}$$

### B.2.2 Jordan-Wigner transformation

In this appendix, we compare the bosonization procedure with a Jordan-Wigner transformation of the periodic two-Majorana chain model (B.46). We define the following Pauli operators in terms of the Majorana operators $a_j, b_j$ with $j = 1, \ldots, L$:

$$\sigma_j^x = \begin{cases} \left(\prod_{k=1}^{j-1}(-1)^k \mathrm{i}a_k b_k\right)b_j, & j \text{ odd}, \\ -\left(\prod_{k=1}^{j-1}(-1)^k \mathrm{i}a_k b_k\right)a_j, & j \text{ even}, \end{cases} \qquad \sigma_j^y = \begin{cases} \left(\prod_{k=1}^{j-1}(-1)^k \mathrm{i}a_k b_k\right)a_j, & j \text{ odd}, \\ \left(\prod_{k=1}^{j-1}(-1)^k \mathrm{i}a_k b_k\right)b_j, & j \text{ even}. \end{cases} \tag{B.55}$$

From this Jordan-Wigner transformation, it follows that for $j \neq L$,

$$-\mathrm{i}a_j a_{j+1} = \begin{cases} \sigma_j^x \sigma_{j+1}^x, & j \text{ odd}, \\ \sigma_j^y \sigma_{j+1}^y, & j \text{ even}, \end{cases} \qquad -\mathrm{i}b_j b_{j+1} = \begin{cases} \sigma_j^y \sigma_{j+1}^y, & j \text{ odd}, \\ \sigma_j^x \sigma_{j+1}^x, & j \text{ even}. \end{cases} \tag{B.56}$$

On the other hand, for $j = L$,

$$-\mathrm{i}a_L a_1 = -\left(\prod_{j=1}^{L}(-1)^j \sigma_j^z\right)\sigma_L^y \sigma_1^y, \qquad -\mathrm{i}b_L b_1 = -\left(\prod_{j=1}^{L}(-1)^j \sigma_j^z\right)\sigma_L^x \sigma_1^x, \tag{B.57}$$

Using (B.56) and (B.57), the two-Majorana chain Hamiltonian can be re-written as

$$H_{2\text{maj}} = \sum_{j=1}^{L-1}(\sigma_j^y \sigma_{j+1}^y + \sigma_j^x \sigma_{j+1}^x) - \left(\prod_{j=1}^{L}(-1)^j \sigma_j^z\right)(\sigma_L^x \sigma_1^x + \sigma_L^y \sigma_1^y). \tag{B.58}$$

Similar to the Jordan-Wigner transformation of the Kitaev chain, the above transformation makes the two-Majorana chain a non-local Hamiltonian in terms of the spin variables. We stress that, despite appearances, (B.58) is an exact re-writing of the two-Majorana chain model. While locally, it resembles the XX model Hamiltonian, it differs in important global aspects, including their global symmetries (cf. [39]).

### B.2.3 Fermionizing by gauging

Following the procedure outlined in Appendix B.1.4, we fermionize the XX model (B.45) on a periodic chain by gauging its $\mathbb{Z}_2^M$ symmetry using fermionic degrees of freedom on the links. The Hilbert space associated to link $\langle j, j+1 \rangle$ is acted on by the Majorana operators

$a_{j,j+1}, b_{j,j+1}$, on which we impose periodic boundary conditions. We gauge the $\mathbb{Z}_2^{\mathrm{M}}$ symmetry of $H_{\mathrm{XX}}$, generated by

$$\eta = \prod_{j=1}^{L}(-1)^j Z_j, \tag{B.59}$$

by defining the Gauss operators,

$$G_j = (-1)^j Z_j\, \mathrm{i} a_{j,j+1} b_{j,j+1}. \tag{B.60}$$

Minimal coupling turns the Hamiltonian $H_{\mathrm{XX}}$ into

$$H_{\mathrm{mc}} = \sum_{j=1}^{L}\left(\mathrm{i} X_j a_{j,j+1} X_{j+1} b_{j+1,j+2} + \mathrm{i} Y_j a_{j,j+1} Y_{j+1} b_{j+1,j+2}\right), \tag{B.61}$$

and the charges $Q^{\mathrm{M}}$ and $Q^{\mathrm{W}}$ into

$$Q_{\mathrm{mc}}^{\mathrm{M}} = \frac{1}{2}\sum_{j=1}^{L} Z_j,$$
$$Q_{\mathrm{mc}}^{\mathrm{W}} = \frac{1}{4}\sum_{n=1}^{L/2}\left(\mathrm{i} X_{2n-1} a_{2n-1,2n} Y_{2n} b_{2n,2n+1} - \mathrm{i} Y_{2n} b_{2n,2n+1} X_{2n+1} a_{2n+1,2n+2}\right). \tag{B.62}$$

Next, we implement a unitary transformation, defined by

$$Z_j \to Z_j\, \mathrm{i} a_{j,j+1} b_{j,j+1}, \qquad X_j \to \begin{cases} X_j, & j \text{ odd},\\ X_j\, \mathrm{i} a_{j,j+1} b_{j,j+1}, & j \text{ even}, \end{cases}$$
$$a_{j,j+1} \to \begin{cases} X_j\, a_{j,j+1}, & j \text{ odd},\\ Y_j\, a_{j,j+1}, & j \text{ even}, \end{cases} \qquad b_{j,j+1} \to \begin{cases} -X_j\, b_{j,j+1}, & j \text{ odd},\\ Y_j\, b_{j,j+1}, & j \text{ even}. \end{cases} \tag{B.63}$$

In this unitary frame, the Gauss laws become $Z_j = 1$, so a projection to the physical, Gauss law-invariant subspace effectively decouples the qubits. Performing a half-lattice translation $\langle j, j+1\rangle \to j$ of the fermions then turns $H_{\mathrm{mc}}$ into the two-Majorana chain (B.46). The momentum and winding charges, in turn, become

$$(Q^{\mathrm{M}})^{\vee} = \frac{1}{2}\sum_{j=1}^{L}\mathrm{i} a_j b_j = Q^{\mathrm{V}}, \qquad (Q^{\mathrm{W}})^{\vee} = \frac{1}{4}\sum_{j=1}^{L}\mathrm{i} a_j b_{j+1} = \frac{1}{2}Q^{\mathrm{A}}. \tag{B.64}$$

We note that this closely parallels (B.52), and is consistent with fermionizing the compact boson CFT at radius $R = \sqrt{2}$ [139].

The gauging procedure discussed above maps the $\eta$-even local operators of the XX model to the $(-1)^F$-even local operators of the two-Majorana chain, preserving their algebra. We refer to this as the fermionization map. This map transforms the $\mathbb{Z}_2^{\mathrm{M}}$ symmetry of the qubits as follows:

$$\eta = \prod_{j=1}^{L}(-1)^j Z_j \to \prod_{j=1}^{L}\mathrm{i} a_j b_j = (-1)^F. \tag{B.65}$$

Furthermore, the action of the fermionization map on the algebra of $\eta$-even local operators is given by the following transformation of its generators:

$$Z_j \to \mathrm{i} a_j b_j, \qquad X_j X_{j+1} \to \begin{cases} -\mathrm{i} a_j a_{j+1}, & j \text{ odd},\\ -\mathrm{i} b_j b_{j+1}, & j \text{ even}. \end{cases} \tag{B.66}$$

We note that this is precisely the inverse of the bosonization map (B.54) when restricted to the $(-1)^F$-even and $\eta$-even local operators of the fermionic and the bosonic models, respectively.

## C  Explicit expressions of the Onsager algebra generators

In this Appendix, we present explicit expressions for the Onsager charges $Q_n$ and $G_n$ discussed in Section 5 of the main text. We find these expressions using the fermionic Onsager charges $Q_n^{\mathrm{f}}$ and $G_n^{\mathrm{f}}$ discussed in Ref. 39, whose explicit expressions are given by Eq. (140). In particular, we apply the bosonization map (142) to $Q_n^{\mathrm{f}}$ and $G_n^{\mathrm{f}}$ in order to find $Q_n$ and $G_n$ in terms of the Pauli operators. Upon doing so, we find

$$
Q_n =
\begin{cases}
\frac{1}{2}\sum_{j=1}^{L} Z_j\,, & n=0\,,\\[2mm]
\frac{(-1)^{\frac{n+2}{2}}}{2}\sum_{j=1}^{L/2}\Big(X_{2j-1}\prod_{k=2j}^{2j+n-2}Z_k\,X_{2j+n-1}+Y_{2j}\prod_{k=2j+1}^{2j+n-1}Z_k\,Y_{2j+n}\Big)\,, & n>0 \text{ even},\\[2mm]
\frac{(-1)^{\frac{n-1}{2}}}{2}\sum_{j=1}^{L/2}\Big(X_{2j-1}\prod_{k=2j}^{2j+n-2}Z_k\,Y_{2j+n-1}-Y_{2j}\prod_{k=2j+1}^{2j+n-1}Z_k\,X_{2j+n}\Big)\,, & n>0 \text{ odd},\\[2mm]
\frac{(-1)^{\frac{n-2}{2}}}{2}\sum_{j=1}^{L/2}\Big(Y_{2j+n-1}\prod_{k=2j+n}^{2j-2}Z_k\,Y_{2j-1}+X_{2j+n}\prod_{k=2j+n+1}^{2j-1}Z_k\,X_{2j}\Big)\,, & n<0 \text{ even},\\[2mm]
\frac{(-1)^{\frac{n+1}{2}}}{2}\sum_{j=1}^{L/2}\Big(X_{2j+n-1}\prod_{k=2j+n}^{2j-2}Z_k\,Y_{2j-1}-Y_{2j+n}\prod_{k=2j+n+1}^{2j-1}Z_k\,X_{2j}\Big)\,, & n<0 \text{ odd},
\end{cases}
$$

$$
G_n =
\begin{cases}
\operatorname{sign}(n)\frac{(-1)^{\frac{n}{2}}}{2}\sum_{j=1}^{L/2}(-1)^j\big(X_jY_{j+n}+Y_jX_{j+n}\big)\prod_{k=j+1}^{j+n-1}Z_k\,, & n \text{ even},\\[2mm]
\operatorname{sign}(n)\frac{(-1)^{\frac{n-1}{2}}}{2}\sum_{j=1}^{L/2}(-1)^j\big(X_jX_{j+n}-Y_jY_{j+n}\big)\prod_{k=j+1}^{j+n-1}Z_k\,, & n \text{ odd}.
\end{cases}
$$

We refer the reader to Ref. 37 for explicit expressions of $Q_n$ and $G_n$ in a different basis.

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
