# Peer review of "Lattice T-duality from non-invertible symmetries in quantum spin chains"

_SciPost Physics, doi:SciPost Phys. 18, 121 (2025)_

## Round 1 · Referee Report · Anonymous (Referee 2) · 2025-2-13

Strengths

  1. A clear derivation of T-duality and $U(1)$ winding symmetry with quantized charge in spin models;
  2. A careful study of the interplay between lattice T-duality, $\mathbb Z_2$ gauging, lattice winding symmetry, anomalies and Onsager algebra.
  3. Application to phase diagrams of XYZ and other models deformed from XX model.

Report

The authors discussed lattice T-duality in the XX model on a spin chain. They derived the T-duality, as a unitary transformation connecting the XX model and the gauged XX model, using the intuition that XX model flows to the compact boson with radius $R=\sqrt{2}$ in the IR.

Using the T-duality, the authors further studied the quantized lattice winding symmetry, a new noninvertible symmetry and the related Onsager algebra.

The authors also gave a careful analysis of the anomaly of various symmetries in the XX model and proved an anomaly-enforced-gaplessness condition for models with $U(1)^{\text{M}}\times U(1)^{\text{W}}$ symmetries. And then they use these theoretical constraints to study the phase diagrams.

Requested changes

The paper is clear and well-organized and does not need significant changes. I am just curious about several questions that I do not understand. It will be helpful is the authors can answer them.

  1. The authors proved the gaplessness using the $U(1)^{\text{M}}\times U(1)^{\text{W}}$ symmetry and what fermionic Hamiltonian looks like with this symmetry. This argument seems very case-dependent and I am curious whether there is a general type of anomaly satisfies the similar condition, different from the most common 't Hooft anomaly that can only preclude the SPTs. And how to characterize such anomaly? Does the Onsager algebra (or integrability) plays a role in the gaplessness?

  2. The authors describe lattice deformations that change the radius of the compact boson but break the lattice $U(1)^{\text{M}}\times U(1)^{\text{W}}$ symmetry and deformations that preserves the lattice $U(1)^{\text{M}}\times U(1)^{\text{W}}$ symmetry. I wonder whether in the latter case the radius is still $R=\sqrt{2}$. Can we find deformations that both change the radius and preserves the radius?

Recommendation

Publish (easily meets expectations and criteria for this Journal; among top 50%)

---

## Round 1 · Referee Report · Anonymous (Referee 1) · 2025-2-13

Strengths

1- A new exact lattice version of T-duality is identified in the XX spin chain. 2- The 't Hooft anomalies among various discrete and continuous symmetries are matched between the lattice and the continuum, and their consequences, including gaplessness, are explored. 3- The rich phase diagram of the XYZ chain and other spin chains that preserve the momentum and winding symmetries are described using these new symmetries and duality.

Weaknesses

1- I feel that the paper is too long, but it is quite readable so this is not a major issue.

Report

It is very interesting to see an exact lattice version of T-duality explicitly in a concrete and well-studied model. I have a few questions/comments:

1- In the introduction (and section 6.2), the authors say that "the existence of $U(1)^m$ and $U(1)^w$ symmetries along with the non-invertible duality symmetry that exchanges them implies T-duality". I think what this means is that any Hamiltonian $H$ that commutes with $Q^m$ and $Q^w$ (and hence also $\mathsf{D}$) satisfies

$$H = U_T H_{/\mathbb Z_2^m} U_T^{-1}.$$
Is my understanding correct? If yes, could the authors clarify why this is the case? I am guessing the starting point is (1.7), but since $\mathsf{D}$ is not invertible, it cannot be cancelled.

2- Footnote 5 is applicable only to the symmetry operators. (There are many continuum operators in section 2, such as the currents $J$, which do not follow this rule.)

3- In the paragraph on anomalies below (1.8), I think it should be $\mathbb Z_2^m \times \mathbb Z_2^C \times U(1)^w$.

4- Is the Onsager algebra in (1.9) related to the $\mathfrak{u}(1)$ Kac-Moody algebra in the continuum? If yes, is there a notion of level (associated to the anomaly) on the lattice? The authors mention [74] in section 3.2.3, and they also say they "leave this for future work" in section 6.2, so this may not be relevant to this paper, but I am curious to know their thoughts.

5- Can the authors say a few more words about the modified translation operator $T\widetilde{T} \prod_j X_j$ below (3.10)? The naive operator $T\widetilde{T}$ also commutes with (3.10), whereas the modified operator maps $Z_j$ to $-Z_{j+1}$. Is there a conceptual way to understand the necessity for the modification other than gauge invariance (i.e., mapping $G_j$ to $G_{j+1}$)?

6- It might be worth noting that the operators in parentheses in (3.11) are CNOT gates.

7- I am intrigued by the fact $U_T^5 = 1$. What does it mean for the orbit of $H_{XX}$ under the action of $U_T$? Acting by $U_T$ once gives $H_{XX/\mathbb Z_2^m}$, but what happens when it's applied again? It cannot be $H_{XX}$ because $2$ and $5$ are relatively prime. So in fact, I think there should be five different Hamiltonians in this orbit. What are they? How are they related to $H_{XX}$?

8- In the paragraph around (6.10), I think it should be $N=3$.

Recommendation

Publish (easily meets expectations and criteria for this Journal; among top 50%)

---

## Editorial Decision

published